# MIROC-INTEG-LAND version 1: A global bio-geochemical land surface model with human water management, crop growth, and land-use change

Tokuta Yokohata[1], Tsuguki Kinoshita[2], Gen Sakurai[3], Yadu Pokhrel[4], Akihiko Ito[1], Masashi Okada[5], Yusuke Satoh[1], Etsushi Kato[6], Tomoko Nitta[7], Shinichiro Fujimori[8], Farshid Felfelani[4], Yoshimitsu Masaki[9], Toshichika Iizumi[3], Motoki Nishimori[3], Naota Hanasaki[1], Kiyoshi Takahashi[5], Yoshiki Yamagata[1], Seita Emori[1]

[1] Center for Global Environmental Research, National Institute for Environmental Studies, Tsukuba 3058506, Japan
[2] Collage of Agriculture, Ibaraki University, Ami 300393, Japan
[3] Institute for Agro-Environmental Sciences, National Agriculture and Food Research Organization, Tsukuba 3058604, Japan.
[4] Department of Civil and Environmental Engineering, Michigan State University, East Lansing, Michigan 48824, USA
[5] Center for Social and Environmental System Research, National Institute for Environmental Studies, Tsukuba 3058506, Japan
[6] Institute of Applied Energy, Minato-ku, Tokyo 105003, Japan
[7] Institute of Industrial Science, The University of Tokyo, Kashiwa 2778564, Japan
[8] Graduate School of Engineering, Kyoto University, Kyoto 6158540, Japan
[9] Graduate School of Science and Technology, Hirosaki University, Hirosaki 0368561, Japan

*Correspondence to*: Tokuta Yokohata (yokohata@nies.go.jp)

## Abstract

Future changes in the climate system could have significant impacts on the natural environment and human activities, which in turn affect changes in the climate system. In the interaction between natural and human systems under climate change conditions, land use is one of the elements that play an essential role. On the one hand, future climate change will affect the availability of water and food, which may impact land-use change. On the other hand, human-induced land-use change can affect the climate system through bio-geophysical and bio-geochemical effects. To investigate these interrelationships, we developed MIROC-INTEG-LAND (MIROC INTEGrated LAND surface model version 1), an integrated model that combines the land surface component of global climate model MIROC (Model for Interdisciplinary Research on Climate) with water resources, crop production, land ecosystem, and land use models. The most significant feature of MIROC-INTEG-LAND is that the land surface model that describes the processes of the energy and water balance, human water management, and crop growth incorporates a land use decision-making model based on economic activities. In MIROC-INTEG-LAND, spatially detailed information regarding water resources and crop yields is reflected in the prediction of

future land use change, which cannot be considered in the conventional integrated assessment models. In this paper, we introduce the details and interconnections of the sub-models of MIROC-INTEG-LAND, compare historical simulations with observations, and identify various interactions between the sub-models. By evaluating the historical simulation, we have confirmed that the model reproduces the observed states well. The future simulations indicate that changes in climate have significant impacts on crop yields, land use, and irrigation water demand. The newly developed MIROC-INTE-LAND could be combined with atmospheric and ocean models to develop an integrated Earth system model to simulate the interactions among coupled natural-human Earth system components.

## 1 Introduction

The problems associated with climate change are related to the various processes involved in natural and human systems, and their interconnections. Changes in the climate system are caused by greenhouse gas emissions and changes in land use resulting from human activity (Collins et al., 2013). At the same time, climate change impacts natural and human systems in a variety of ways (e.g., Arent et al., 2014; Porter et al., 2014; Jiménez-Cisneros et al., 2014; Romero-Lankao et al., 2014). According to research on the linkage of various risks caused by climate change (e.g., Yokohata et al., 2019), changes in the climate system affect the natural environment, leading to changes in the socio-economic system, and finally impacting human lives.

One of the factors that play an essential role in the interaction between the natural and human systems is land use (van Vuuren et al., 2012; Rounsvell et al., 2014; Lawrence et al., 2016). In general, changes in land use are driven by changes in various socio-economic factors, such as an increase in food demand (Foley et al., 2011; Weinzettel et al., 2013; Alexander et al., 2015). At the same time, changes in the climate system affect the water resources available to agriculture and the size of the food supply through changes in crop yield (Rosenzweig et al. 2014; Liu et al. 2016; Pugh et al., 2016), significantly affecting human land use (Parry et al., 2004; Howden et al., 2007). Furthermore, climate mitigation measures often include the use of biofuel crops, which can significantly influence human land use (Smith et al., 2013; Humpenöder et al., 2015; Popp et al., 2017). On the other hand, land-use change is known to have bio-geophysical and bio-geochemical effects on the earth system (Mahmood et al., 2014; Chen and Dirmeyer, 2016; Smith et al., 2016), as changes in land use bring about changes in surface heat and water budget, which, in turn, affects air temperature and precipitation (Feddma et al., 2005; Findell et al., 2017; Hirsch et al., 2018). Changes in land use also affect the terrestrial carbon budget, thereby influencing the concentration of greenhouse gases (GHGs) in the atmosphere (Brovkin et al., 2013; Lawrence et al. 2016; Le Quéré et al., 2018). It seems clear, then, that climate change induces land-use change by affecting various human activities, and that human land-use change affects changes in the climate system (Hibbard et al., 2010; van Vuuren et al., 2012; Alexander et al., 2017; Calvin and Bond-Lamberty 2018, Robinson et al., 2018).

Various numerical models have been developed to describe the interaction between natural and human systems in order to project future conditions as they relate to climate change (van Vuuren et al., 2012; Calvin and Bond-Lamberty 2018).

Generally, in models dealing with the details of natural systems, elements related to human activity are simplified, and in models dealing with the details of human activities, elements related to natural systems tend to be likewise simplified (Muller-Hansen et al., 2018; Robinson et al., 2018). An Earth System Model (ESM) describes in detail the physical and carbon cycle processes in a natural system. A number of ESMs take human activities into consideration (Calvin and Bond-Lamberty 2018). iESM (Collins et al., 2015) is based on a CESM (Community Earth System Model Project, 2019) that incorporates GCAM (Calvin, 2011; Wise et al., 2014), an integrated assessment model (IAM) that provides a comprehensive description of human economic activities. With iESM, it is possible to capture the various interactions between the natural environment and human economic activities (Collins et al., 2015), but the model used to indicate the impact of climate change on water resources and crops is rather simplified (Thornton et al., 2017; Robinson et al., 2018; Calvin and Bond-Lamberty 2018).

IAMs consider supply and demand equations across the entire range of economic transactions and calculate the changes in surface air temperature resulting from increased GHGs in the atmosphere (Moss et al., 2010). IAMs can also project future changes in human land use (Wise and Calvin, 2011, Letourneau et al., 2012, Hasegawa et al., 2017). In general, however, IAMs simplify processes related to the natural environment (water resources, the ecosystem, crop growth, etc.) (Robinson et al., 2018), and thus do not explore the interactions between the natural and human systems on a spatially disaggregated basis (Alexander et al., 2018).

Many models for predicting changes in human land use have been developed (e.g., Hurrt et al., 2006; Lotze-Campen et al., 2008; Havlik et al., 2011; Wise and Calvin 2011; Meiyappan et al., 2014; Dietrich et al., 2019). Among these, the LPJ-GUESS and PLUMv2 coupled model is able to consider spatially specific interactions between changes in vegetation, irrigation, crop growth, and land use (Warlind et al., 2014; Engström et al., 2016; Alexander et al., 2018). However, LPJ-GUESS (Olin et al., 2015) is a dynamic vegetation model that is incapable of exploring interactions related to physical processes, such as bio-geophysical effects or future changes in water resources. On the other hand, LPJ-mL is a well-established global dynamical vegetation, hydrology, and crop growth model that can also consider the nitrogen and carbon cycle (Rolinski et al., 2018; von Bloh et al., 2018). The output of LPJmL (Bondeau et al., 2007), such as crop yield, land/water constraints, and vegetation and soil carbon, is used in the land use model MAgPIE (Lotze-Campen et al., 2008; Popp et al., 2011; Dietrich et al., 2013; Kriegler and Lucht 2015; Dietrich et al., 2019). Although the gridded information of LPJmL is linked to MAgPIE (Alexander et al., 2018), the land-use change calculated by MAgPIE is not communicated to LPJmL (one-way coupling), making interactive calculations using the dynamic vegetation, hydrology, crop growth, and land use models impossible.

In this study, we develop a global model that can evaluate the spatially detailed interactions between physical and biological processes, human water use, crop production, and land use related to economic activities. The model is based on the land surface component of global climate model MIROC (Model for Interdisciplinary Research on Climate version: Watanabe et al., 2010), into which we have incorporated water resources, land-ecosystem, crop growth, and land use models. In the integrated model, which we call MIROC-INTEG-LAND (MIROC INTGrated LAND surface model version 1), the

budgets of energy, water, and carbon are determined by consistently considering the processes related to land surface physics, ecosystems, and human activities.

Section 2 in this paper explains the overall structure of MIROC-INTEG-LAND. The component models of MIROC-INTEG-LAND (climate, land ecosystem, water resource, crop growth, and land use), here called "sub-models", are described in detail in Section 3. Special attention is given to the land use sub-model, as it was specifically developed for inclusion into MIROC-INTEG and is expected to play a pivotal role. The other sub-models—the climate, water resources, crop growth, and land ecosystem models—are based on models developed in the course of previous research. Section 3 outlines how the sub-models used here differ from the original models. Section 4 explains the numerical procedure used to combine the sub-models in the integrated model. Section 5 describes the data used for the various inputs and boundary conditions required to operate the integrated model. Section 6 verifies model reliability by comparing historical simulation results with various observational data. A summary of the results from simulations by MIROC-INTEG-LAND of future conditions and a discussion of the interactions between climate and water resources, crops, land use, and ecosystem are presented in Section 7. Finally, in Section 8, we discuss possible research themes regarding the interaction between natural and human systems that can be addressed using MIROC-INTEG-LAND.

## 2 Overall feature of MIROC-INTEG-LAND

### 2.1 Model structure

The distinctive feature of MIROC-INTEG-LAND (Figure 1) is that it couples human activity models to the land surface component of MIROC, a state-of-the-art global climate model (Watanabe et al., 2010). The MIROC series is a global atmosphere-land-ocean coupled global climate model, one of the models contributing to the Coupled Model Inter-comparison Project (CMIP). MIROC's land surface component, MATSIRO (Minimal Advanced Treatments of Surface Interaction and Runoff, Takata et al. 2003, Nitta et al., 2014) can consider the energy and water budgets consistently on the land grid with a spatial resolution of 1 degree. MIROC-INTEG-LAND performs its calculations over the global land area only, and neither the atmosphere nor ocean components of MIROC are coupled. One of the advantages of running only the land surface model is that it can be used to assess the impacts of land on climate change, taking into account the uncertainties of future atmospheric projections.

Human activity models are included in MIROC-INTEG-LAND: HiGWMAT (Pokhrel et al., 2012), a global land surface model with human water management modules, and PRYSBI2 (Sakurai et al., 2014), a global crop model. In HiGWMAT, models of human water regulation such as water withdrawals from rivers, dam operations, and irrigation (Hanasaki et al., 2006; 2008a; 2008b, Pokhrel et al. 2012a; 2012b) are incorporated into MATSIRO, the above-mentioned global land surface model. In PRYSBI2, the growth and yield of four crops (wheat, maize, soybean, rice) are calculated. In addition, TeLMO (Terrestrial Land-use MOdel), a global land use model developed for the present study, calculates the grid ratio of cropland (food and bio-energy crops), pasture, forest (managed and unmanaged) as well as their transition. The land-use transition

matrix calculated by TeLMO is used in the process-based terrestrial ecosystem model, VISIT (Vegetation Integrative SImulator for Trace gases; Ito and Inatomi 2012).

In MIROC-INTEG-LAND, various socio-economic variables are given as the input data for future projections. For example, domestic and industrial water demand is used in HiGWMAT. The crop growth model PRYSBI2 uses future GDP projections in order to estimate the "technological factor" that represents crop yield increase due to technological improvement. The land use model TeLMO uses future demand for food, bio-energy, pasture, and round wood, as well as future GDP and population estimates. For future socio-economic projections, we use the scenarios associated with Shared Socio-economic Pathways (SSP, O'Neil et al. 2017) and Representative Concentration Pathways (RCP, van Vuuren et al., 2011). These are generated by an integrated assessment model, AIM/CGE (Asia-Pacific Integrated Model / Computable General Equilibrium, Fujimori et al., 2012; 2017).

Interactions of the natural environment and human activities are evaluated through the exchange of variables in MIROC-INTEG-LAND (Figure 1). The calculations in HiGWMAT are based on atmospheric variables (e.g., surface air temperature, humidity, wind, and precipitation) that serve as boundary conditions. The HiGWMAT model calculates the land surface and underground physical variables for three tiles (natural vegetation, rain-fed, and irrigated cropland) in each grid; a grid average is calculated by multiplying the areal weight of the three tiles. In HiGWMAT, water is taken from rivers or groundwater based on water demand (domestic, industrial, and agricultural). Agricultural demand is calculated endogenously in HiGWMAT, and withdrawn water is supplied to the irrigated cropland area, which modifies the soil moisture. The operation of dams and storage reservoirs also modifies the flow of the river. Using the soil moisture and temperature calculated in HiGWMAT, the crop model PRYSBI2 simulates crop growth and yield. PRYSBI2 also uses the same atmospheric variables that are used as input data in HiGWMAT.

The land use model TeLMO uses the yield calculated by PRYSBI2. In TeLMO, the ratios of food and bio-energy crop, pasture, and forest in each grid are calculated based on socio-economic input variables such as the demand for food, bio-energy, pasture, and round wood, as well as crop yield and ground slope. TeLMO also calculates the transition matrix of land usage (e.g., forest to cropland, cropland to pasture), which is passed to the terrestrial ecosystem model VISIT to evaluate the carbon cycle. The land uses calculated by TeLMO are also used as the grid ratios of natural vegetation and cropland area (rainfed and irrigated) in HiGWMAT.

## 2.2 Novelty of MIROC-INTEG-LAND

An important feature of MIROC-INTEG-LAND is that the land allocation model is coupled to the state-of-the-art land surface model, and that the impact of future climate and socio-economic changes on water resources and land use can be considered consistently. In general, future land-use changes are often assessed by using an IAM. However, as mentioned earlier, IAMs are not grid-based, but rather they divide the world into dozens of regions and describes the entirety of economic activity in these regions. Therefore, IAMs has a simplified description of the processes related to water resources and crop growth. In contrast, MIROC-INTEG-LAND provides capabilities to calculate complex physical processes over the

land, and considers the changes in water resources, taking into account human activities such as irrigation and reservoir operation. Furthermore, process-based crop models allow for an explicit and detailed consideration of growth process of five different crops.

For the projection of future land use, IAMs usually 1) calculate the area of agricultural land by using yield information averaged over these regions based on the balance between supply and demand, and 2) allocate the agricultural land by using a downscaling approach (e.g., Hasegawa et al. 2017). As pointed out in previous studies (Alexander et al. 2017), the problem with this method is that it is does not allow an explicit consideration of spatiotemporal information such as yield and production cost when determining land use change. The Food Cropland Model in TeLMO addresses this issue by making it is possible to consistently consider the spaciotemporal information such as crop yields and the balance between supply and demand when allocating the agricultural land, by using the Food Cropland Down-scale Module and the International Trade Module as explained in Appendix B.

As for the projection of future land use change, TeLMO enables the calculation of future land use change as an offline simulation, by using the crop yield data calculated in advance. On the other hand, crop yield depends on water resource availability that is affected by the changes in soil physical processes due to future climate change, as well as the changes in irrigated cropland area caused by the increases in future food demands. MIROC-INTEG-LAND couples the models of land-physical processes, human water management, and crop growth processes with the land-use allocation model to consider these various interactions, as explained above.

## 3 Sub-models

### 3.1 Global land surface model with human water management HiGWMAT

The HiGWMAT model (Pokhrel et al., 2015) is a global land surface model (LSM) that simulates surface and sub-surface hydrologic processes considering both the natural and anthropogenic flow of water globally (1° in latitude and longitude). It incorporates human water management schemes (Pokhrel et al., 2012a; Pokhrel et al., 2012b), into the global LSM MATSIRO (Minimal Advanced Treatments of Surface Interaction and Runoff) (Takata et al., 2003). In MIROC-INTEG-LAND, HiGWMAT calculates the physical states (based on the changes in the energy and water budgets), including human water use and management. In HiGWMAT, the biophysical fluxes are updated after water use and management processes are simulated (Pokhrel et al. 2012a). Since our previous publications provide a detailed description of the MATSIRO model (Takata et al., 2003), groundwater scheme (Koirala et al., 2014), and the human impact representations (Pokhrel et al., 2012a; Pokhrel et al., 2015; Pokhrel et al., 2012b), we include here only a brief overview of these models or schemes.

### 3.1.1 MATSIRO land surface model

MATSIRO (Takata et al., 2003, Nitta et al. 2014) was developed at the University of Tokyo and the National Institute for Environmental Studies in Japan as the land surface component of the MIROC (K-1 Model Developers 2004; Watanabe et al.,

2010) general circulation model (GCM) framework. MATSIRO estimates the exchange of energy, water vapor, and momentum between the land surface and the atmosphere on a physical basis. The effects of vegetation on the surface energy balance are calculated based on the multilayer canopy model of Watanabe (1994) and the photosynthesis-stomatal conductance model of Collatz et al., (1991) following the scheme in the SiB2 model (Sellers et al., 1996). The vertical movement of soil moisture is estimated by numerically solving the Richards equation (Richards, 1931) for soil layers in the unsaturated zone. The original version of MATSIRO (Takata et al., 2003) did not include an explicit representation of water table dynamics. To represent surface and subsurface runoff processes, a simplified TOPMODEL (Beven and Kirkby 1979; Stieglitz et al., 1997) is used. The surface heat balances are solved by an implicit scheme at the ground and canopy surfaces in the snow-free and snow-covered portions (i.e., four different surfaces within a grid cell) to determine ground surface and canopy temperature. The temperature of snow is prognosticated by using a thermal conduction equation, and the snow water equivalent (SWE) is prognosticated by using the mass balance equation considering snowfall, snowmelt, and freeze. The number of snow layers in each grid cell is determined from SWE. The albedo of snow in the model is varied using an aging factor (Wiscombe and Warren 1980) and in accordance with the time since the last snowfall and snow temperature, considering the densification, metamorphism, and soilage of the snow.

### 3.1.2 Human water management schemes

The original MATSIRO was enhanced by Pokhrel et al., (2012a; 2012b) through the incorporation of a river routing model and human water management schemes (i.e., irrigation, reservoir operation, water withdrawal, and environmental flow requirement). The irrigation scheme is based on the soil moisture deficit in the top 1 m (i.e., the root zone) of the soil column; that is, irrigation demand is estimated as the difference between the target soil moisture set for each crop type and the actual simulated soil moisture (Pokhrel et al., 2012b). Irrigation water is added as sprinkler irrigation on top of vegetation, part of which is lost as evapotranspiration and the rest returns back to the soil column. Subgrid variability of vegetation is represented by partitioning each grid cell into three tiles: natural vegetation, and rain-fed and irrigated cropland.

The crop growth module for irrigation water is based on the H08 model (Hanasaki et al., 2008a, 2008b), where the crop vegetation formulations and parameters are adopted from the Soil and Water Integrated Model (SWIM) (Krysanova et al., 1998). The crop growth module for irrigation water in HiGWMAT estimates the cropping period that is necessary to obtain mature and optimal total plant biomass for 18 different crop types. Irrigation is activated during the entire growing season, but only for the irrigated portion of a grid cell using a tile approach (Pokhrel et al., 2012a).

Crop growth considered in the irrigation scheme is simulated within the HiGWMAT model using a crop growth module, which differs from the crop scheme in PRYSBI2 that simulates crop yields (Section 3.2). The reasons why different crop models are used to calculate irrigation water (HiGWMAT) and crop yields (PRYSBI2) are that 1) HiGWMAT has been used a crop model based on SWIM, and it has been validated that the water withdrawal in various regions is consistent with the statistical data (Pokhrel et al. 2014), and 2) PRYSBI2 has been used a crop model based on SWAT, and crop yield in PRYSBI2 has been calibrated using the agricultural statistics (Sakurai et al. 2014). MIROC-INTEG-LAND uses different

crop models to obtain realistic water withdrawal in HiGWMAT and to calculate realistic crop yields in PRYSBI2. The differences in the formulation between the crop models in PRYSBI2 and HiGWMAT are that the former uses more detailed crop modeling of the two-layer crop canopy, Farquhar photosynthetic $CO_2$ assimilation, and the cropping period based on Sacks et al. (2010) (see details in Appendix A.2), while the latter employs the simpler crop modeling of the single-layer crop canopy, radiation-use efficiency-type biomass accumulation, and the hypothetical planting date that gives the highest yield under the given weather conditions (Okada et al. 2015).

The reservoir operation and environmental flow requirement schemes are based on the H08 model (Hanasaki et al., 2008a, 2008b). The reservoir operation scheme (Hanasaki et al., 2006) is integrated within the TRIP global river routing model (Oki and Sud, 1998) to simulate reservoir storage and release for grid cells that contain reservoirs. The reservoir database is taken from Lehner et al. (2011). Large reservoirs having a storage capacity greater than $1 km^3$ are explicitly simulated; medium-sized reservoirs with a storage capacity ranging from $3 \times 10^6$ to $1 \times 10^9 \, m^3$ (Hanasaki et al., 2010) are considered as ponds holding water temporarily and releasing it entirely during the dry season. The withdrawal module extracts the total (domestic, industrial, and agricultural) water requirements, first from river channels and surface reservoirs and then from groundwater; the lower threshold of river discharge prescribed as the environmental flow requirement is considered when extracting water from river channels. While irrigation demand is simulated by the irrigation module, domestic and industrial water uses are prescribed based on the AQUASTAT database of the Food and Agricultural Organization (FAO; see Pokhrel et al., 2012b). We use the same prescribed values for domestic and industrial water uses in both historical and future simulations, as future projections of water withdrawal are not available.

## 3.2 Global crop growth model PRYSBI2

PRYSBI2 (Process-based Regional-scale crop Yield Simulator with Bayesian Inference 2) (version 2.2) is a semi-process-based global-scale crop growth model in which the daily biomass growth and resulting crop yield are calculated for the same grid cell as HiGWMAT (1° in latitude and longitude) (Sakurai et al., 2014). In MIROC-INTEG-LAND, PRYSIB2 is used to calculate crop yields. The target crops are maize, soybeans, wheat, and rice. Daily biomass growth is calculated using daily meteorological data (precipitation, temperature, wind speed, humidity, solar radiation and atmospheric $CO_2$ concentration) according to the photosynthetic rate calculated by a simple big leaf model (Monsi & Saeki 1953) and the enzyme kinetics model developed by Farquhar et al., (1980). To determine the water stress, the soil moisture and temperature calculated by HiGWMAT (Section 3.1) are used. In PRYSBI2, the planting date is given by using the data of Sacks et al. (2010). The harvesting date is determined by when the crops accumulate their total number of heat units (THU) up to the threshold values. Crop yields for each year are calculated from the above-ground biomass and harvest indexes (Appendix A.2).

The process of fertilizer input is not included in this model. Rather, parameters relating to technological factors that include the effect of fertilizer are set and input into the model (Appendix A.7). We call this model a semi-process-based model because some of the parameters, including the parameters relevant to technological factors, are statistically estimated using historical crop yield data (Iizumi et al., 2014) for each grid cell by the DREAM (DiffeRential Evolution Adaptive

Metropolis) algorithm (Vrugt et al., 2009). The parameters were estimated by Markov chain Monte Carlo methods (MCMC) with 20,000 steps for each grid cell (Sakurai et al., 2014). The parameter values of the technological factors in future scenarios are estimated as a linear function of the Gross Domestic Products (GDPs) of each Shared Socio-economic Pathway (SSP) for each country (see details in Appendix A.7).

In the original photosynthesis model by Farquhar et al., (1980), the photosynthesis rate is directly stimulated by the increase of $CO_2$ concentration, which is called the $CO_2$ fertilization effect. However, it is also known that the $CO_2$ fertilization effect is downregulated by environmental limitations such as sink-source balance and nitrogen supply (Ainthworth and Long 2005). In this model, the downregulation of the $CO_2$ fertilization effect is described as a function of atmospheric $CO_2$ concentration, in which the potential photosynthesis rate (maximum carboxylation rate of Rubisco and the

potential rate of electron transport) gradually decreases according to the increase of $CO_2$ concentration (see Appendix A.6).

The crop model used in this study is an updated version (version 2.2) of the model described in Sakurai et al., (2014) (which gives a detailed description of PRYSBI2 version 2.0) and Müller et al., (2017) (which gives a brief description of version 2.1). The structure of the model is quite similar to versions 2.0 and 2.1. However, there are some parts of the version 2.2 structure that are slightly different. In Appendix A, we present a summary of the model and identify the elements that

differ from the earlier versions.

### 3.3 Global land ecosystem model VISIT

The functions of the natural land ecosystem and their environmental responses are simulated by the sub-model VISIT (Vegetation Integrative SImulator for Trace gases) (Ito 2010; Ito et al., 2018). In MIROC-INTEG-LAND, VISIT is used to calculate the carbon and nitrogen cycles. VISIT is a process-based terrestrial biogeochemical model that simulates the

atmosphere-land surface exchange of greenhouse gases such as $CO_2$ and $CH_4$ and trace gases such as biogenic volatile organic compounds. Carbon, nitrogen, and associated water cycles are fully simulated in the model using ecophysiological relationships, but in a simplified manner. The model operates at the global scale with a spatial resolution of $0.5° \times 0.5°$. The ecosystem carbon cycle is simulated using a box-flow scheme composed of three plant carbon pools (leaf, stem, and root) and two soil carbon pools (litter and humus). Photosynthetic carbon acquisition is a function of the leaf area index, light

absorptance, and photosynthetic capacity, which respond to temperature, ambient $CO_2$, and humidity. Soil carbon dynamics are simplified by the litter-humus scheme but works well to simulate microbial decomposition and carbon storage. The model has two layers, i.e., natural vegetation and cropland, at each grid that are weighted by a landcover fraction to obtain the total grid-based budget. Impacts of land-use change on the ecosystem carbon budget are taken into account using a simple scheme by McGuire et al., (2001) in which typical fractionation factors are applied to deforested biomass (e.g.,

immediate emission, 1-yr 10-yr and 100-yr pools). The difference in carbon emissions from primary and secondary forests is included by using a different biomass density; regrowth of abandoned croplands is also simulated as the recovery of the mean biomass of the natural vegetation in the same grid. For brevity, croplands are categorized into three types (rice paddy,

other $C_3$ crops such as wheat, and $C_4$ crops such as maize); the crop calendar and management practices such as fertilizer input are simulated within the VISIT model (i.e., independent of PRYSBI2) in a conventional manner. Planting and harvest dates are determined by monthly mean temperature; country-specific fertilizer inputs derived from the FAO country statistics (FAOSTAT, FAO 2019) are used. In PRYSBI2, the effects of fertilizer are included in the technological factors, and crop yields are calibrated based on the technological factors, As described in Section 3.2 and Appendix A.7. On the other hand, VISIT has been applied and validated at various scales from flux measurement sites to the global scale (e.g., Ito et al., 2017) based on the treatment of fertilizer input, as described above. The consistent treatment of fertilizer processes in PRYSBI2 and VISIT should be important future work.

### 3.4 Land use model TeLMO

In the course of developing the integrated terrestrial model MIROC-INTEG-LAND, we developed the Terrestrial Land-use MOdel (TeLMO) for projecting global land use with a resolution of 0.5°×0.5°. TeLMO projects land use in each grid cell based on socio-economic data such as demand for food and biofuel crops obtained from the AIM/CGE (Fujimori et al., 2012, 2017). In MIROC-INTEG-LAND, TeLMO is used to estimate land use change. For long-term projections, TeLMO assumes that there is a preferential order to land use by humans (i.e., urban, food cropland, bio-energy cropland, pasture land, and managed forests). That is, it assumes that land is used in the order of highest to lowest value added per unit area. After allocating land use in this manner, TeLMO calculates a transition matrix for each grid in order to evaluate the impact of land-use change on terrestrial ecosystems. Details of the five models comprising TeLMO—(1) the food cropland model, (2) the bio-energy cropland model, (3) the pastureland model, (4) the managed forest model, and (5) the land-use transition matrix model—are explained in Appendix B.

### 4 Numerical procedure of model coupling

In MIROC-INTEG-LAND, sub-models with different time-steps are executed simultaneously by exchanging variables as shown in Figure 1. The numerical procedure for exchanging variables between the sub-models is shown in Figure 2. Exchanging variables among sub-models is accomplished in one of two ways: on-line coupling or off-line coupling (Collins et al., 2015). In on-line coupling, the values calculated by a sub-model are exchanged with other sub-models via internal memory (i.e., the values calculated in one subroutine are passed directly to other subroutines). In off-line coupling, the output of a particular sub-model is written to a file; the other sub-models then read the file as needed. The far-right "Data" box in Figure 2 indicates the files used for saving sub-model output data. The arrows show the exchanges that are made. The arrows between one sub-model box and another indicate on-line coupling; those between a sub-model box and the data box indicate off-line coupling. The flow of sub-model calculations is described below.

### (1) TeLMO

The land use model TeLMO (Section 3.4) calculates the areal fraction of each land use within a grid (natural vegetation, cropland, pasture, etc.) and the transitions among them once a year, using the decadal average of crop yields calculated by PRYSBI2. The start year of TeLMO calculation is 2005. Since the exchange of variables is not so frequent, TeLMO is coupled to the other models via off-line coupling (as shown in Figure 2). That is, the output of TeLMO (grid fraction of land uses and transitions) is written to files, and the other sub-models read the files as necessary. As shown in the figure, TeLMO reads the output files of PRYSBI2 (crop yields) for its calculations.

**(2) HiGWMAT + PRYSBI2**

HiGWMAT (Section 3.1), the global land surface model that considers human water management, is used to calculate the physical states (surface and soil temperature and moisture, as well as energy and water fluxes) at hourly to daily time steps. The crop model PRYSBI2 (Section 3.2) is used to calculate crop yields at daily time steps using the soil moisture and temperature values generated by HiGWMAT. Since the exchange of variables between HiGWMAT and PRYSBI2 is very frequent (i.e., daily), these two sub-models are joined through on-line coupling.

As shown in Figure 2, in the future simulations, the MIROC-INTEG-LAND calculations start with TeLMO (TeLMO is switched off before 2004). After the output of TeLMO is written to files, the online-coupled HiGWMAT and PRYSBI2 make their calculations using the land use grid ratio produced by TeLMO. Once the output of the HiGWMAT-PRYSBI2 combination is written to files, TeLMO again starts it calculations for the next year using the 10-yr output. The exchange continues in this fashion.

**(3) VISIT**

As shown in Figure 2, VISIT (Section 3.3), the terrestrial ecosystem model, calculates the carbon and nitrogen cycles using the output of the land use model TeLMO. In MIROC-INTEG-LAND, no variable exchange between HiGWMAT-PRYSBI2 and VISIT is performed at this stage since the structures of these two sub-models differ significantly. In the current version of MIROC-INTEG-LAND, we first calculate the TeLMO-HiGWMAT-PRYSBI2 until 2100, and then perform the VISIT calculations from preindustrial time (including spin-up simulations) to the end of the 21st century by using the TeLMO output (TeLMO is used only for the future period, and LUH data is used for other periods.)

**(4) Model coupling**

The proper choice of coupling method depends on the specific features of the variable exchange between sub-models (Collins et al., 2015). One of the advantages of off-line coupling is that the structure of the original model (e.g., the relationships between the main program and the subroutines) can be preserved, at least to some extent, in the coupling. This is not the case for on-line coupling. For example, for on-line coupling, either the main program of the original model needs to be modified in order for it to serve as a subroutine, or a special program for connecting stand-alone models (i.e., a coupler) needs to be developed. In MIROC-INTEG, off-line coupling is suitable for coupling TeLMO since the model

structure of TeLMO is different from the other sub-models (TeLMO solves equations with various spatial resolution: global 30 sec., 0.5 deg., and 17 regions. See Appendix B for details) and data exchange occurs only once per year (so that the calculation cost for the input/output procedure can be minimized). On the other hand, on-line coupling is appropriate for connecting HiGWMAT and PRYSBI2, since the structure of the two sub-models is similar (spatial resolution with a global 1° grid), and the exchange of variables is frequent (daily). In MIROC-INTEG, some of the subroutines of the original PRYSBI2 models that calculate the crop growth processes are called from HiGWMAT.

## 5 Experimental settings

Since MIROC-INTEG-LAND is based on a global land surface model, atmospheric boundary data (hereafter "forcing" data) are required to operate the model. The global land surface model with human water management HiGWMAT uses atmospheric temperature, humidity, wind, and surface precipitation as the forcing data to calculate the physical processes. In this study, we use forcing data from the Inter-Sectoral Impact Model Inter-comparison Project (ISIMIP) fast track (Hempel et al., 2013). In ISIMIP, historical and future climate simulations by five global climate models (GCMs) with bias correction are used as the distributed forcing data. The methodology of bias correction is described in Hempel et al., (2013). The five GCMs include GFDL-ES2M (Dunne et al., 2012), HadGEM2-ES (Jones et al., 2011), IPSL-CM5A-LR (Dufresne et al., 2012), Nor-ESM (Bentsen et al., 2012), and MIROC-ESM-CHEM (Watanabe et al., 2011). Uncertainties in the atmospheric predictions of the model can be considered by using the output data from the various GCMs. In ISIMIP data, correction for model bias is based on historical observations (Hempel et al., 2013). Thus, we can expect that over- and underestimation errors are removed, at least to some extent.

Since the time interval in the original ISIMIP data is daily and the time step in the land surface model HiGWMAT is sub-daily, we generated three-hourly data from the ISIMIP fast track daily data, based on the methods described in Debele et al., (2007) and Willet et al., (2007), where diurnal variations are generated based on the daily mean data.

In order to obtain a stable state of model variables, we performed spin-up simulations following the procedure defined in the ISIMIP fast track protocols. We first generated de-trended 20-year data using 1951-1970 forcing data. The 20-year dataset was then replicated and assembled back-to-back to obtain an extended dataset. The order of years was reversed in every other copy of the 20-year block in order to minimize potential discontinuities in low-frequency variability. The time duration of the spin-up simulations was 400 years for the land surface model HiGWMAT and the crop growth model PRYSBI2, and 3000 years (repeated 100 times using the first 30-years de-trended climate) for the terrestrial ecosystem model VISIT. The spin-up time of VISIT is longer than that for the other sub-models because it requires more time to reach a stable state, especially in the case of soil organic carbon.

After the spin-up simulations, we performed historical (1951-2005) and future (2006-2100) simulations based on the ISIMIP fast track protocols. For the future simulations, we used the forcing data of the five global climate models based on

four RCPs (van Vuuren et al., 2011)—RCP2.6, 4.5, 6.0, and 8.5—corresponding to radiative forcings of 2.6, 4.5, 6.0, and 8.5 $Wm^{-2}$ in 2100, respectively.

In the historical simulations of HiGWMAT, we used the land use data (grid ratio of natural vegetation, rainfed and irrigated cropland) provided by the Land Use Harmonized (LUH) project (LUHv2h, Lawrence et al., 2016): TeLMO was
switched off. In the future simulations of HiGWMAT, the rainfed and irrigation cropland area is varied according to the output of TeLMO (Section 3.4). Since TeLMO projects the future total cropland area (irrigated plus rainfed), the future irrigated area is calculated by multiplying the grid irrigation ratio (irrigated / [rainfed + irrigated]) and the total cropland area calculated by TeLMO. The grid irrigation ratio is calculated by using the irrigated and rainfed cropland area determined by LUHv2h in 2005 and is fixed throughout the future simulation period. Although TeLMO also calculates the future bio-
energy cropland area, we assume that bio-energy cropland is all rainfed.

TeLMO starts its calculations in 2005. As input data for TeLMO, we use the output variables based on the Shared Socio-economic Pathways (SSPs, O'Neil et al., 2017) calculated by an integrated assessment model, AIM/CGE (Fujimori et al., 2017b). In this study, we use outputs of SSP2 scenario calculated by AIM/CGE (Fujimori et al. 2017b). Since RCP8.5 scenario is not available in SSP2, we use the output of baseline scenario by AIM/CGE for the calculation of RCP8.5.
TeLMO uses future projections of GDP per capita, demand for food and bio-energy crops, pasture, and round wood (Section 3.4, Appendix B). AIM/CGE calculates the aggregated transactions associated with the activities of economic actors; the energy system is represented in detail by dividing the globe into 17 regions (Fujimori et al., 2012).

The terrestrial ecosystem model VISIT is forced by the same ISIMIP forcing data used in HiGWMAT (Hempel et al. 2013). In the historical simulations, VISIT uses the historical land use data from LUHv2h (Lawrence et al., 2016), as
described above. In the VISIT future simulations, the output variables calculated by TeLMO, such as land use (cropland, pasture, forest) and the transition matrix describing transitions from one use to another (see Section 3.4 for details) are used as the forcing data.

It should be noted that the socio-economic scenario that is used in climate forcing data by ISIMIP fast track (Hempel et al. 2013) does not match exactly the SSP scenarios (O'Neil et al. 2017), because the former is based on CMIP phase 5 (CMIP5,
Taylor et al. 2012) and the latter on CMIP phase 6 (CMIP6, Eyring et al. 2016). This should not be a serious problem because the atmospheric processes are not coupled, and the radiative forcing (i.e., the RCP scenarios) used in ISMIP fast track and the SSP scenarios is consistent. The ISIMIP phase three (ISIMIP3, https://www.isimip.org/protocol/#isimip3b), which recently started distributing the climate forcing data, uses CMIP6 GCM simulations based on the SSP scenarios and is consistent with the present study.

## 6 Historical simulations and comparisons with observations

### 6.1 HiGWMAT

Offline simulations from the original MATSIRO and HiGWMAT models have been extensively validated with ground- and satellite-based observations of various hydrologic fluxes and forms of storage (e.g., river discharge, irrigation water use, water table depth, and terrestrial water storage (TWS)) at varying spatial domains and temporal scales in numerous global-scale studies (Felfelani et al., 2017; Pokhrel et al., 2016; Pokhrel et al., 2017; Pokhrel et al., 2012a; Pokhrel et al., 2015; Pokhrel et al., 2012b; Veldkamp et al., 2018; Zaherpour et al., 2018; Zhao et al., 2017). For completeness, we provide here a brief evaluation of TWS and irrigation simulations, since TWS is an indicator of overall water availability in a region and a primary determinant of terrestrial water fluxes (e.g., ET and river discharge), and irrigation is an important component of the global freshwater systems that share the largest fraction of human water use globally (Hanasaki et al., 2008a; Pokhrel et al., 2016). Figure 3 plots the comparison of simulated TWS with observations by the Gravity Recovery and Climate Experiment (GRACE) satellite for the 2002-2005 period. The results shown are spatial averages over 18 major global river basins selected by considering a wide coverage of geographical and climate regions (Felfelani et al., 2017; Koirala et al., 2014). For the GRACE data, we use the mean of mass concentration (mascon) products from the Center for Space Research (CSR; Save et al., 2016) at the University of Texas at Austin and the Jet Propulsion Laboratory (JPL; Watkins et al., 2015; Wiese, Yuan, et al., 2016) at the California Institute of Technology. It is evident from Figure 3 that the model accurately captures the temporal variations as well as the seasonal cycle of TWS in most basins. Certain difference between model and GRACE can be seen in basins such as the Brahmaputra, Huanghe, and Volga river basins but such disagreements have been commonly reported in the literature owing to limitations in model parameterizations in simulating TWS components (e.g., the representation of snow physics and human activities) and inherent uncertainties in GRACE data (Felfelani et al., 2017; Scanlon et al., 2018; Chaudhari et al., 2019).

Figure 4 compares the irrigation water demand simulated by MIROC-INTEG-LAND with the results from offline HIGWMAT simulation obtained from Pokhrel et al., (2015), which is forced by the observed climate data. It is evident from this comparison that the broad spatial patterns seen in the offline simulations are clearly captured by MIROC-INTEG-LAND. Certain disagreements are, however, apparent. For example, MIROC-INTEG-LAND tends to overestimate irrigation demand over highly irrigated areas in the central United States, northwestern India, parts of Pakistan, and northern and eastern China, which is likely due to the drier and warmer climate simulated by the MIROC (Watanabe et al. 2010) in these regions. The total global irrigation demand simulated by MIROC-INTEG-LAND is 1,750 km3, which is greater than the $1,238 \pm 67$ km3 from the offline simulations but falls near the upper bound of estimates by various other global studies (see Table 1 in Pokhrel et al., 2015). The overestimation comes primarily from the highly irrigated regions noted above. Given that our meteorological forcing data are from GCM simulations, we consider our results for both TWS and irrigation demand to be acceptable.

**6.2 PRYSBI2**

Figure 5 shows historical simulation results for crop yield using ISIMIP forcing data as the baseline climate during the period from 1981 to 2005. The historical simulation results were compared with the gridded global data set of historical yield (Iizumi et al., 2017), which is a hybrid of satellite-derived vegetation index data and FAOSTAT (FAO 2019). The spatial aggregation to the country scale was conducted by using the harvested area (Monfreda et al., 2008). The area of wheat was separated into spring and winter wheat by using their production proportions (The United States Department of Agriculture, 1994).

The results of the comparison in the crop yields show the simulated yields in most countries were underestimated to some degree (Figure 5). Notably, using Watch Forcing Data as the reference data in the bias correction for the ISIMIP dataset tends to underestimate solar radiation compared to the observation data (Iizumi et al., 2014; Famien et al., 2018), which in turn causes an underestimation of crop yields. The uncertainty of the projected yields as measured by the differences in outcomes for the five climate forcings was relatively small. The reason for this is that ISIMIP climate forcing data were bias-corrected using the same historical weather dataset and the same method. For all crops, most of the relationship between the simulated and reported data was distributed along the 1:1 line. These results indicate that the model is capable of capturing the relative spatial difference of long-term average crop yield across countries.

**6.3 VISIT**

The VISIT model captured the spatial and temporal patterns of terrestrial ecosystem productivity and carbon budget with satisfactory accuracy. Figure 6 shows the latitudinal distribution of gross primary production for the 2000-2010 period in comparison to up-scaled flux measurements (Beer et al., 2010) and satellite observation (Zhao et al., 2005). High productivity in the humid tropics and low productivity in the arid middle-latitudes and arid cold high-latitudes were effectively reproduced by the model simulation, although mean global total GPP was slightly higher than the observation (127.5 Pg C yr$^{-1}$ by VISIT, 114.0 Pg C yr$^{-1}$ by flux upscaling, and 121.7 Pg C yr$^{-1}$ by satellite). Global carbon stocks in vegetation and soil organic matter were estimated as 499 and 1308 Pg C, respectively, in 2010; this is comparable to the contemporary synthesis (Ciaes et al., 2013). Because of historical atmospheric $CO_2$ rise, climate change, and land-use change, substantial changes in terrestrial ecosystem properties were simulated (not shown). As demonstrated by model validation and inter-comparison studies, the VISIT model allows us to effectively capture the terrestrial ecosystem functions under changing environmental conditions.

**6.4 TeLMO**

In Figure 7, the cropland area simulated by TeLMO in MIROC-INTEG-LAND is compared with the cropland area reported in FAOSTAT (FAO 2019) and to the area simulated by AIM/CGE (Fujimori et al., 2017b), whose output of food demand and GDP per capita is used as input in TeLMO. Output of the TeLMO 0.5° grid data is aggregated by country to facilitate

comparison with the FAOSTAT data. In order to also compare the TeLMO 0.5° grid data with the AIM/CGE cropland area, we used a 0.5° downscaled land use data based on the AIM/CGE calculation (the methodology of downscaling is described in Fujimori et al. 2017a). With the adjustment parameter $C_j$, the cropland area in TeLMO in 2005 is the same as that of LUH (Lawrence et al., 2016). As shown in Figure 7, MIROC-INTEG-LAND roughly reproduces the cropland area by country shown in FAOSTAT (FAO 2019). The differences in the five climate forcings given to MIROC-INTEG-LAND cause variance in crop yields, which in turn results in the variance in cropland area results shown in Figure 7.

In Russia, Brazil, and Australia, the recorded cropland area (i.e., FAOSTAT) is within the range of the MIROC-INTEG-LAND cropland area simulations using the different climate forcings. In Brazil and Russia, the variations in cropland area are mainly due to the difference in climate forcings. In the United States, the reported cropland area in FAOSTAT (FAO 2019) is closely reproduced by MIROC-INTEG-LAND until around 2010; however, the declining trend of cropland area in the second half is not effectively reproduced. The reason for the overestimation seen here may be related to the under-estimating of crop yield in PRYSBI2 (Section 6.3). The slight overestimation of the global cropland area trend (Figure 7h) may stem from the same cause. Also, in China, although there is a declining trend of cropland area in MIROC-INTEG-LAND, in reality, the cropland area remained nearly constant until 2014 and increased slightly thereafter. The increase of cropland area in China is considered to be influenced by policy, which is not considered in TeLMO.

In MIROC-INTEG-LAND, TeLMO uses the food demand and GDP per capita calculated by AIM/CGE under the socio-economic scenario SSP2 (Fujimori et al., 2017b). Therefore, the difference between TeLMO and AIM/CGE is due to the difference in crop yield as well as the mechanism for the allocation of agricultural land. As explained in Appendix B.1, TeLMO can consider the spatial distribution of crop yield when allocating agricultural land. On the other hand, in AIM/CGE, land use change is calculated by aggregating crop yield information in the regions where the model calculation is performed (AIM/CGE divides the world into 17 regions). In large countries such as Australia, Brazil and Russia, the allocation method in TeLMO shows good performance.

Figure 8 shows a comparison of TeLMO, AIM, and LUH data for pasture. Unlike cropland, pastures are compared with LUH data because there are no long-term global observation data. TeLMO calculates pasturelands such that the area matches that in the AIM for the AIM calculation domain (17 regions around the world). Because AIM treats China and the United States as one region, the results of TeLMO and AIM for China, the United States, and the globe are almost the same. On the other hand, in Australia, TeLMO is closer to LUH. Similarly, Figure 9 shows a comparison between TeLMO, AIM, and FAO data of forest area. TeLMO refers to MODIS data and calculates forest area taking into account deforestation and changes in crop area. Some differences can be seen between TeLMO and FAO, probably because TeLMO refers to MODIS and not to FAO; however, the differences are relatively small. Given that its performance is similar to that of AIM/CGE, the TeLMO sub-model in MIROC-INTEG-LAND can be considered useful for future land use prediction.

## 7 Future simulations and interaction of sub-models

In the MIROC-INTEG-LAND future simulations, the RCP2.6, 4.5, 6.0, and 8.5 scenarios provided by ISIMIP1 (Hempel et al. 2013) serve as the climate scenario, while the output of AIM/CGE (demand for food and bioenergy crops, pasture, wood, etc.) according to the four RCPs under SSP2 (Fujimori et al. 2017b) serves as the socio-economic scenario. The results in this section provide an understanding of the interactions between climate, water resources, crops, ecosystems, and land use that MIROC-INTEG-LAND accommodates.

Figure 10 shows the various time series related to climate system change. Figure 10a depicts the change in surface air temperature used as forcing data in MIROC-INTEG-LAND. It is displayed as the deviation from the average value of the 10-year period around the start year of the future simulations (2005). As shown in Figure 10a, the increase in average global land surface air temperature in 2100 is approximately 6 °C for RCP8.5, 3 °C for RCP6.0, 2.5 °C for RCP4.5, and 1 °C for RCP2.6. Figure 10b shows the change in soil moisture calculated by MIROC-INTEG-LAND. Although the annual variation of soil moisture is considerable, the global land average soil moisture content tends to decrease in the 21st century. The reduction in soil moisture is largest in the RCP8.5 scenario, where the rise in surface air temperature is substantial. Results for the irrigation water supply are shown in Figure10c. As indicated in Section 3.1, water is supplied from rivers to the soil through irrigation until the ratio of soil moisture reaches a certain threshold. The irrigated area is calculated by multiplying the cropland area (as calculated by TeLMO) by the irrigation ratio, a fixed value corresponding to the ratio of irrigation cropland area to the total cropland area in 2005. Therefore, the changes in irrigation water supply in Figure 10c reflect the changes in the irrigation area and the irrigation water supplied from rivers to the soil to compensate for the decrease in soil moisture. Although the global total cropland area increases in the first half of the 21st century (Figure 12), in regions with a high irrigation ratio (e.g., India, China), cropland area decreases by the end of the century (Figure 12). As a consequence, the irrigation area in MIROC-INTEG-LAND decreases, and, accordingly, the irrigation water supply also decreases, as shown in Figure 10c.

Changes in crop yield calculated for the various future scenarios are shown in Figure 11. The crop growth model PRYSBI2 in MIROC-INTEG-LAND can calculate the yields [t / ha] of four crops (wheat, maize, soybean, rice), with a clear distinction between winter and spring wheat (meaning five crops in all). In Figure 11f, the global average of the grid maximum yield value among the crops, which is used in the TeLMO calculation, is also shown. As described in Section 3.2, the future simulations by PRYSBI2 take into account the effects of climate change, as well as the $CO_2$ fertilization effects due to rising greenhouse gas concentrations (Appendix A.6) and the increase in technical coefficients due to future technological improvement (Appendix A.7).

As shown in Figure 11a-e, the yields of each of the crops rise over the first half of the 21st century. This is due to the $CO_2$ fertilization effect and technological improvement. In general, the increase in yield is more significant in the high-GHG scenarios such as RCP8.5 than in the low-GHG scenarios such as RCP2.6. Such differences can be considered due to the fertilization effect and impact of climate change, since all the RCPs feature the same technological coefficient under the

same SSP scenario (i.e., SSP2). On the other hand, in the latter half of the 21$^{st}$ century, the negative impact of climate change on crop yield is evident. In the RCP 8.5 scenario, in particular, crop yields decline sharply. PRYSBI2 results show that the crop type most sensitive to climate change is maize: in 2100, the yield of maize under RCP2.6 is highest, while the yield of maize under RCP8.5 is lowest.

5    Figure 12a shows the change in the food cropland area calculated by TeLMO. As described in Section 3.4 and Appendix B, TeLMO uses the yield calculated by PRYSBI2 (grid maximum value as shown in Figure 11f) and the food demand output of AIM/CGE. As shown in the Figure 12a, crop area increases to meet the increase in food demand in the first half of the 21st century. Compared to other RCP scenarios during this time period, the RCP2.6 scenario requires more food cropland area, since the increase in crop yield is smaller in the RCP2.6 scenario. In the second half of the 21st century, the food cropland area tends to decrease as crop yield increases more than food demand. The decrease is smallest under RCP2.6 and largest under RCP6.0, and RCP8.5 actually requires an increase in food cropland area, as in this scenario, crop yields decline late in the century. Although there are differences among the results using the five different climate model forcings (the thin lines in Figure 12a), using the average value lines (the thick lines in the figure) for comparison indicates that, by the end of the 21st century, the food cropland area is largest under RCP8.5.

15    Figure 12b shows the time series of the sum of food and bioenergy cropland area calculated by TeLMO. As described in Section 3.4, TeLMO calculates the distribution of the global bioenergy cropland area needed to meet the bioenergy demand calculated by AIM/CGE. It is known that the future bioenergy cropland area will change substantially depending on crop yield, and it should be noted that the setting in which crop yield is calculated can significantly affect the bioenergy cropland area (Kato and Yamagata 2014). As shown in Figure 12b, the bioenergy cropland area is significantly increased under RCP2.6 and RCP4.5. These climate scenarios require large areas of bioenergy crops for future climate mitigation. Although the food cropland area tends to decrease in the late 21th century (except in the RCP8.5 scenario), if we consider both food cropland and bioenergy cropland, more cropland area will be needed.

    Figure 13 shows the global distribution of changes in food and bioenergy cropland areas, using the difference in 10-year averages around 2100 and 2005. As described in Figure 12a, RCP 2.6 tends to reduce the food cropland area in the latter half of the 21st century. Figure 13a and 13b show that the food cropland area decreases in Africa, India, and China. As is explained in Appendix B, TeLMO relies on the premise that the distribution of food cropland area is determined by changes in crop yield, food prices, wages (corresponding to changes in GDP per capita) and the demand for food. Thus the decreases in food cropland area shown in Figure 13a and 13b are due to the increase in yield (meaning demand can be met with less cropland area) and the increase in GDP per capita (which means the population engaged in agriculture decreases due to development) in the SSP2 scenario. It should be noted that the change in cropland area at a particular grid is not determined solely by food production (the product of cropland area and crop yield) at that grid, as TeLMO considers the food trade among the 17 regions. As shown in Figure 12 and noted earlier, the food cropland area will increase in the late 21st century in the RCP8.5 scenario. Accordingly, in comparison to the RCP 2.6 scenario, the food cropland area in South America and central Africa increases in the RCP8.5 scenario.

As shown in Figure 13, bioenergy cropland areas increase in various regions, especially in the RCP 2.6 scenario. As discussed in Appendix B, TeLMO assumes that biofuel cropland is allocated based on the Agricultural Suitability Index (Eq. B-14), which is a function of the yield and price of the bioenergy crop, GDP per capita, etc. At the same time, TeLMO also assumes that regions with high biodiversity are protected, and calculations are performed so as not to allocate biofuel cropland to the protected areas as shown in Figure B-2 (Wu et al., 2019). As a result, bioenergy cropland area is allocated to regions where the agricultural index is high—northwest and southern South America, central Africa, and Australia—but it cannot be allocated to protected areas such as the Amazon.

Figure 14 and 15 show the effects of changes in food and bioenergy cropland area on the terrestrial ecosystem calculated by VISIT in MIROC-INTEG-LAND. The impact of land-use change on terrestrial ecosystems is evaluated by comparing the calculation with and without considering the land-use change. The global time sequence (Figure 14) shows that the changes in food and bioenergy cropland area have a significant impact on terrestrial ecosystems, especially in RCP 2.6, where the above-ground biomass will decrease by approximately 50 Pg C (about 10% of the present biomass stock) by 2100 due to deforestation for land use conversion. The decrease in soil carbon after deforestation is much smaller than the decrease in above-ground biomass, as the carbon supply from crop residue compensates for the soil carbon loss. Consequently, this simulation implies that the impacts of land-use change occur heterogeneously and differ in their magnitude and direction between vegetation and soil. Figure 15 shows the global distribution of the effect of land-use change on above-ground biomass and soil carbon. The impact on above-ground biomass is projected to be greater in northwest South America, central Africa, northeast North America, and Australia, where the bioenergy cropland area is expanding. In these regions, even under the mitigation-oriented scenario, considerable declines in ecosystem structure and functions would occur, leading to deterioration, for example, of habitats for natural organisms, water holding capacity, and soil nutrients. Consequently, these functional degradations would degrade ecosystem services such as biodiversity, regulation, and provision. On the other hand, in Asia, the decrease in food cropland area tends to increase the above-ground biomass in both the RCP2.6 and RCP8.5 scenarios, possibly leading to leading to the enhancement of above-ground biomass, and thus ecosystem services.

Figure 16 shows the results of simulations to evaluate the effects of climate change on crop yield, land use, and water demand. In Figure 16, the RCP8.5 simulations with climatic factors (temperature, water vapor, wind speed, soil moisture, soil temperature) and $CO_2$ concentration fixed at 2006 (noCL+noFE), those with fixed climatic factors (noCL+FE), and those with variable climatic factors and $CO_2$ concentrations (CL+FE) are compared. The CL+FE simulations are the same as the RCP8.5 results shown in Figure 11f (crop yield), 12a (food cropland area), and 10c (irrigation demand).

As shown in Figure 16a, the crop yield is significantly larger in the noCL+FE experiment than in the CL+FE experiment. This result indicates that climate change can significantly reduce crop yields. One of the reasons for the observed reduction in crop yield in the CL+FE experiment is that the growing season is shortened due to an increase in surface air temperature, which adversely affects crop growth (Sakurai et al. 2014). The impacts of climate change on crop growth increase with increasing temperature, and in 2100, crop yields in the CL+FE experiment are projected to decrease by approximately 60% relative to the yields in the noCL+FE experiments.

As shown in Figure 16a, the crop yield was much smaller in the noCL+noFE simulations than that in the CL+FE simulations. The reason for the yield in the noCL+noFE experiment being smaller than that in the CL+FE experiment is because the crop yield increases due to the $CO_2$ fertilization effect in the latter. The increase in crop yield in the noCL+noFE experiment is due to technological developments (Section 3.2 and Appendix A.7). Although there is a great deal of uncertainty regarding the treatment of $CO_2$ fertilizer effects in crop models (Sakurai et al. 2014), the increase in crop yields due to the $CO_2$ fertilizer effect is significant in the simulations of MIROC-INTEG-LAND.

Due to the changes in crop yields resulting from the changes in climate and fertilization effects, future cropland area and irrigation demand will also change significantly. In the CL+FE experiment, the food cropland area (Figure 16b) and irrigation demands (Figure 16c) become lager than those in the noCL+FE experiments because of the larger decrease in crop yields due to the impacts of climate change (Figure 16a). On the other hand, the noCL+noFE experiment requires more food cropland area (Figure 16b) and irrigation demand (Figure 16c) compared to the CL+FE experiment, because of the smaller increase in crop yields mainly due to the absence of $CO_2$ fertilization effects (Figure 16a). In summary, the changes in climate and $CO_2$ fertilization effects are expected to have marked impacts on crop yields, land use and water demands in the future.

## 8 Implications and future research

With MIROC-INTEG-LAND, it is possible to calculate the interaction between climate, water resources, crops, land use, and ecosystems. The discussion in Section 7 suggests the type of feedback processes that can occur. As shown in Figure 11, future climate change can affect crop yields. Especially under a scenario of large temperature increases (RCP8.5), crop yields will decrease in the latter half of the 21st century (Figure 11). Here, the influence of the $CO_2$ fertilization effect is also a very important factor affecting future changes in crop yields (Figure 16a). Changes in crop yields due to climate change also have a large impact on cropland area (Figure 12, 16b). Future cropland area may increase in response to an increase in food demand due to population growth, as well as due to increases in biofuel crop cultivation in response to global warming countermeasures. Such an increase in cropland area will cause a concomitant increase in water demand due to an increase in irrigated cropland area (Figure 10, 16c). In addition, an increase in cropland area can affect carbon uptake in terrestrial ecosystems (Figure 14). Increased human water use and changes in terrestrial carbon uptake can further affect the water, crop yields, and carbon budgets on the land surface. A real novelty of MIROC-INTEG-LAND is that the availability of both water and agricultural land can be consistently considered in conjunction with changes in climate conditions.

While this study showed only the results of the SSP2 scenario, in the SSP3 scenario, where the world is divided, the demand for food will be greater and more cropland area will be needed (O'Neill et al., 2017). Investigating the impacts of various natural and socio-economic factors (climate, irrigation, fertilization effects, population, food demands, etc.) on land use change and land ecosystems is an important future research direction as an extension of the present study.

In addition to analyzing interactions, it is crucial to analyze the impacts of climate change and the effectiveness of countermeasures using MIROC-INTEG-LAND. The combined impacts of climate change on water resources, crops, land use, and ecosystems can be mitigated by enhancing various adaptation measures. For example, the use of water resources to control crop yield loss, changes in cropping calendars, and breeding can reduce the adverse effects of climate change on food

and land use. With MIROC-INTEG-LAND, it is possible to assess the efficiency of adaptation measures designed to address the impacts of climate change on water resources, crops, land use, and ecosystems (Alexander et al., 2018). With consistent consideration of climate change, water resources, and land use, the competition between water, food, and bioenergy use can be analyzed (e.g., Smith et al., 2010). The model also provides useful insights into the trade-offs of biodiversity loss from land-use change and the benefits of climate mitigation.

MIROC-INTEG-LAND provides a way to integrate various human activity models based on the global climate model as shown in Section 4. This paper introduced illustrative simulation results produced by our application of MIROC-INTEG-LAND as a land surface model driven by meteorological forcing data. We plan to extend the model by enabling it to consider the physical processes and carbon/nitrogen cycle in the atmosphere and ocean. The MIROC community has developed MIROC-ES2L, an earth system model for CMIP6 (Hajima et al.,2020). By incorporating the water resource

model (HiGWMAT), the crop growth model (PRYSBI2), and the land use model (TeLMO) used in MIROC-INTEG-LAND into MIROC-ES2L, we are developing an integrated earth system model that we call MIROC-INTEG-ES. In MIROC-INTEG-ES, the interactions between the earth system and human activities are consistently considered. By using this integrated earth system model, the impact of land-use changes on the climate system, including bio-geophysical and bio-geochemical effects (Lawrence et al., 2016), can be more consistently investigated.

**Appendix A: Description of crop model PRYSBI2 Version 2.2**

In the following description, we present a summary of the crop model used in MIROC-INTEG-LAND (PRYSBI2 Version 2.2) and identify the elements that differ from the earlier versions (Version 2.0: Sakurai et al., 2014, Version 2.1: Müller et al., 2017).

**A.1 Input data**

As input data, the PRYSIB2 Version 2.2 uses the cropping period based on the planting and harvesting date by Sacks et al. 2010. Soil field capacity (Scholes et al. 2011), and atmospheric data (average, maximum and minimum daily temperature, daily shortwave and longwave radiation, daily humidity, and $CO_2$ concentration) are also used as input data. We use the same atmospheric data as HiGWMAT described in Section 5 (i.e., ISIMIP fast track data by Hempel et al. 2013).

**A.2 Growing period, maturity and harvest**

The time of seedling emergence after the planting date is determined by a parameter relevant to the average period between planting and emergence ($l_{emerge}$). The period from emergence to maturity is determined by the total number of heat units (THU) (Neitsch et al., 2005). The crop is mature when THU is equal to a threshold value ($thu_{total}$), at which point it is harvested. THU thresholds were estimated for each grid by performing calibration between 1980 and 2006, so that harvest dates fit the data from Sacks et al. (2010). If future projections are performed using this threshold value, then the harvest date will deviate from Sacks et al. (2010) because of the temperature rise in future climates (i.e., harvest dates become earlier due to the increase in temperature). Using the biomass values obtained at the time of crop maturity, the yield is calculated as follows:

$$Yield = hi_{base} \cdot BIO_{above(maturity)} \tag{A-1}$$

where *Yield* is the crop yield (kg ha$^{-1}$), $hi_{base}$ is the harvest index, and $BIO_{above(maturity)}$ (kg ha$^{-1}$) is the above-ground biomass at the time of crop maturity. Although the harvest index changes according to atmospheric $CO_2$ concentration in version 2.0, in version 2.2, for simplicity, it is fixed.

**A.3 Photosynthesis**

The photosynthesis processes in version 2.2 are the same as in the previous versions. The photosynthesis rate is calculated according to the daily meteorological data. The instantaneous global radiation and temperature at time (*t*) of the day are estimated from the daily global radiation and daily maximum and minimum temperature on a given day (*td*) according to the method described by Goudriaan and van Laar (1994). The amount of photosynthetically active radiation, $PAR_{t,td}$ (MJ m$^{-2}$ s$^{-1}$), intercepted by the leaf at time *t* on a given day *td* is calculated using Beer's law (Monsi & Saeki 1953). We used the model described by Baldocchi (1994) to calculate the photosynthetic rate.

**A.4 Temperature stress**

The equations for the effects of temperature on the maximum carboxylation rate of Rubisco and dark respiration rate are changed from those in version 2.0. The influence of temperature on the maximum carboxylation rate of Rubisco and the potential rate of electron transport is given as follows (Kaschuk et al., 2012, Medlyn et al., 2002):

$$C_{vcmax(t,td)} = \exp\left[(TM_{t,td} - 25) \cdot \frac{ep_{vcmax}}{298 \cdot R \cdot (273 + TM_{t,td})}\right] \tag{A-2}$$

$$C_{jmax(t,td)} = \exp\frac{E_{jmax}(TM_{t,td} - 25)}{298 \cdot R \cdot (TM_{t,td} + 273)} \cdot \frac{1 + \exp\dfrac{298 \cdot S_{jmax} - H_{jmax}}{298 \cdot R}}{1 + \exp\dfrac{(TM_{t,td} + 273) \cdot S_{jmax} - H_{jmax}}{(TM_{t,td} + 273) \cdot R}} \tag{A-3}$$

where $C_{vcmax(t,td)}$ and $C_{jmax(t,td)}$ represent the effect of temperature on the maximum carboxylation rate of Rubisco and the potential rate of electron transport, respectively; $TM_{t,td}$ is the air temperature (°C) at time *t* on day *td*; $ep_{vcmax}$, $E_{jmax}$, $S_{jmax}$, and $H_{jmax}$ are parameters that describe the shape of the curve (Kaschuk et al., 2012, Medlyn et al., 2002), and R is the universal

gas constant (8.314 J mol$^{-1}$ K$^{-1}$).

The influence of temperature on the dark respiration of leaves is given as

$$C_{dark(t,td)} = \exp\left[(TM_{t,td} - 25) \cdot \frac{ep_{rd}}{298 \cdot R \cdot (273 + TM_{t,td})}\right] \tag{A-4}$$

where $C_{dark(t,td)}$ represents the effect of temperature on dark respiration at time $t$ on day $td$ and $ep_{rd}$ is the parameter that describes the shape of the curve (Kaschuk et al., 2012).

The maximum carboxylation rate of rubisco, the potential rate of electron transport, and the dark respiration rate are modified by temperature effects:

$$V_{cmax(t,td)} = \Theta \cdot \xi_V \cdot C_{vcmax(t,td)} \cdot v_{cmax} \cdot W_{stress(td)} \tag{A-5}$$

$$J_{max(t,td)} = \Theta \cdot \xi_J \cdot C_{jmax(t,td)} \cdot j_{max} \cdot W_{stress(td)} \tag{A-6}$$

where $V_{cmax(t,td)}$ is the maximum carboxylation rate of Rubisco, $J_{max(t, td)}$ is the potential rate of electron transport, $v_{cmax}$ and $j_{max}$ is the potential maximum carboxylation rate and the potential rate of electron transport, respectively. $W_{stress(td)}$ represents water stress, which is explained in A5. $\Theta$ is the compensation variable (0–1) that represents the discrepancy between the ideal photosynthetic potential and the actual one. $\xi_V$ and $\xi_J$ are photosynthesis compensation variables that change according to $CO_2$ concentration. These variables ($\Theta$, $\xi_V$, and $\xi_J$) are described in the following section. The dark respiration rate is calculated as follows:

$$R_{d(t,dt)} = rd \cdot C_{dark(t,td)} \cdot v_{cmax} \tag{A-7}$$

where $R_{d(t,td)}$ is the dark respiration rate (μmol m$^{-2}$ s$^{-1}$), and $rd$ is the leaf respiration factor (Collatz et al., 1991, Sellers et al., 1996a, b). The maintenance respiration and growth respiration are also considered. The formulations of the respiration models are also the same as those of the previous versions.

### A.5 Soil water balance and water stress

In PRYSBI2, the calculation of water stress follows the SWAT (Neitsch et al., 2005) algorithm. In SWAT, the daily water stress is calculated according to soil water, soil characteristics (field capacity and water content at saturation), root depth and crop field evapotranspiration. PRYSBI2 uses the soil water calculated in HiGW-MAT as explained in Section 3.2. The crop field evapotranspiration is calculated in SWAT according to the leaf area index.

### A.6 Correction of parameters according to $CO_2$ concentration

The correction of parameters based on $CO_2$ concentration is included in the model using the following equations:

$$\xi_V = 1 - \frac{r_{\phi 1}(c_a - c_{base}) + r_{max1} - \sqrt{\left(r_{\phi 1}(c_a - c_{base}) + r_{max1}\right)^2 - 4r_\theta r_{\phi 1} r_{max1}(c_a - c_1)}}{2r_\theta} \tag{A-8}$$

$$\xi_J = 1 - \frac{r_{\phi2}(c_a - c_{base}) + r_{max2} - \sqrt{\left(r_{\phi1}(c_a - c_{base}) + r_{max2}\right)^2 - 4r_\theta r_{\phi1} r_{max2}(c_a - c_{base})}}{2r_\theta} \qquad \text{(A-9)}$$

$$r_{\phi1} = \frac{dr_{vcmax}}{c_{base}} \qquad \text{(A-10)}$$

$$r_{\phi2} = \frac{dr_{jmax}}{c_{base}} \qquad \text{(A-11)}$$

$$r_{max1} = dr_{vcmax}\left(\frac{600}{c_{base}} - 1\right) \qquad \text{(A-12)}$$

$$r_{max2} = dr_{jmax}\left(\frac{600}{c_{base}} - 1\right) \qquad \text{(A-13)}$$

where $\xi_V$ and $\xi_J$ are photosynthesis compensation variables, $dr_{vcmax}$ and $dr_{jmax}$ describe the parameters, $c_a$ is atmospheric $CO_2$ concentration (mol mol$^{-1}$), and $c_{base}$ is the baseline atmospheric $CO_2$ concentration (mol mol$^{-1}$). In this model, if $dr_{vcmax}$ and $dr_{jmax} > 0$, $\xi_V$ and $\xi_J$ decrease linearly with increasing atmospheric $CO_2$. If $dr_{vcmax}$ and $dr_{jmax} = 0$, $\xi_V$ and $\xi_J$ do not depend on atmospheric $CO_2$. In these equations, $r_{max1}$ and $r_{max2}$ are the respective asymptotic lines. $r_\theta$ is the parameter that determines

the curvature of the lines; we set $r_\theta = 0.99$. The parameters $dr_{vcmax}$ and $dr_{jmax}$ are based on the results of Ainsworth and Long (2005).

**A.7 Time trend of the parameter relevant to agricultural management**

When using historical yield data to calibrate model parameters, we need to consider temporal trends in the effects of non-climatic factors. Crop yield should improve from year to year because of agricultural factors, such as the decrease in harvest

loss and the use of improved crop cultivars and pesticides. We, therefore, assumed the following linear trend in non-climatic effects when evaluating the long-term yield data:

$$\Theta = \theta_{base} + \theta_{trend}(Year - y_{base}) \qquad \text{(A-14)}$$

where $\Theta$ is the compensation variable (0–1) that represents the discrepancy between the ideal photosynthetic potential and the actual one, which is used in Eq. A-5 and A-6; $\theta_{base}$ is the value of $\Theta$ in year $y_{base}$ and must be calibrated for each cell of the grid; $\theta_{trend}$ is the annual increase in $\Theta$ due to non-climatic factors (which also must be calibrated for each cell of the grid);

*Year* is the year; and $y_{base}$ is the criterion year (2006). In this study, we analyzed the relationship between $\theta_{base}$ and GDP for each crop and used the estimated relationship for future prediction.

**Appendix B: Description of land-use model TeLMO**

**B.1 Food Cropland Model**

For each grid, TeLMO first allocates the area for urban use; it then allocates the area for food cropland. For the allocation of

the urban area, we use the Land Use Harmonization phase 2 future data that are used in Coupled Model Intercomparison

Project Phase 6 (CMIP6) (LUH2f, Lawrence et al., 2016). It is generally expected that the food cropland area is determined by the balance between the supply and demand for food crops. The estimation of the supply potential of food crops requires the spatial distribution of crop production, which is related to the natural environment. On the other hand, the balance between the supply and demand for food crops is influenced by socio-economic factors (e.g., populations, crop prices) related to international food trade. For this reason, TeLMO projects future land-use change by allowing the Food Cropland Down-scale Module (B.1.1), which projects the global cropland distribution at a resolution of 0.5° by considering environmental factors, to interact with the International Trade Module (B1.2), which describes food supply and demand based on the General Equilibrium Model by dividing the world into 17 countries/regions. The primary objective of using TeLMO is to describe the long-term trend in land-use change, not the detailed year-to-year variations in land-use change. Therefore, we use 10-year average values as input to the model.

A major feature of TeLMO is that it does not project the local cropland distribution by unidirectionally downscaling the total cropland area for countries/regions obtained by integrated assessment models. This is because the total cropland area for each country/region depends on the local distribution of the cropland area. Therefore, TeLMO consistently treats the cropland distribution calculated by the Food Cropland Down-scale Module and the total cropland area for countries/regions obtained from the International Trade Module to project future land-use change. The Food Cropland Down-scale Module and International Trade Module are explained below.

**B.1.1 Food Cropland Down-scale Module**

The Food Cropland Down-scale Module divides the Earth into 0.5°×0.5° (latitude×longitude) grid cells (hereinafter "0.5° cells") and calculates the percentage of each cell occupied by cropland. The percentage of cropland is estimated by calculating the probability that each 30″×30″ grid cell (hereinafter "30″ cell") is used as cropland and averaging these probabilities over the entire 0.5° cell. A 30″ cell allocated to urban use is not used for cropland. The probability $r_i$ of a given 30″ cell being used as cropland is calculated as

$$r_i = \frac{1}{1 + \exp(1.228 + 0.237\phi_i - 0.206 p_k y_j / w_k)} C_j \qquad \text{(B-1)}$$

where $\phi$ is the slope, $y$ is the yield per unit area [t/ha], $p$ is the price of food crops, $w$ is the wage, and $C$ is an adjustment parameter. The subscript $i$ identifies the 30″ cell, $j$ identifies the 0.5° cell containing the $i$-th grid cell, and $k$ identifies the country/region containing the $i$-th and $j$-th grid cell. The definition of countries/regions is the same as that used in AIM/CGE (Fujimori et al., 2012, 2017). Eq. (B-1) is formulated based on the fact that the cropland area is determined as a function of slope, crop price and yield, and the wages of farmers. The first term of Eq. (B-1) is defined as the Agricultural Suitability Index (ASI), which represents the relationship between cropland area and the explanatory variables. The adjustment parameter $C_j$ is used to reproduce the cropland area of LUH (Lawrence et al., 2016) in the base year 2005 and to connect the future TeLMO projection with the historical simulation.

The ASI is derived from a logistic regression analysis using past statistical data. We use the global 0.5° MODIS cropland area (Friedl et al., 2010) as the objective variable, and the Global 30 Arc-Second Elevation (GTOPO30, Verdin and Greenlee 1996), the FAOSTAT food crop yield and price (FAO 2019), and GDP per capita as the explanatory variables. GDP per capita rather than the wages of farmers is used for the reason indicated in the discussion of Eq. B-4 below. The logistic regression coefficient was derived from 23,000 data values that were randomly selected from the set of global 0.5° grids at year 2005. A comparison of the MODIS cropland areas and the calculated ASI values is shown in Figure B-1. The 23,000 randomly selected cropland area values were sorted in descending order and divided into 10 categories and the average MODIS cropland area and the average ASI-based cropland area in each category were compared. As shown in Figure B-1, the values calculated by the logistic regression effectively reproduce the distribution of the MODIS cropland area data.

In the MIROC-INTEG simulations, GTOPO30 (Verdin and Greenlee 1996) is used for the slope $\phi_i$, and the food price $p_k$ and wage $w_k$ are obtained in the International Trade Module as explained in B.1.2. PRYSBI2 results (1.0° resolution, Section 3.2), converted to a resolution of 0.5°, are used for the yield $y_j$. In TeLMO, total food cropland area is projected by using the maximum yield across the five cereal types (winter and spring wheat, maize, soybean, and rice). The reason for this formulation is explained in Section B.1.2. $y_j$ in Eq. (B-1) is calculated from the yields of the five cereals types by PRYSBI2. As discussed above, TeLMO is a model that evaluates the long-term trend in land-use change. Therefore, the crop yield and wage $w_k$ in Eq. (B-1) is the average value of 10 years (using the data from the one year to the ten years before the calculation year).

The 0.5° cell cropland area ($R_j$) is calculated by averaging the cropland probability in each of the 30" cells ($r_i$) as follows:

$$R_j = \sum_i^{J_i} \frac{r_i}{J_i} \qquad \text{(B-2)}$$

where $J_i$ is the number of $i$ cells (3600) in each 0.5° cell. The adjustment parameter $C_j$ in Eq. (B-1) is set so that the cropland area in the first year of calculation equals the data from LUH2f (Lawrence et al., 2016).

As explained above, the cropland distribution $R_j$ projected at a spatial resolution of 0.5° by the Food Cropland Down-scale Module is used in calculations in the International Trade Module (B.1.2).

**B.1.2 International Trade Module**

Our model was developed by extending one of the simplest of the basic models, the Ricardian model. The Ricardian model is a one- production-factor (productivity per capita), 2-country/2-commodity (food and non-food) model that attempts to describe the essence of free trade behavior based on the theory of comparative advantage. Because of its simple structure, the Ricardian model can be extended to a multi-country and multi-commodity model (Ejiri 2008). In the International Trade Module, we extend the Ricardian model to be a multi-country (the entire world)/2-commodity (food and non-food) general equilibrium model. In addition, we account for decreasing returns in terms of production efficiency following the approach of Ejiri (2008). That is to say, we assume that agricultural production efficiency declines with increasing cropland area (and,

conversely, that agricultural production efficiency increases as cropland area decreases). For this reason, industrial specialization, which has been pointed out as a problem of the Ricardian model, is unlikely to occur.

In order to construct a multi-country/2-commodity model, the subscript $k$ was used to indicate country/region (the same 17 countries/regions defined in AIM/CGE), and subscripts 1 and 2 were added to indicate agricultural and non-agricultural sectors, respectively. The prices and wages in Eq. (B-1) are those in the agricultural sector, which are represented by $p_{1,k}$ and $w_{1,k}$, respectively.

First, wages in the agricultural sector, $w_{1,k}$, are defined by using labor input and gross domestic production (GDP). In the International Trade Module, economic variables (e.g., food prices, wages, labor, and GDP) are described as the relative ratio to the base year (2005), the first year of calculation. Here, we assume that the total labor population ratio (relative to the base year) equals the total population ratio (relative to the base year).

$$l_{1,k} + l_{2,k} = L_k \qquad \text{(B-3)}$$

where $l_{1,k}$, and $l_{2,k}$ are the labor input of the agricultural and non-agricultural sectors, respectively, and $L_k$ is the total labor population (Murakami and Yamagata 2019). GDP can then be described as total domestic income:

$$GDP_k = w_{1,k} \cdot l_1 + w_{2,k} \cdot l_2$$

where the value calculated by AIM/CGE is used for $GDP_k$ (units: USD). If we assume that the wage (ratio relative to the base year) for the non-agricultural sector is the same as that of the agricultural sector, the agricultural worker wage $w_{1,k}$ is calculated as:

$$w_{1,k} = \frac{GDP_k}{l_{1,k} + l_{2,k}} = \frac{GDP_k}{L_k} \qquad \text{(B-4)}$$

In other words, it is assumed that the change in agricultural worker wage (relative to the base year) is equal to the change in per capita GDP. It is known that the employment rate have changed by a small percentage in the past. However, it is difficult to project the future changes in the employment rate, and thus the employment rate is assumed to be constant in the standard CGE models (e.g. Fujimori et al. 2012). Similarly, it is not easy to confirm the historical changes in wages for each country, nor to estimate their future change; thus, similar to that for employment rate, the future changes in wages are usually kept constant in the CGE models (e.g., Fujimori et al. 2012). It should be noted that a small increase in employment rate (compared to the base year) can slightly decrease the wages as indicated in Eq. (B-4), possibly leading to an increase in cropland area (Eq. B-1).

Next, the price for agricultural sector $p_{1,k}$ is calculated using the multi-country/2-commodity general equilibrium model. The prices for agricultural and non-agricultural sectors are calculated using Eqs. (B-5) and (B-6), respectively:

$$p_{1,k} = w_{1,k} \frac{l_{1,k}}{x_{1,k}} \qquad \text{(B-5)}$$

$$p_{2,k} = w_{2,k} \frac{l_{2,k}}{x_{2,k}} \qquad \text{(B-6)}$$

where $x_{1,k}$ and $x_{2,k}$ are the production index in the agricultural and non-agricultural sectors, respectively. Here, the production index in the agricultural sector in region $k$ ($x_{1,k}$,) can be calculated as the sum of the products of $0.5°$ crop yield $y_j$ and cropland area $R_j$ using Eq. (7):

$$x_{1,k} = \sum_{j}^{K_j} y_j R_j \qquad (B\text{-}7)$$

where $K_j$ indicates the number of $0.5°$ cells within the country/region $k$ (3600). As described above, the cropland distribution $R_j$ generated by the Food Cropland Down-scale Module (B.1.1) is used in Eq. (B-7). The domestic price $p$ in Eqs. (B-6) and (B-7) is expressed in terms of the local currency unit (LCU). This is converted to the international price $P$ (USD) using the exchange rate $\pi$ (LCU/USD) in Eqs. (B-8) and (B-9):

$$p_{1,k} = \pi_k \cdot P_{1,k} \qquad (B\text{-}8)$$

$$p_{2,k} = \pi_k \cdot P_{2,k} \qquad (B\text{-}9)$$

The price $p$ and production index $x$ can then be connected using a relational equation for the trade budget as follows. Imposing the condition that the international budget for any country is zero results in Eq. (B-10) for the international balance of payments:

$$p_{1,k} \cdot \left(x_{1,k} - X_{1,k}\right) + p_{2,k} \cdot \left(x_{2,k} - X_{2,k}\right) = 0 \qquad (B\text{-}10)$$

where $X_{1,k}$, and $X_{2,k}$ are the demands for each good in each region. As described previously, the output generated by AIM/CGE based on the socio-economic scenario is used for food demand $X_{1,k}$. In this study, livestock feed demand is not included in $X_{1,k}$. The international balance of payments as shown in Eq. (B-10) consists of the current, capital and financial accounts. The imbalance in the international budget corresponds to foreign exchange reserve. The foreign exchange reserve changes over periods longer than 10 years, but it is not possible to predict its future variation, and thus it is not considered in the standard CGE models (e.g., Ejiri 2008). In the real world, if foreign exchange reserve increases, amount of import goods tends to be decreased because money is not used for them. Consequently, in food importing countries, food production tends to be increased, possibly leading to an increase in cropland area.

In addition, the price $p$ and product index $x$ can be related through Eq. (B-11) by expressing economic growth in terms of GDP:

$$GDP_k = P_{1,k} \cdot x_{1,k} + P_{2,k} \cdot x_{2,k} \qquad (B\text{-}11)$$

In Eq. (B-3) and Eqs. (B-5) to (B-11) above, the eight unknown values are $p_{1,k}$, $p_{2,k}$, $x_{1,k}$, $x_{2,k}$, $l_{1,k}$, $l_{2,k}$, $\pi_k$, and $X_{2,k}$. Of these, because the reference for the international price $P$ is the United States (region index $k = 1$), $P_{1,1}$ and $P_{2,1}$ (along with $p_{1,1}, p_{2,1}$) cannot be set. For this reason, the condition is imposed that total global net exports and imports equal to zero:

$$\sum_{k=1}^{K_{all}} \left(x_{1,k} - X_{1,k}\right) = 0 \qquad (B\text{-}12)$$

$$\sum_{k=1}^{K_{all}} (x_{2,k} - X_{2,k}) = 0 \tag{B-13}$$

As explained above, TeLMO uses 10-year averages as input to the model to represent long-term trends inland-use change (B.1.1). We assumed that the global total production is equal to consumption, i.e., the total global net exports and imports equal to zero. In reality, there are certainly stock changes in various goods but it would not be counterfactual to assume that they are net zero at longer time scale. The unknown values for $p_{1,k}$, $p_{2,k}$, $x_{1,k}$, $x_{2,k}$, $l_{1,k}$, $l_{2,k}$, $\pi_k$, and $X_{2,k}$ are calculated by simultaneously solving eight equations, Eq. (B-3) and Eqs. (B-5) to (B-11), for all 17 regions ($k = 1 - 17$) subject to the conditions imposed by Eqs. (B-12) and (B-13). The $p_{1,k}$, and $w_{1,k}$ values obtained from Eq. (B-4) are entered into Eq. (B-1). Finally, the share of cropland for each 0.5° cell $R_j$ can then be calculated using Eq. (B-2).

As explained in Section B.1.1, TeLMO uses the maximum yield of five cereals types to project the total cropland area. Alternatively, it is possible to increase the number of agricultural sectors in Eqs. (B-3) to (B-12), solve the prices for each crops, and allocate the cropland area according to the ASIs for each crop. Although we attempted this formulation in the course of our development of TeLMO, it was found that the results were similar to those obtained from the current formulation. On the other hand, the solution of general equilibrium models did not converge in some cases because the number of sectors increases in the equations. For this reason, we decided to adopt the current formulation, while recognizing that calculating cropland areas for each crop is an important future work.

## B.2 Bio-energy Cropland Model

The Bio-energy Cropland Model uses 30″ cells that are not assigned to urban use or food cropland use. Whereas adjustment parameter $C_j$ in the Food Cropland Model (Eq. B-1) could be set using observed cropland area for the first year of the TeLMO calculation (the base year 2005), there is no corresponding adjustment parameter in the case of bio-energy cropland because sufficient cropland devoted to biofuel crops did not exist in the base year. Accordingly, the Bio-energy Cropland Model allocates bio-energy cropland around the globe so that the global total biofuel crop production equals the global total biofuel crop demand obtained by AIM/CGE. The Bio-energy Cropland Model uses the same formularization to that in the Food Cropland Down-scale Module (B.1.1) to evaluate the probability of bio-energy cropland in 30″ cells using the following equation:

$$r_{bio,i} = \frac{C_{bio}}{1 + \exp\left(1.228 + 0.237\phi_i - 0.206 p_{bio,k} y_{bio,j}/w_{1,k}\right)} \tag{B-14}$$

where $\phi_i$ is the slope in 30″ cell $i$, $p_{bio,k}$ is the biofuel crop price in region $k$, $y_{bio,j}$ is the yield [t/ha] of biofuel crops in 0.5° cells, and $w_{1,k}$ is the agricultural sector wage in region $k$. For the biofuel crop price $p_{bio,k}$, the values generated by AIM/CGE are used. For biofuel crop yield $y_{bio,j}$, the yield for miscanthus or switchgrass, whichever is greater in a given cell, is calculated for the entire globe by using the biofuel crop model developed in Kato and Yamagata (2014). The

biofuel crop model in Kato and Yamagata (2014) considers the future changes in climate based on the RCP scenarios. In this study, we also consider the future changes in fertilizer input based on the SSPs adopted in Mori et al. (2018). Because of the uncertainty in future fertilizer application for crop management, we set the high end of the N fertilizer input threshold according to Tilman et al. (2011). The nitrogen fertilizer application was set to increase from the current level according to the increasing rate of GDP in the SSP2 scenario up to 160 kg N ha$^{-1}$ yr$^{-1}$ if the fertilizer input at the country level was below 160 kg N ha$^{-1}$ yr$^{-1}$ in the 2000s. Also, the phosphorus fertilizer input in each country was set to follow the same annual increase rate as the nitrogen fertilizer application.

Our use of the same formularization for the Food Cropland Model and the Bio-energy Cropland Model is based on the assumption that the factors determining both cropland areas are similar.

The adjustment parameter $C_{bio}$ is set so that the global total biofuel crop production volume (product of yield and cropland area) equals the global total biofuel crop demand calculated by AIM/CGE:

$$\sum_k^{K_{all}} X_{bio,k} = \sum_j^{J_{all}} y_{bio,j} R_{bio,j} \qquad \text{(B-15)}$$

where $X_{bio,k}$ is the biofuel crop demand for region $k$ calculated by AIM/CGE, $K_{all}$ and $J_{all}$ are the total number of regions (17) and the total number of 0.5° cells (259,200), respectively. $R_{bio,j}$ is the average percentage of bio-energy cropland for all 30″ cells in a given 0.5° cell, where the individual 30″ cell percentages are determined by Eq. (B-14).

If bio-energy cropland were allocated based on the principle described above, a massive development of bio-energy cropland would occur in regions with high ecosystem production such as the Amazon. For this reason, the model accounts for protected areas that cannot be allocated as bio-energy cropland as shown in Figure B-2. Two sources were used for protected areas (Wu et al., 2019): the World Database for Protected Areas (WDPA) (IUCN and UNEP-WCMC 2018) and the World Database of Key Biodiversity Areas (KBA) (BirdLife International 2017). As of 2018, the WDPA covered an area of 33.6 million km$^2$, and the KBA covered an area of 19.9 million km$^2$. In this study, we did not consider the protected area for the calculation of the food cropland and pasture, under the assumption that food has a higher priority than ecosystem protection.

**B.3 Pastureland Model**

Whereas the Food Cropland Model uses statistical relationships between cropland area, yield, and economic variables, because reliable statistical data do not exist for pastureland, a simpler approach is taken to estimate pastureland. The probability of pastureland in each 30″ cell is determined based on net primary production (NPP) and slope, given by:

$$r_{past,i} = \frac{C_{past,j} \times NPP_j}{\left(1 + \phi/20\right)} \qquad \text{(B-16)}$$

The denominator in Eq. (B-16) reflects the fact that the use of land as pasture decreases with the angle of inclination, as is shown in the LUH2f data (Lawrence et al., 2016). The results of an off-line simulation by VISIT (Ito and Inatomi 2012) assuming the entire world to be grassland are used here for $NPP_j$. The boundary condition of the VISIT off-line simulations is fixed at year 2005. $C_{past,j}$ is the adjustment parameter for 0.5° cells. The value of $C_{past,j}$ changes from year to year. The

adjustment parameter for the base year, $C_{past,j}(t = 0)$ is set so that the pastureland distribution equals that of LUH2f (Lawrence et al., 2016) for the base year (2005). Adjustment parameters for years other than the base year, $C_{past,j}(t)$, are set by applying a proportionality factor $\alpha(t)$ to the base-year parameter:

$$C_{past,j}(t) = \alpha(t) \times C_{past,j}(t = 0) \qquad \text{(B-17)}$$

where $\alpha(t)$ is set so that regional total pastureland area equals the regional total pastureland demand calculated by AIM/CGE. In other words, $\alpha(t)$ is set so that the condition

$$S_{past,k}(t) = \sum_{j}^{J_k} R_{past,j}(t) \qquad \text{(B-18)}$$

is met, where $S_{past,k}(t)$ is the pastureland demand calculated by AIM/CGE for region $k$, $R_{past,j}(t)$ is the average of percentage of pastureland for all 30″ cells (from Eq. (B-16)) in a given 0.5° cell, and $J_k$ is the total number of 0.5° cells in each region $k$.

**B.4 Managed Forest Model**

In the Managed Forest Model, satellite data are used to determine forest area; the share of forest area where timber harvesting occurs is allocated as managed forest in the manner described below. The distribution of managed forests in 0.5° cells, $R_{mfr,j}(t)$, is formularized in terms of the area of managed forests in the base year and the population density:

$$R_{mfr,j} = A_{fr,j} \times \frac{\rho_{j*}}{C_{manfr,k} + \rho_{j*}} \qquad \text{(B-19)}$$

where $A_{fr,j}$ is the area of managed forest in 0.5° cells in the base year (2005), $\rho_{j*}$ is the mean population density in the 5×5 grid (2.5° cell) of cells centred on the 0.5° cell in question. Larger 2.5° cells were used instead of 0.5° cells based on the

assumption that harvested timber is transported within an approximately 100-km radius and that the amount of harvested timber is determined by the population density in each 2.5° cell. The 100-km radius is estimated from the distance where the transportation cost of timber (~ 1 \$/km/tons) is balanced with the price of timber (~ 100 \$/tons). Here, the transportation cost and price of timber are estimated using the FAOSTAT data (FAO 2019). Moderate Resolution Imaging Spectroradiometer (MODIS) satellite data (Friedl et al., 2010) are used for the base-year forest area (2005), and data from Murakami and

Yamagata (2019) are used for the population density, $\rho_{j*}$. $C_{mfr,k}$ is an adjustment parameter that is set for each of the 17 regions ($k$) so that the managed forest area conforms to the round-wood demand $X_{mfr,k}$ [kg/yr] calculated by AIM/CGE. We use the region-level adjustment factors for managed forest ($C_{mfr,k}$) because the grid-level reference data is not available.

In other words, $C_{mfr,k}$ is set so that the total regional amount of harvested timber equals the regional total round-wood demand:

$$X_{mfr,k} = \sum_{j}^{J_k} R_{mfr,j} \times \frac{B_j}{L_j} \qquad \text{(B-20)}$$

where $B_j$ is the distribution of forest biomass [kg/m$^2$] in 0.5° cells, calculated by VISIT (Ito and Inatomi 2012) off-line simulations assuming the entire world to be forest with the fixed boundary conditions (2005). $J_k$ is the total number of 0.5° cells in each region $k$. $L_j$ is the harvesting period [yr], which is estimated as follows, based on the $NPP_j$ for 0.5° cells obtained from VISIT (Ito and Inatomi 2012):

$$L_j = \begin{cases} \infty & NPP_j < 4 \\ 500/NPP_j & 4 \leq NPP_j \leq 25 \\ 20 & 25 < NPP_j \end{cases} \qquad \text{(B-21)}$$

$L_j$ reflects the fact that the harvesting period decreases with increases in net primary production, as is shown in the LUH2v data (Lawrence et al., 2016). The amount of forest harvested in a given year can also be calculated as $R_{mfr,j} \times \frac{B_j}{L_j}$ [kg/yr] based on the distribution of managed forests $R_{mfr,j}$, forest biomass $B_j$, and the felling period $L_j$ for 0.5° cells.

**B.5 Formulation of Transition Matrix Model**

Evaluating the impact of land-use change on terrestrial ecosystems requires not only the spatial distribution of land use but also information on the land-use transition. For example, in areas where shifting cultivation is practiced, even though the overall cropland area within a cell does not change, a particular area may be cleared as cropland while another area is abandoned. In such cases, there is a transition from cropland to secondary land, which impacts the above-ground biomass and carbon budget. Thus, matrix information regarding the transition from one land use to another land use is essential.

For the landcover types used in the transition matrix, we use the five classes (urban, cropland, pasture, secondary/primary land) used in the VISIT terrestrial ecosystem model (Ito and Inatomi 2012). TeLMO forecasts eight landcover types, including the previously described urban, cropland (food and bio-energy), pasture, managed forest, and unmanaged forest classes as well as "grassland" (obtained from MODIS satellite data, Friedl et al., 2010) and "other" landcover types that are not used by humans (for example, glaciers, lakes and marshes, as defined by MODIS satellite data, Friedl et al., 2010). The correspondence between the landcover types used in TeLMO and those used in the land-use transition matrix is presented in Table B-1.

The primary/secondary land classes in the land-use Transition Matrix Model are defined as land that has never been used by humans or land that has been used at least once by humans, respectively. Here, unmanaged forest and grassland are classified as primary or secondary land based on data from LUH2f supplied by LUH2v (Lawrence et al., 2016). Unmanaged forest or grassland areas that are classified as secondary land in the base year (2005) remain classified as secondary land in

subsequent years. In the case in which unmanaged forest or grassland areas are classified as primary land in the base year, if the area is converted to cropland or pasture and then later returned to being unmanaged forest or grassland, it is classified as secondary land. In TeLMO, land classified as "other" is considered the land that cannot be used by humans and is therefore not included in the land-use transition matrices.

The method used to create the land-use transition matrices is shown in Figure B-3. As explained above, TeLMO assumes that land is used in order of highest to lowest value added per unit area (i.e., urban, food cropland, bio-energy cropland, pastureland, managed the forest, and unmanaged forest). Aligning these land-use classes with corresponding classes in the transition matrix (Table B-1), the preferential order of the latter becomes urban, cropland (food + bioenergy), pasture, secondary land, primary land. To calculate land-use transition matrices, the percent areas of the different landcover types in

each 0.5° cell in a given year are first sorted in order of preference ("Pre" in Figure B-3). In Figure B-3, the length of each colored bar represents the percent area of a given landcover type. The sum of the percent areas for all land-use classes is 100%. Next, the percent areas of different landcover types in each 0.5° cell in the following year are again sorted in order of preference ("Post" in Figure B-3).

As shown in Figure B-3, the percent areas of transitioned land defined in transition matrices can be calculated by

comparing the percent areas for each landcover type in a given year and the next year. For example, the area indicated in column "a" in Figure B-3 corresponds to the percent area of land that transitioned from pasture to cropland. Similarly, the area indicated in column "b" in Figure B-3 corresponds to the percent area of land that transitioned from secondary land to pasture. In this manner, it is possible to calculate the transition between landcover types by assuming a preferential order to land use.

Shifting cultivation is taken into account when making the land-use transition matrices. We assume that the share of cultivated land does not change over time on the larger (i.e., 0.5° cell) scale. Data from Butler (1980) are used for the global allocation of shifting cultivation on this larger scale. Furthermore, in regions where shifting cultivation is practiced, we assume that cropland is used sequentially with a fixed rotation (Butler 1980). Under this assumption, in areas where shifting cultivation is practiced, 1/15 of the cropland area is newly cultivated, and 1/15 of the cropland area is abandoned each year.

Thus, 1/15 of the cropland area is transitioned from secondary land to cropland, and 1/15 of the cropland area is transitioned from cropland to secondary land. These transitions are added to the transition matrices for areas where shifting cultivation is practiced.

**Code and data availability**

The MIROC-INTEG source code for this study is available to those who conduct collaborative research with the model users

under license from the copyright holders. For further information on how to obtain the code, please contact the corresponding author. The data from the model simulations and observations used in the analyses are available from the corresponding author upon request.

**Acknowledgements**

This research is supported by the "Integrated Research Program for Advancing Climate Models (TOUGOU Program)" sponsored by the Ministry of Education, Culture, Sports, Science, and Technology (MEXT), Japan. It was carried out as part of the Integrated Climate Assessment–Risks, Uncertainties, and Society (ICA-RUS) project funded by the Environment Research and Technology Development Fund (S-10, JPMEERF12S11000) of the Ministry of the Environment of Japan. Model simulations were performed on the SGI UV20 at the National Institute for Environmental Studies. We gratefully acknowledge the helpful discussions with Kaoru Tachiiri, Tomohiro Hajima, Takashi Arakawa, Junichi Tsutsui, and Michio Kawamiya. The authors are much indebted to Mr. Keita Matsumoto, Mr. Kuniyasu Hamada, Mr. Kenryou Kataumi, Mr. Eiichi Hirohashi, Mr. Futoshi Takeuchi, Mr. Nobuaki Morita, and Mr. Kenji Yoshimura at NEC Corporation for their support in model development.

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

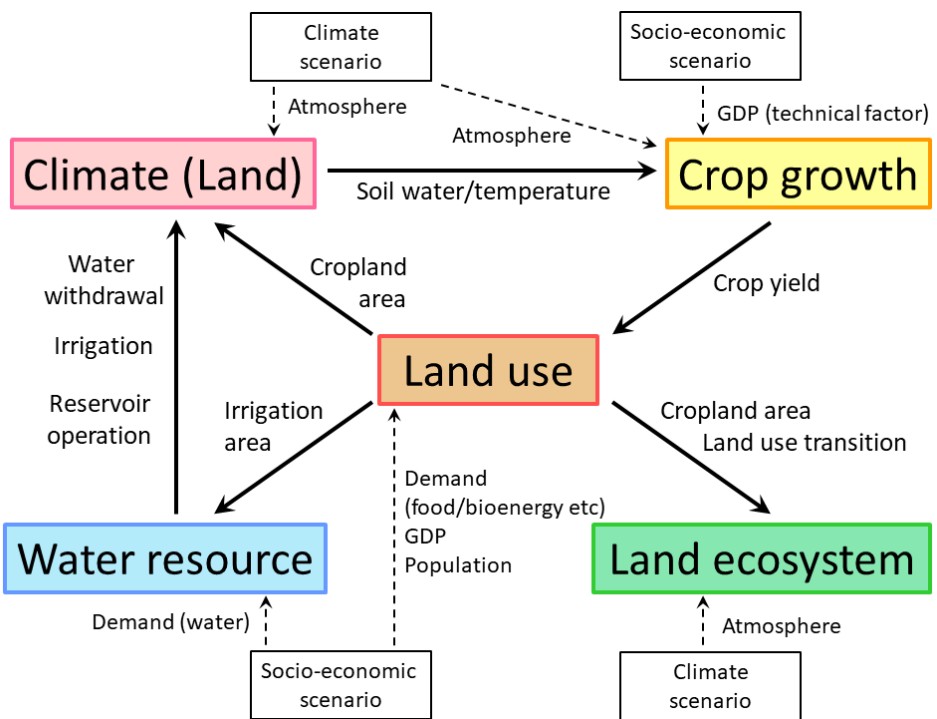

**Figure 1: Relationship among variables in MIROC-INTEG-LAND. Components of the integrated model (sub-models) are shown as colored boxes. Climate (land surface) and water resource components are HiGWMAT (Pokhrel et al. 2012a), which is based on the land surface model MATSIRO (Nitta et al. 2014) in a global climate model MIROC (Watanabe et al. 2010). Land ecosystem and crop growth components are VISIT (Ito and Inatomi 2012) and PRYSBI2 (Sakurai et al. 2014), respectively. The land use model, TeLMO, is developed in this study. Inputs into the model are shown as boxes of climate and socio-economic scenarios. Solid arrows between the boxes indicate the exchange of variables between the sub-models. Dashed arrows indicate the input variables of the sub-models.**

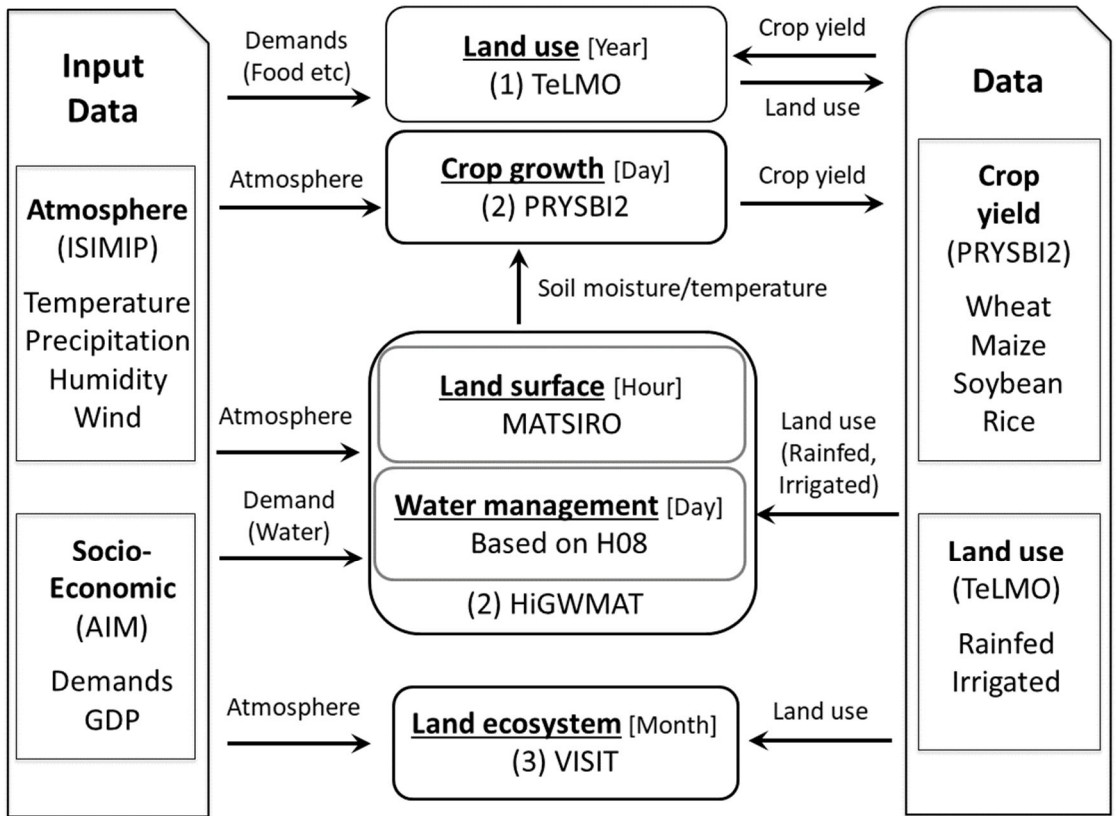

**Figure 2: The numerical simulation procedure in MIROC-INTEG-LAND. The order of the numerical integration is (1) TeLMO, (2) HiGWMAT + PRYSIB2, (3) VISIT as described in Section 4. Boxes indicate the sub-models and data. For the sub-models, the name and time-step of the models are indicated in the boxes. In the "data" box, the name of the variable saved as a file is indicated. In the "input data" box, information regarding the input data is indicated.**

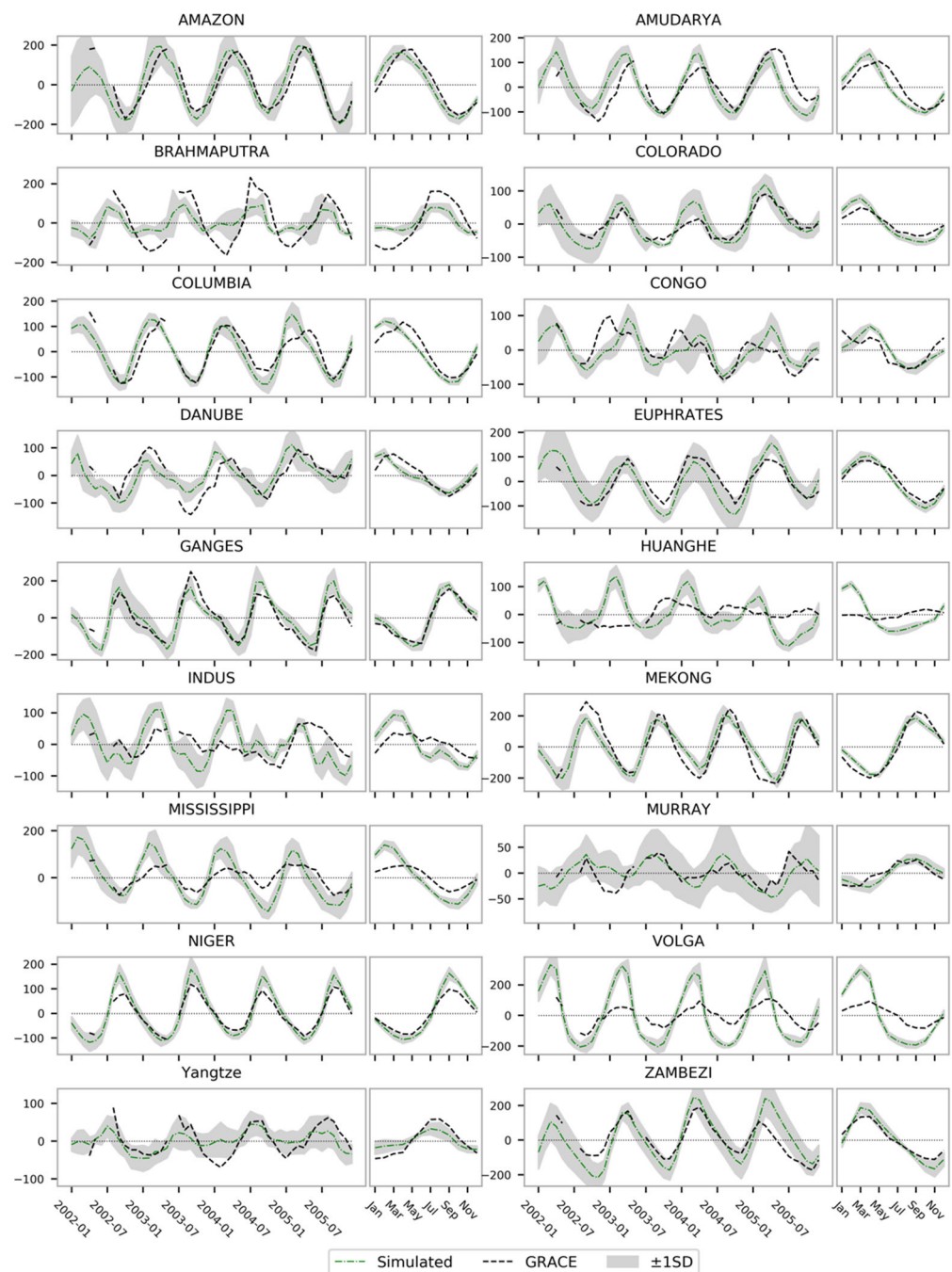

**Figure 3: Comparison of historical terrestrial water storage (TWS) simulated by MIROC-INTEG-LAND with GRACE satellite data. For each river basin, the panel on the right shows the seasonal cycle. The GRACE data shown are the mean of the mass concentration products from two processing centers: CSR and JPL. Simulated results are the average of five climate model simulations. Grey shading indicates the uncertainty range shown by one standard deviation from the mean.**

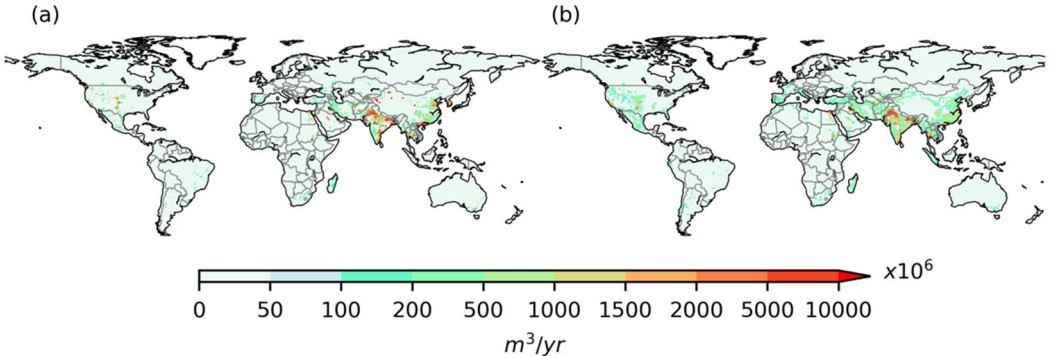

**Figure 4: Comparison of irrigation demands simulated by MIROC-INTEG-LAND (a) with the results from offline simulations using HiGW-MAT (b) forced by observed climate forcing data (Pokhrel et al., 2015) for 1°×1° grids shown as the mean for 1998-2002 period.**

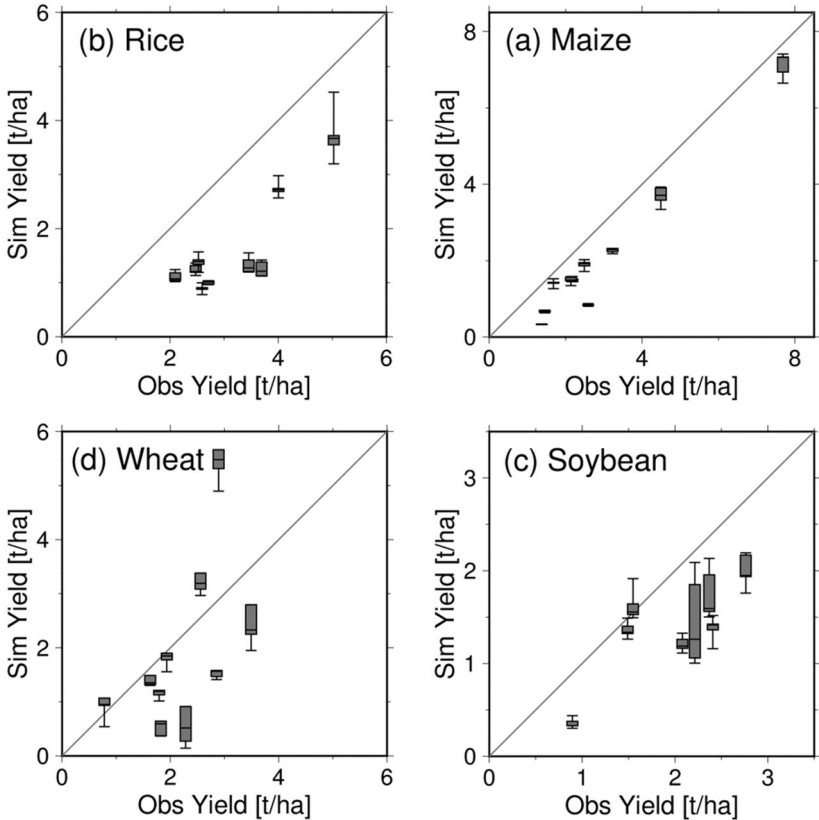

**Figure 5: Comparison of model estimation with reference data on average yield during the period 1981-2005 for the top ten countries producing each crop. The Box plot shows the median and range of model results estimated from the five GCM outcomes. The main production countries were identified according to the country-based harvested area for each crop.**

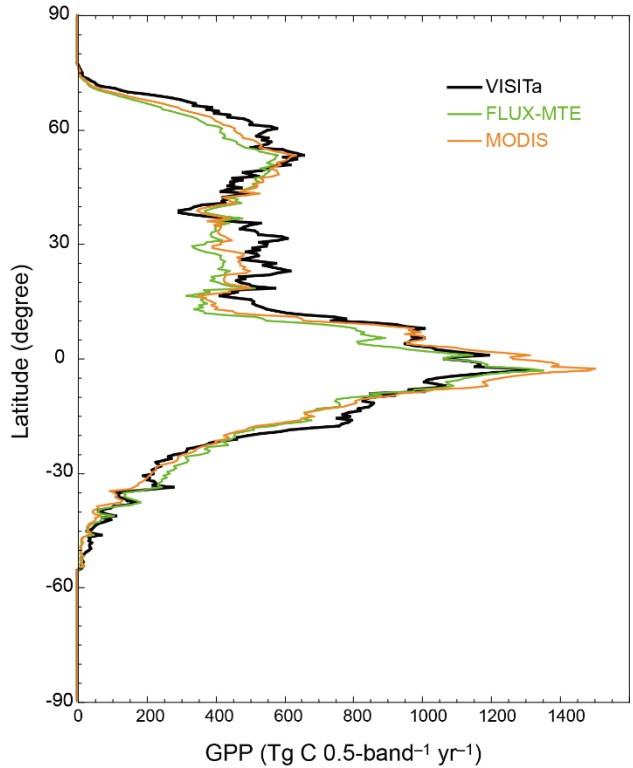

**Figure 6: Comparison of latitudinal distribution of gross primary production in 2000–2010 with up-scaled flux measurements (Model-Tree Ensemble (MTE); Beer et al., 2010) and satellite observation (MODIS; Zhao et al., 2005).**

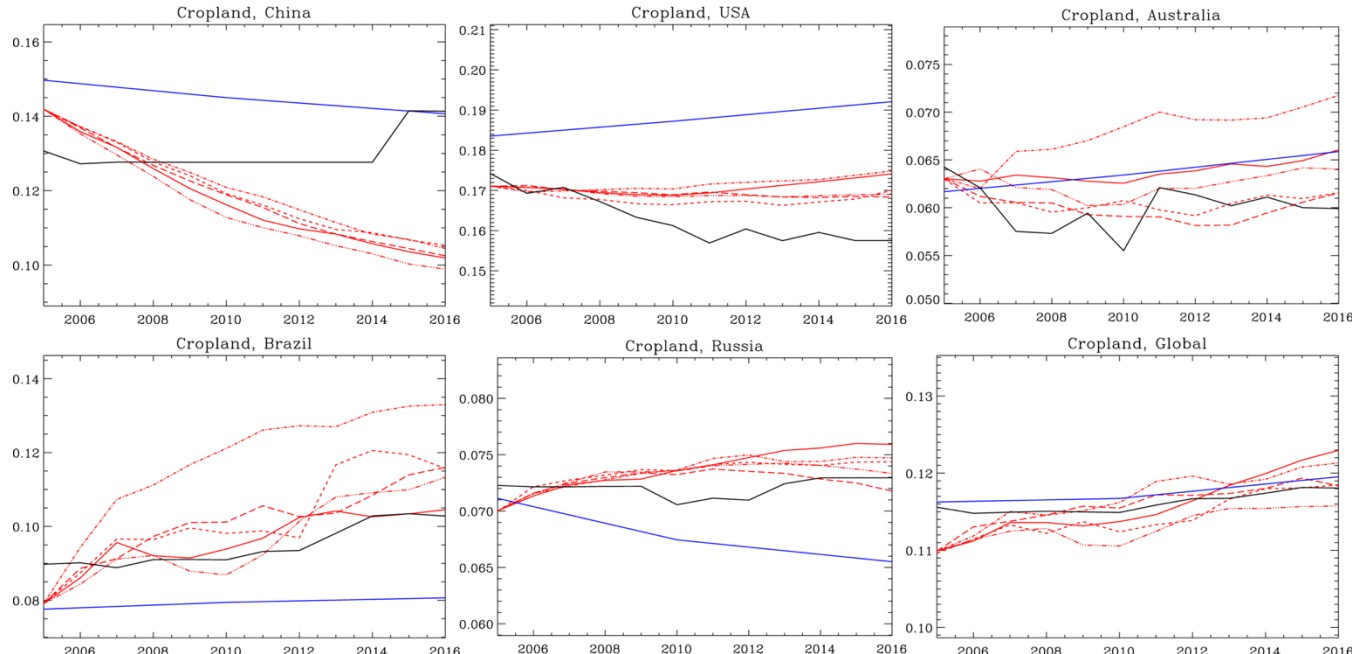

**Figure 7: Comparison of historical cropland area simulated by MIROC-INTEG (red), AIM/CGE (blue), and FAOSTAT (black), using the ratio of cropland area to total area. For MIROC-INTEG simulations, the cropland area results for the five different climate forcings are shown.**

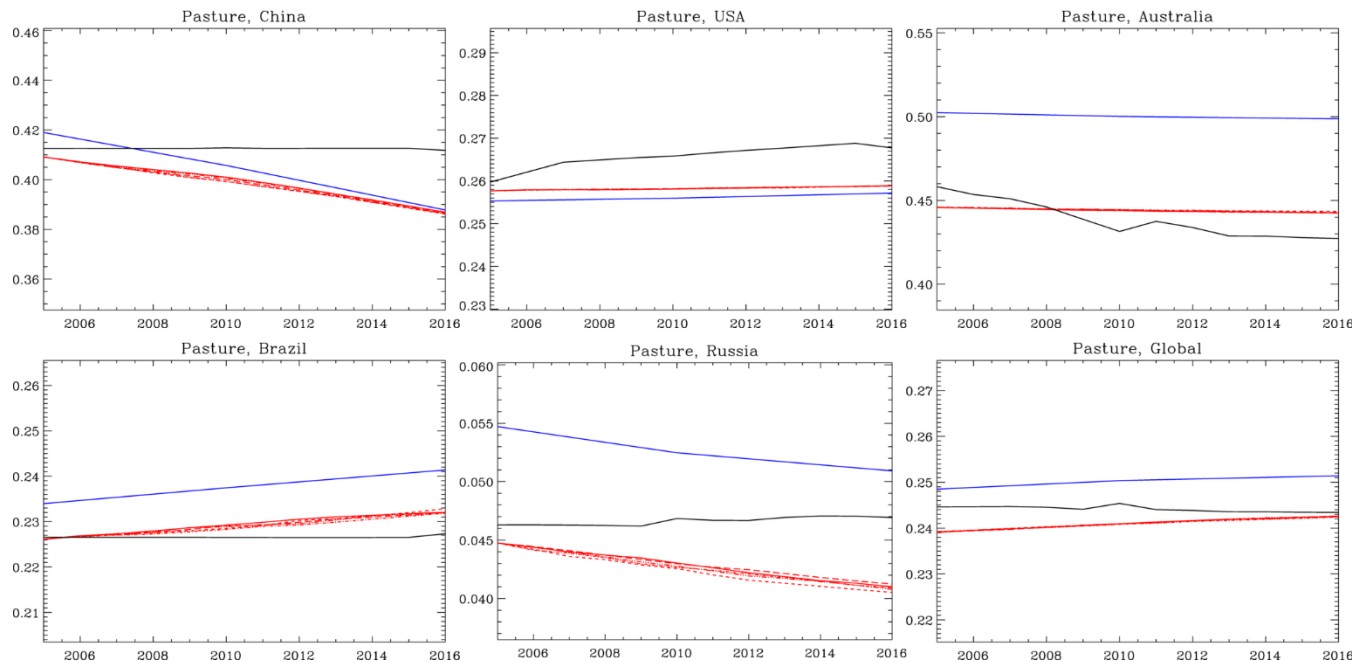

**Figure 8: Same as Figure 7, but for the comparison of historical pasture area simulated by MIROC-INTEG (red), AIM/CGE (blue), and LUH (black), using the ratio of pasture area to total area.**

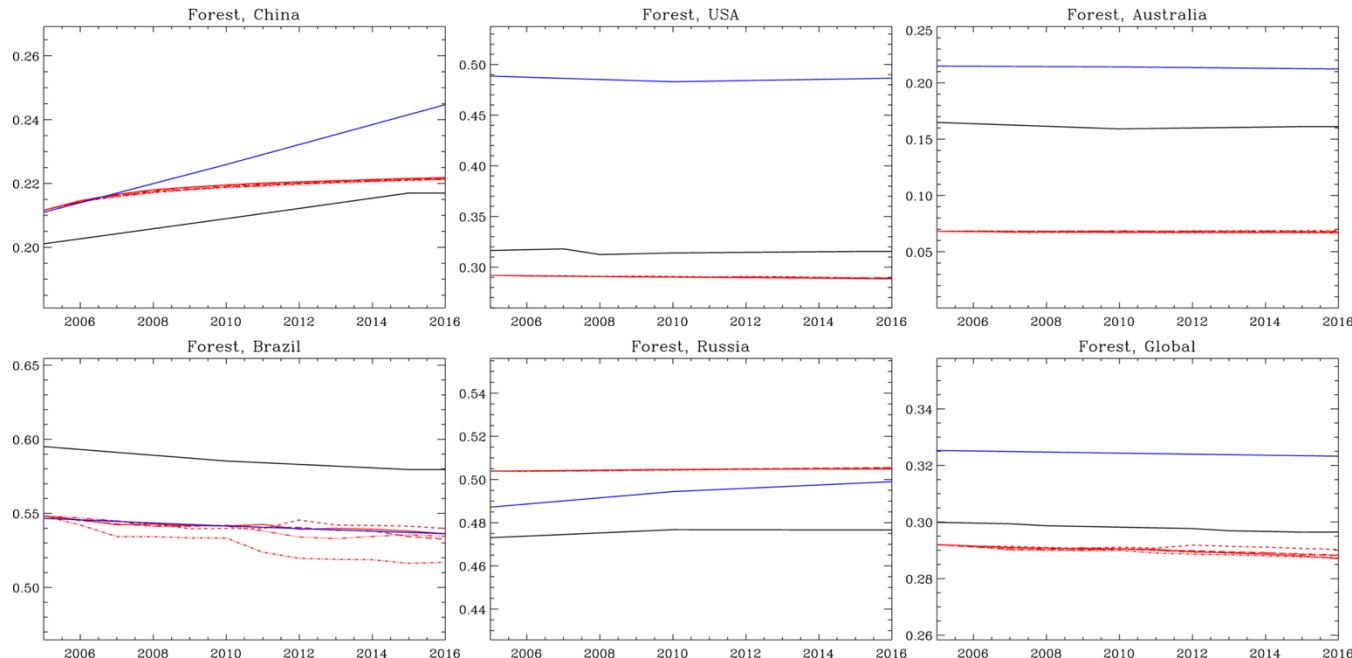

**Figure 9: Same as Figure 7, but for the historical forest area simulated by MIROC-INTEG (red), AIM/CGE (blue), and FAO (black), using the ratio of forest area to total area.**

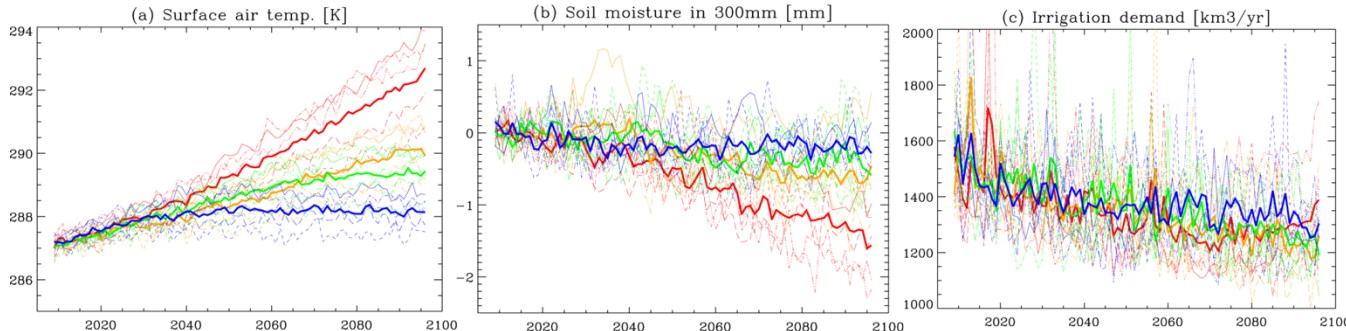

**Figure 10: Time series of changes in the climate system based on the forcings of the five climate models. Results shown are for (a) surface air temperature [K], (b) soil moisture in the top 300 mm of the soil column [mm], shown as an anomaly from first 20-year average, (c) Irrigation water supply [km³/yr]. Thin curves indicate the global average of results for each of the five climate model forcings. Thick curves show the overall average of results based on the five forcings. The colors indicate RCP2.6 (blue), RCP4.5 (green), RCP6.0 (orange), and RCP8.5 (red).**

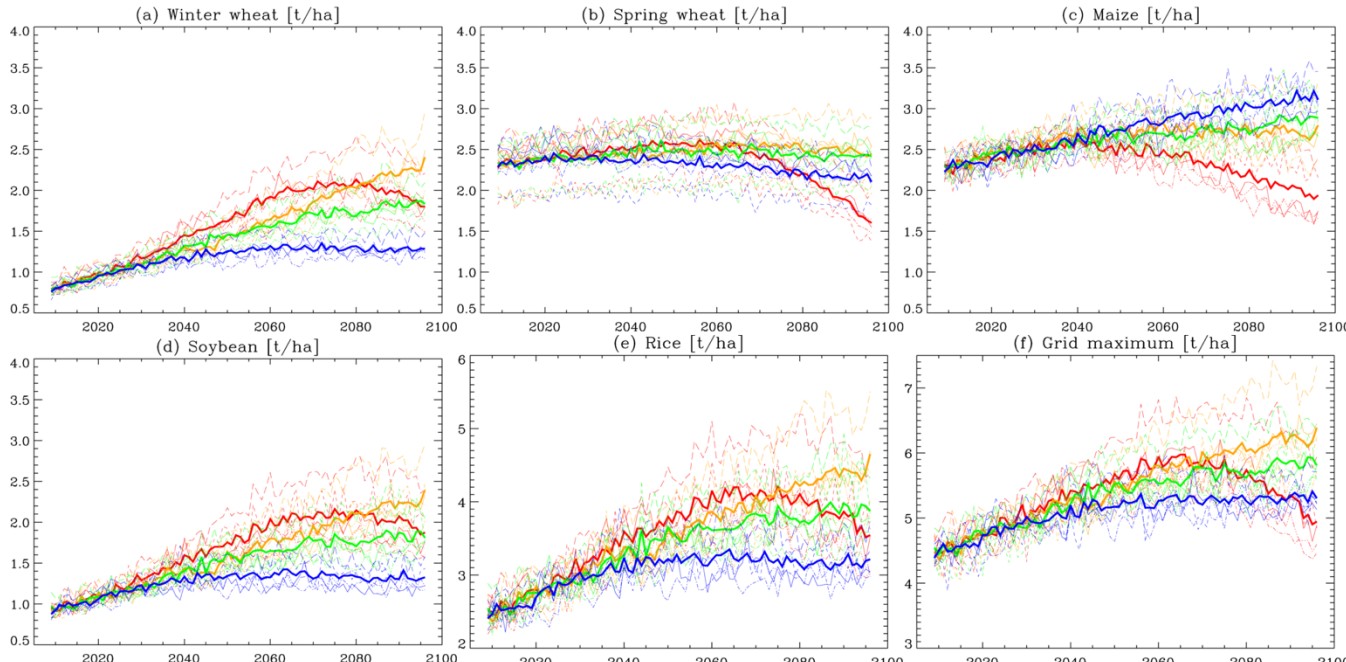

**Figure 11: Time series of changes in crop yield [unit: tons/ha] based on the forcings of the five climate models. Results shown are for (a) winter wheat, (b) spring wheat, (c) maize, (d) soybean, (e) rice, and (f) grid maximum value for the five crop types. Thin curves indicate the global average of results for each of the five climate model forcings. Thick curves show the overall average of results based on the five forcings. The colors indicate RCP2.6 (blue), RCP4.5 (green), RCP6.0 (orange), and RCP8.5 (red).**

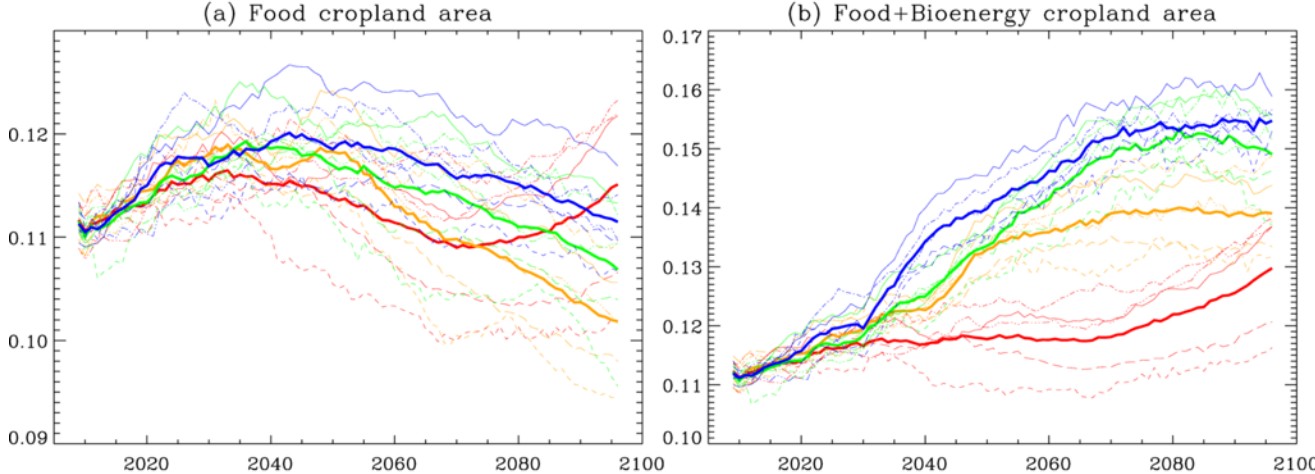

**Figure 12: Time series of changes in cropland area based on the forcings of the five climate models. The vertical axis is the cropland area as a fraction of total land area. The results are for (a) food cropland area, and (b) food + bioenergy cropland area. Thin curves indicate the global average of results for each of the five climate model forcings. Thick curves show the overall average of results based on the five forcings. The colors indicate RCP2.6 (blue), RCP4.5 (green), RCP6.0 (orange), and RCP8.5 (red).**

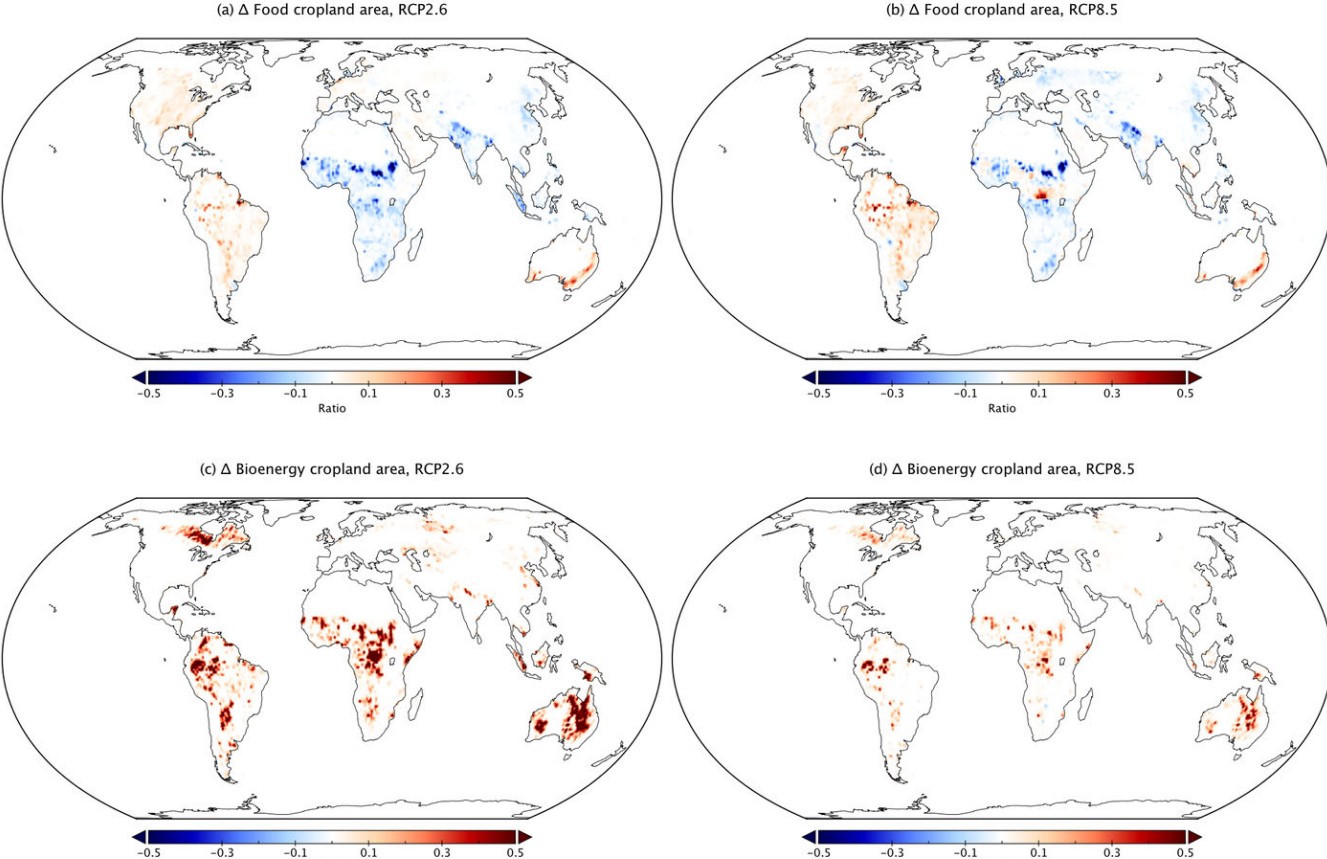

**Figure 13: Spatial distribution of land-use change [units: a ratio of the grid box area]. The results are for (a, b) food cropland area, and (c, d) bioenergy cropland area. Average of the five climate projection-based simulations under (a, c) RCP2.6 and (b, d) RCP8.5 scenarios in the 2090s.**

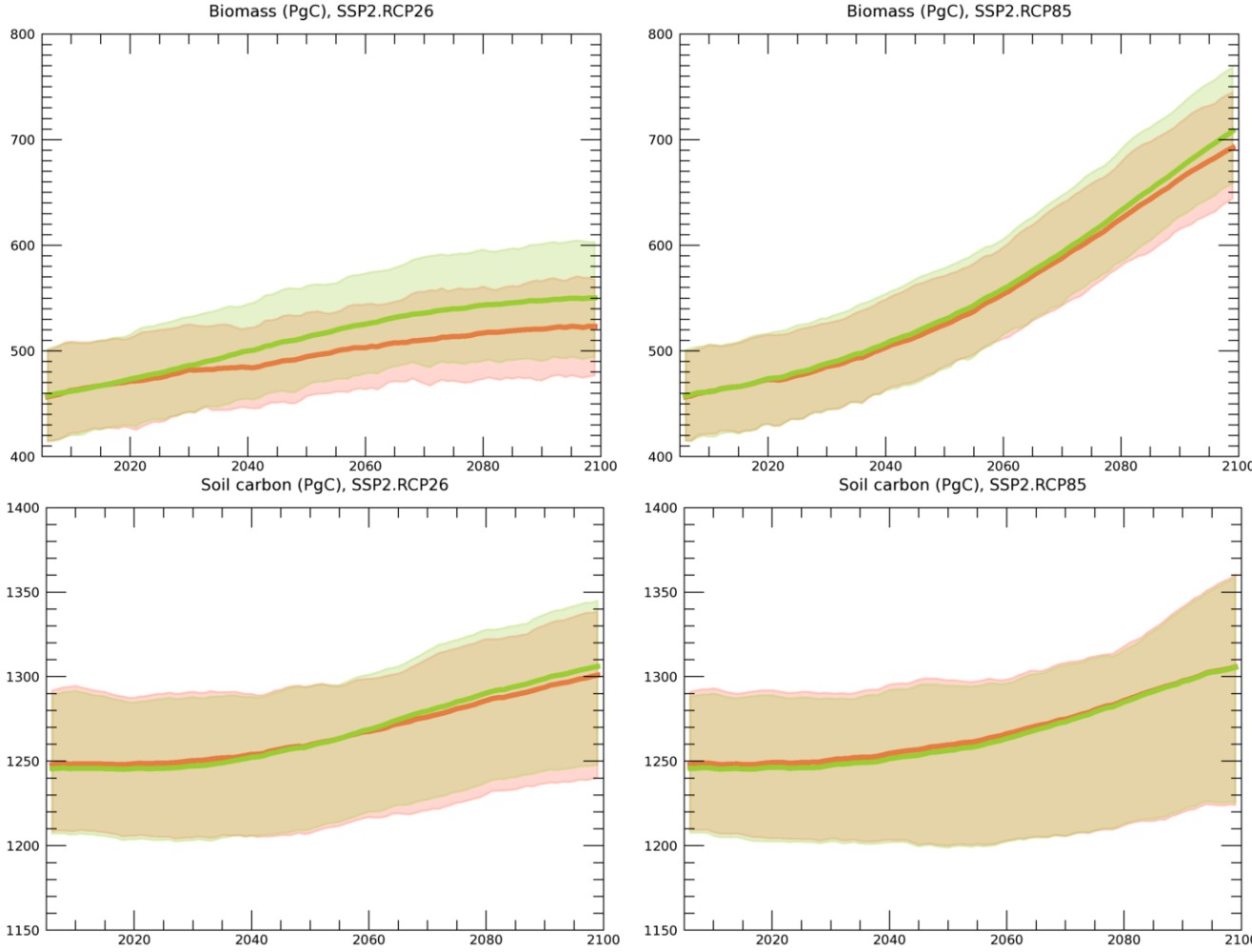

**Figure 14: Temporal change in global carbon stock in (top) vegetation biomass and (bottom) soil organic carbon, (red) with and (green) without land-use change, under (left) RCP2.6 and (right) RCP8.5 scenarios. Thick lines show the median and light zones show the maximum–minimum range of the five climate projection-based simulations.**

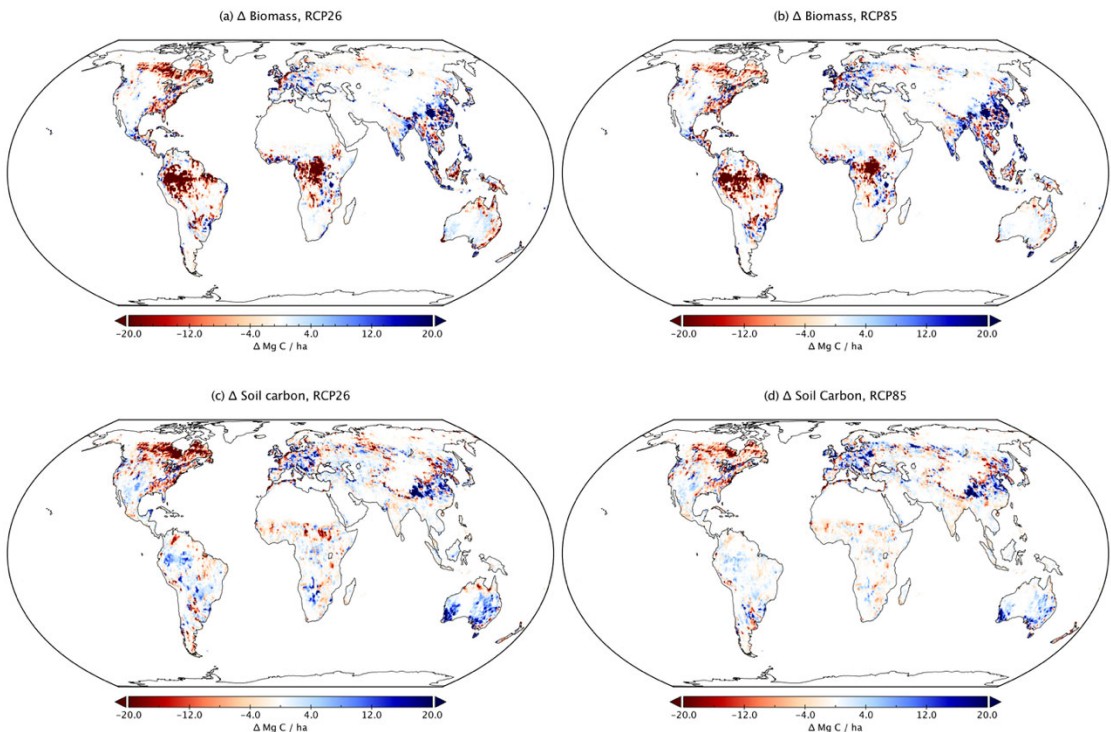

**Figure 15: Spatial distribution of land-use-induced changes in terrestrial ecosystem carbon stock. Results are for (a, b) vegetation biomass and (c, d) soil carbon stock. Average of the five climate projection-based simulations under (a, c) RCP2.6 and (b, d) RCP8.5 scenarios in the 2090s.**

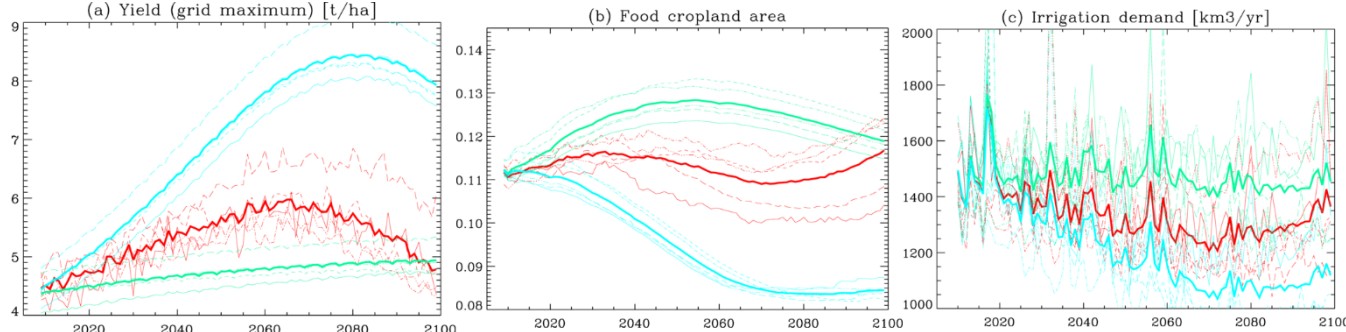

**Figure 16: Time series of changes in a) cropland yield (maximum across five crops in each grid, t/ha), b) food cropland area (a fraction of total land area), and irrigation demand (km$^3$/yr) based on the forcings of the five climate models under the RCP8.5 scenario. Simulations with climatic factors and CO$_2$ concentrations fixed at 2006 (light green, noCL+noFE), those with climatic factors fixed (cyan, noCL+FE), and those with varying climate and CO$_2$ concentrations (red, CL+FE).**

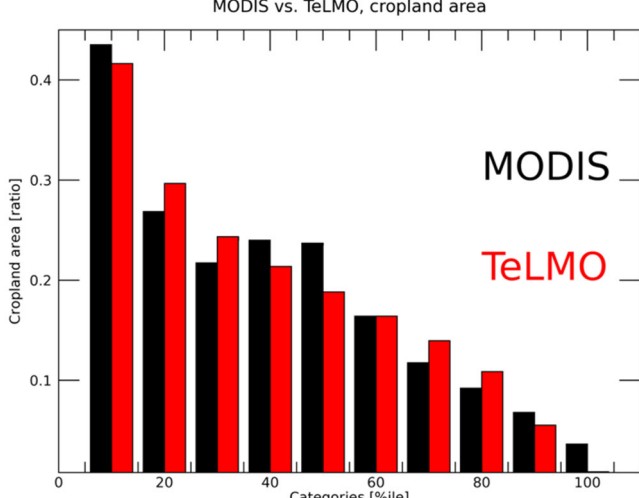

**Figure B-1: Comparison of the global MODIS cropland area and the calculated area using the agricultural suitability index (ASI). Here, 23,000 randomly selected cropland area values are arranged in descending order and divided into 10 categories; the average value of MODIS (black) and ASI values calculated by TeLMO (red) in each category are compared. The horizontal axis is the higher percentile of cropland area data that is randomly selected from the global 0.5° grids at year 2005.**

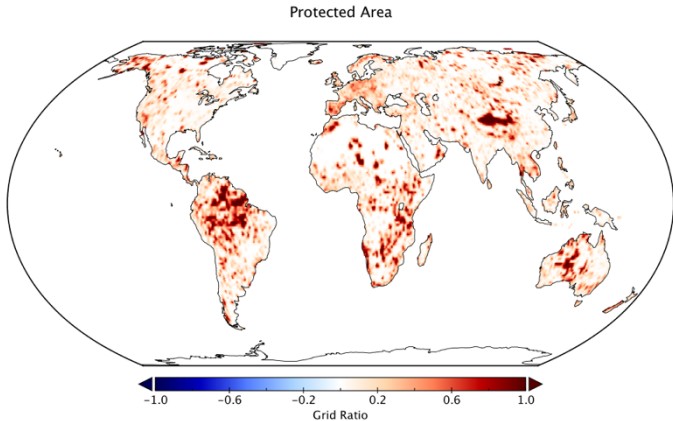

**Figure B-2: Global distribution of areas protected from bioenergy production.**

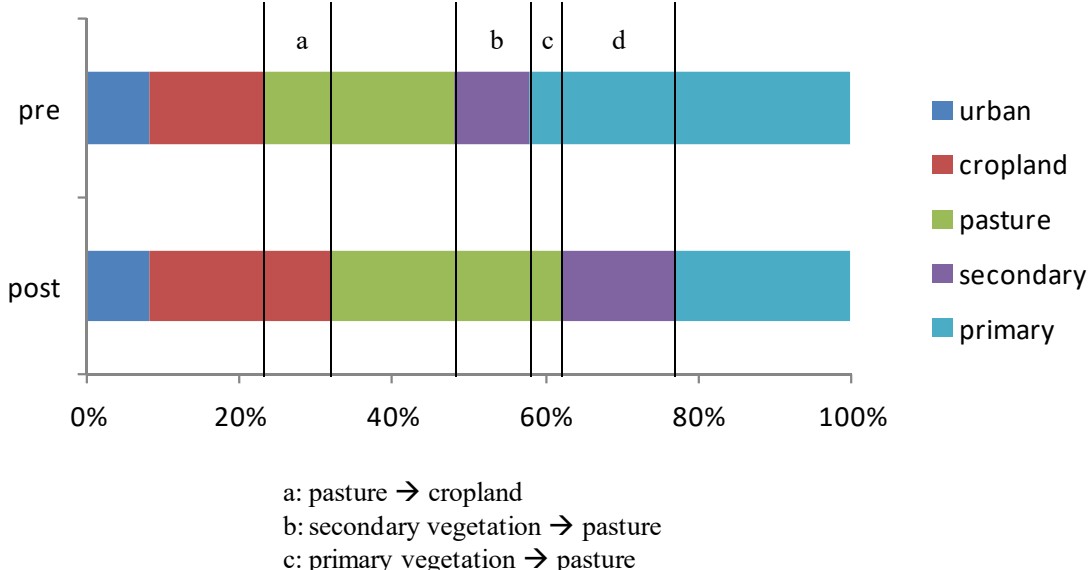

a: pasture → cropland
b: secondary vegetation → pasture
c: primary vegetation → pasture
d: primary vegetation → secondary vegetation

**Figure B-3: Schematic diagram of landcover transition. Details are explained in the main text.**

**Table B-1 Correspondence of landcover type in land-use model and transition matrix.**

| Landcover type in land-use model | Landcover type in transition matrix |
|---|---|
| Urban | Urban |
| Cropland (food) | Cropland |
| Cropland (bio-crop) | |
| Pasture | Pasture |
| Managed forest | Secondary land |
| Unmanaged forest | Primary land |
| | Secondary land |
| Grassland | Primary land |
| | Secondary land |
| Other | - |