# Peer review of "MIROC-INTEG-LAND version 1: A global bio-geochemical land surface model with human water management, crop growth, and land-use change"

_Geoscientific Model Development, 2019_

## Referee Comment (RC1) · Anonymous Referee #1 · 3 Dec 2019

**GENERAL COMMENTS**

In this manuscript, Yokohata et al. describe a model integration that brings together (mostly) process-based representations of land use decisionmaking, land surface, hydrology, vegetation, and agriculture. This is part of an important recent trend in making integrated models that actually account for the effects of changing climate and CO2 on agricultural productivity, and how that changing productivity will affect the future trajectory of land use. The manuscript is well-written, with a decent amount of technical detail, as one would hope for a paper in this journal. However, I have reservations about how the integrated model is framed at the beginning of the manuscript. For that

reason, coupled with a number of clarifications that need to be made, I suggest that the paper be resubmitted with major revisions.

SPECIFIC COMMENTS

The beginning of the manuscript sets up a loftier goal than is actually achieved by the presented model. I got the impression that the climate system was included as a component, while in reality the demonstrated version uses offline climate forcings. This begins with the model name, which includes MIROC—a well-known climate model. This idea is reinforced in the Abstract at P1 L27–28, in the Introduction at P3 L20–22, and in Sect. 2 at P4 L8–10. In fact, it's unclear why MIROC is included at all—its only relevance to the work presented here is that it's the source of one of the five climate forcings used. It would make more sense to call the integrated model presented here INTEG1, and reserve "MIROC-INTEG1" for a future version that does actually include MIROC coupling.

In addition to these specific instances of the authors claiming an integration that does not appear to exist, the text of the Introduction sets up MIROC-INTEG1 as being able to "examine the impact of land-use change on the climate system" (P3 L25) and "quantitatively evaluate the interactions and feedback related to climate, water, crop, land use, and ecosystem" (P3 L27). While the work here gets at a proxy variable—terrestrial carbon storage—that relates to the land-atmosphere carbon flux, actually directly assessing land-use impact on climate is impossible without coupling to a climate model. It is also unclear exactly what feedbacks the authors are referring to at P3 L27, considering that most of the sub-models do not seem to be connected in a two-way manner (see Robinson et al., 2018 Fig. 2).

Coupling a climate model is clearly outside the scope of the present work, but I want to reiterate that the analyses presented ARE publication-worthy—they just need to be set up in a less misleading way.

I am also concerned by the relative lack of space spent evaluating TeLMO. Considering

that it is the primary piece of model development introduced in this manuscript (as opposed to integration of existing models), the authors should evaluate more than just regional and global cropland area over 12 years (Fig. 7, which by the way needs area units specified). At the very least, similar analyses need to be presented for pasture and forest area.

I also have a number of other issues I would like the authors to address in a revised version.

- Is livestock feed production included in crop demand?

- Why do protected areas protect only from bioenergy and not food crops or pasture?

- Sect. A.5 (P19 L15–21): What is "irrigation water stock" and how does it relate to things calculated in HiGWMAT? It seems completely separate, since the controlling parameter Irr_capacity "is estimated at each cell of the grid by MCMC" (P19 L20–21).

- By using the Kato & Yamagata (2014) biofuel yields map, it seems that changes in biofuel yield are not considered. Could this be added by scaling based on changing yield of wheat and/or maize?

- Historical domestic and industrial water extractions are taken from FAOSTAT; how are projected extractions calculated?

- With regard to adjustment factors for matching LUH2 or AIM land use areas/demand: What is the justification for using a gridcell-level adjustment factor for cropland but region-level adjustment factors for others?

- Fig. 2 should be revised to separate (a) input data coming from outside the model system and (b) data being moved between sub-models. Generally, using different kinds of boxes (rounded rectangles vs. circles, for instance) for models vs. data would be helpful. "Atmosphere" should be added to the arrow pointing to Land Ecosystem.

TECHNICAL CORRECTIONS

- P3 L2: Misspelling: "temperture"

- Throughout: "Chapters" should be referred to as "sections"

- P6 L25: Unnecessary comma in "et al., (2011)"

- P11 L21: "LUH2h2v"

- P12 L6: What is "mascon"?

- P13 L10: Closing parenthesis should come after "satellite"

- P14 L1: "reproducibility" doesn't seem like the right word. Perhaps "performance"?

- P14 L13: 2a should be 8b

- P14 L20: "global average" should be "global total" or just "total"

- P16 L15: I'm unclear as to what this explanation means: "Soil carbon is less impacted by the land-use change compared to the above-ground biomass, likely because of the carbon supply from crops in the VISIT calculation."

- P16 L20–22: It is unclear how increasing above-ground biomass would negatively affect ecosystem services.

- P21 L9–10: This is unclear; please revise.

- P28 L6: Citation missing

- Fig. 4: Labels (a) and (b) should be referenced in caption.

- Fig. 10: Y-axis units?

- Fig. B-1: (1) Bin boundaries should be labeled in terms of real units. (2) What year(s) are being compared? At what resolution? (Include this info in caption, not just main text.)

WORKS CITED IN REVIEW

Robinson, D.T., Di Vittorio, A., Alexander, P., Arneth, A., Barton, C.M., Brown, D.G., Kettner, A., Lemmen, C., O'Neill, B.C., Janssen, M., Pugh, T.A.M., Rabin, S.S., Rounsevell, M., Syvitski, J.P., Ullah, I., Verburg, P.H., 2018. Modelling feedbacks between human and natural processes in the land system. Earth Syst. Dynam. 9, 895–914. doi:10.5194/esd-9-895-2018.

---

## Referee Comment (RC2) · Anonymous Referee #2 · 18 Dec 2019

Review of gmd-2019-184: MIROC-INTEG1: A global bio-geochemical land surface model with human water management, crop growth, and land-use change

Summary

The authors present a new land model that includes the effects of climate on land allocation by constraining irrigation due to water availability and by calculating yield based on current climate. The primary novelty here is that the land allocation scheme has been included in the land component of an earth system model. This enables land allocation to be determined by the changing state of the earth system in conjunction with estimates of demand for food and other land-based commodities. The

model reproduces historical conditions well, and future projections show reasonable results. Future goals include full coupling with the atmosphere and ocean components to incorporate additional human-earth system feedbacks.

Overall impression

This is a big step toward full human-earth system coupling, with a couple of novel developments including the impacts of water scarcity on land allocation and the inclusion of land allocation in the land model. My main concern is that these developments presented in this paper are not highlighted as providing new information. The examples do not show the benefits of these developments over not having them, and as such their value is not made clear. The paper can be strengthened by some reframing that brings these novel improvements to forefront, along with more critical examination of their strengths and weaknesses. I recommend some considerable revisions, and please see the detailed comments that follow.

1) There are two novel developments here: water availability effects on irrigation/land allocation, and the inclusion of land allocation in the land model. They each have unique contributions that should be highlighted. The inclusion of land allocation in the land model is unique and enables direct response of land use to changing conditions, including both the climate and the water availability as determined through the hydrological model. While there is not yet feedback with the atmosphere, the response of growth/yield to climate is more detailed than is otherwise considered in IAMs and some other land use models, and is also directly embedded in the full land model, which is a feat in itself. The water/irrigation linkage to land allocation is even more novel, as there have been only regional studies on this with loose coupling, I know of only one IAM that has just finally made this work but without connections to a full land model. The uniqueness of this new system should be more clearly defined such that your examples show the benefits of these developments.

2) Provide examples that show the benefits of your novel developments. This will

require additional simulations that shut off water scarcity effects on potential irrigation and alternatively shut off the climate effects on yield. As it is, your examples just show outputs that can be generated by a variety of other models. You want to highlight the value added of your developments.

3) Discuss how these developments relate to existing alternatives and what the limitations are. For example, IAMs and other land models can project land use/cover under changing climate and feed this information to a land model. Why not just do this in AIM and feed it to MIROC? What do you gain and lose with TeLMO inside MIROC? How is TeLMO different from AIM and when do you expect one to be more robust than the other? The same questions could be asked with respect to water.

4) One concern I have about the model itself is the inconsistency between the crop model for growth/yield and the biogeochemical/biophysical model for cropland. You may not be able to fix this right now, but it is a problem that there are two different crop growth models to represent different processes of the same land area. In particular, your yield model does not have explicit fertilizer, but your VISIT model does. For a variety of reasons, the growth values will not be the same between the two representations, but they should be because the growth determines the geochemical and physical characteristics of the cropland. In the end your yields are not consistent with how cropland affects the geochemical and physical processes in the land model that will eventually feedback to the atmosphere. This should be fixed before full coupling with the atmosphere.

5) Some of the description is not very clear or complete for the reader to understand what has been done or how the components interact. See comments below for details on what needs clarification. In particular, it isn't clear how the non-cropland is affected by climate in this model.

Specific comments and suggestions

Abstract

The abstract is not clear about what is presented here or what the outcomes of evaluation are. While some interactions from climate to land allocation are included, feedbacks between the land system and climate are not because the atmospheric inputs are fixed. Be clear about the novelty here, and state how well the evaluation performs.

Introduction

page 2, line 21: Bond-Lamberty this reference is incorrect throughout

Model structure

Sub models

It seems that there are two crop models: PRYSBI2 and also one in HiGWMAT Please clarify how these are different and why they are separate.

VISIT

page 8, lines 12-13: abandoned cropland recovers to mean biomass of what? and is always considered secondary, or can secondary land revert to primary?

page 8, lines 14-16: these fertilizer and crop calendar inputs seem inconsistent with the crop model. if the crop model doesn't use fertilizer inputs, then how are they used in VISIT? if crop growth is calculated with implicit fertilizer, then this specific nitrogen input doesn't match. And why would VISIT be using a crop calendar and not the crop model? your biogeochemical fluxes are not going to correspond with your crop growth.

Model coupling

Experimental settings

page 11, lines 3-19: which SSPs for which RCPs?

Historical simulations

page 13, lines 1-3:

The results in figure 5 do not all line up along the 1:1 line. You have to adjust this statement.

page 14, lines 1-2: TeLMO crop area is not very similar to AIM, and it is more similar to FAO in most cases presented. Since AIM is a driver of TeLMO, more explanation is required here of why they are different. Furthermore, the similarity to FAO is more compelling as evidence for usability of TeLMO, than any similarity to AIM.

Future simulations

page 15, line 8: why the maximum value? this would underestimate the land area because each crop may be grown in a cell, with varying yields.

page 15, lines 19-21: what is the basis for your bioenergy crop calculations here?

page 16, lines 23-24: Not sure what you mean here. You haven't untangled land use effects here, just showed results of land area changes and the biomass affected. Many ecosystem and earth and integrated assessment models do this. what is unique here? If you were to show how the land allocation and the biomass effects differed due to the climate effects on yields, then this would show the benefit of this model. To do this you have to do another set of runs where the yields are the base year yields plus the non-climate changes in yield, and compare these to the runs you have done. Alternatively, you could show how the inclusion of water availability in determining irrigation changes the crop area/production by turning off the irrigation dependence on the available water.

Implications

page 16, lines 30-31: this is where things get more interesting

page 17, lines 6-8: it seems like you have outputs in this paper that could be used for this

Figures and tables

Figure 3 What do you mean by multi-model mean? You are using only one model here.

Figure 10 What is the vertical axis? Is it the annual change in cropland area as a fraction of total land area, or is it the cropland in that year as a fraction of total land area?

Appendix A

page 17, line 28: what are these? can you give the types of data if not able to list them all here?

page 19, line 3: i think you mean water stress and shouldn't this go into section A.5?

page 19, line 11: so there is a cold stress factor applied on top of the temperature dependent equation?

page 20, lines 7-16: it isn't clear what this is or how it is applied. is it a fraction that is applied to a given yield and then added to that yield? what are equations 32 and 33?

Appendix B

page 22, lines 7-11: it isn't clear why you are using the max yield, and it sounds like it is per crop here, while in the text it sounded like it was the max across crops. don't you need to apply each crop yield to its own prices in the 30sec cells to get distinct ASI values for crops? maybe a crop with a lower yield has higher ASI due to higher price. so are you essentially just selecting one crop for the half-degree cell? this will underestimate cropland.

page 25, lines 20-21: for bioenergy crops, does this mean that these are fixed values, or are they dependent on the atmospheric inputs? More info is needed here.

page 26, lines 8-24: Is this NPP for pasture calculated beforehand, offline? Does it include matching climate drivers to the scenarios here? Or are these NPP numbers independent of the climate change?

page 27, line 10: where is the base-year managed forest area from?

page 27, lines 14-19: also for forest, are these NPP values calculated beforehand, offline? and are there matching values corresponding with the scenarios here, or are they a single set of values for all sims?

---

## Author Comment (AC1) · 31 Mar 2020

*GENERAL COMMENTS*

*In this manuscript, Yokohata et al. describe a model integration that brings together (mostly) process-based representations of land use decision-making, land surface, hydrology, vegetation, and agriculture. This is part of an important recent trend in making integrated models that actually account for the effects of changing climate and CO2 on agricultural productivity, and how that changing productivity will affect the future trajectory of land use. The manuscript is well-written, with a decent amount of technical detail, as one would hope for a paper in this journal. However, I have reservations about how the integrated model is framed at the beginning of the manuscript. For that reason, coupled with a number of clarifications that need to be made, I suggest that the paper be resubmitted with major revisions.*

*SPECIFIC COMMENTS*

*The beginning of the manuscript sets up a loftier goal than is actually achieved by the presented model. I got the impression that the climate system was included as a component, while in reality the demonstrated version uses offline climate forcings. This begins with the model name, which includes MIROC—a well-known climate model. This idea is reinforced in the Abstract at P1 L27–28, in the Introduction at P3 L20–22, and in Sect. 2 at P4 L8–10. In fact, it's unclear why MIROC is included at all—its only relevance to the work presented here is that it's the source of one of the five climate forcings used. It would make more sense to call the integrated model presented here INTEG1, and reserve "MIROC-INTEG1" for a future version that does actually include MIROC coupling.*

Thank you very much for your suggestions. The reason why the name of the model is MIROC-INTEG is because the models of terrestrial ecosystem, water management, crop growth and land use are combined with the land surface model (MATSIRO) included in MIROC, and because this integrated model will also be combined with the Earth System Model (MIROC-ES2L, Hajima et al. 2020) in the ongoing work. Various improvements have been made to the MIROC land surface model, MATSIRO (Takata et al. 2003, Nitta et al. 2014, Pokhrel et al. 2012). Unlike standard hydrological models, it is possible for MATSIRO to consistently solve complicated processes related to energy and water balances on land. One of the advantages of running the land surface model alone in MIROC is that it can be used for assessing the impacts of climate change on land, taking into account the uncertainty of future climate projections. HiGWMAT (Pokhrel et al. 2014), MATSIRO combined with a water resources model, has contributed to the Inter-Sector Impact Assessment Project (ISIMIP). On the other hand, since MIROC is the name of a well-known climate model, as the reviewers point out, the model name "MIROC-INTEG" can be misleading to the reader by implying that it involves air-land surface interactions. For this reason, we have re-named the model "MIROC-INTEG-LAND" in the revised manuscript. In addition, it is clearly described that MIROC-

INTEG-LAND couples the land surface model with various sub-models, and it does not include interaction with the atmosphere. Furthermore, important features of the land surface model in MIROC were described earlier in the paper, and the advantages of running only the land surface model in MIROC were described.

The model title has been changed as follows.
MIROC-INTEG-LAND version 1: A global bio-geochemical land surface model with human water management, crop growth, and land-use change

Because of this modification, the model name in the original manuscript (MIROC-INTEG1) is changed to MIROC-INTEG-LAND in the revised manuscript. The abstract was modified as follows.

To investigate these interrelationships, we developed MIROC-INTEG-LAND (MIROC INTEGrated LAND surface model version 1), an integrated model that combines the land surface component of global climate model MIROC (Model for Interdisciplinary Research on Climate) with water resources, crop production, land ecosystem, and land use models.

The introduction was changed as follows.

The model is based on the land surface component of global climate model MIROC (Model for Interdisciplinary Research on Climate version: Watanabe et al., 2010), into which we have incorporated water resources, land-ecosystem, crop growth, and land use models.

The first paragraph of Section 2 is modified to explain that MIROC-INTEG is based on the land surface component of MIROC, MATSIRO. In addition, the advantages of MATSIRO and running the land surface model alone are also explained.

The distinctive feature of MIROC-INTEG-LAND (Figure 1) is that it couples natural ecosystem and human activity models to the land surface component of MIROC, a state-of-the-art global climate model (Watanabe et al., 2010). The MIROC series is a global atmosphere-land-ocean coupled global climate model, one of the models contributing to the Coupled Model Inter-comparison Project (CMIP). MIROC's land surface component, MATSIRO (Minimal Advanced Treatments of Surface Interaction and Runoff, Takata et al. 2003, Nitta et al., 2014) can consider the energy and water budgets consistently on the land grid with a spatial resolution of 1 degree. MIROC-INTEG-LAND performs its calculations over the global land area only, and neither the atmosphere nor ocean components of MIROC are coupled. One of the advantages of running only the land surface model is

that it can be used to assess the impacts of land on climate change, taking into account the uncertainties of future atmospheric projections.

*In addition to these specific instances of the authors claiming an integration that does not appear to exist, the text of the Introduction sets up MIROC-INTEG1 as being able to "examine the impact of land-use change on the climate system" (P3 L25) and "quantitatively evaluate the interactions and feedback related to climate, water, crop, land use, and ecosystem" (P3 L27). While the work here gets at a proxy variable—terrestrial carbon storage—that relates to the land-atmosphere carbon flux, actually directly assessing land-use impact on climate is impossible without coupling to a climate model. It is also unclear exactly what feedbacks the authors are referring to at P3 L27, considering that most of the sub-models do not seem to be connected in a two-way manner (see Robinson et al., 2018 Fig. 2). Coupling a climate model is clearly outside the scope of the present work, but I want to reiterate that the analyses presented ARE publication-worthy—they just need to be set up in a less misleading way.*

Thank you very much for your suggestions. In response to this comment, the two sentences below have been removed in the revised manuscript.

P3 L25 in the original manuscript
By taking into account changes in the socio-economic scenario, it is possible to examine the impact of land-use change on the climate system while simultaneously investigating the impact of climate change on the water and food sector.
P3 L27: MIROC-INTEG1 can quantitatively evaluate the interactions and feedback related to climate, water, crop, land use, and ecosystem. Such an evaluation is simply not possible with conventional integrated assessment and earth system models.

Instead, we added new section 2.2 to state the novelty of MIROC-INTEG-LAND more clearly. We also clearly stated that some of the interaction in MIROC-INTEG-LAND is one-way, but there are some advantages which were not treated in the conventional integrated assessment models (IAMs).

*I am also concerned by the relative lack of space spent evaluating TeLMO. Considering that it is the primary piece of model development introduced in this manuscript (as opposed to integration of existing models), the authors should evaluate more than just regional and global cropland area over 12 years (Fig. 7, which by the way needs area units specified). At the very least, similar analyses need to be presented for pasture and forest area.*

Thank you very much for your suggestion. We evaluated the regional and global pasture and forest area over 12 years as in Figure 7. This is added in the revised manuscript and explained as follows.

Figure 8 shows a comparison of TeLMO, AIM, and LUH data for pasture. Unlike cropland, pastures are compared with LUH data because there are no long-term global observation data. TeLMO calculates pasture lands such that the area matches that in the AIM for the AIM calculation domain (17 regions around the world). Because AIM treats China and the United States as one region, the results of TeLMO and AIM for China, the United States, and the globe are almost the same. On the other hand, in Australia, TeLMO is closer to LUH. Similarly, Figure 9 shows a comparison between TeLMO, AIM, and FAO data of forest area. TeLMO refers to MODIS data and calculates forest area taking into account deforestation and changes in crop area. Some difference between TeLMO and FAO can be seen, but the two are relatively close. Overall, TeLMO, AIM, and FAO closely agree at the regional scale.

Captions for Figure 8 and 9 are added as follows.

Figure 8: Same as Figure 7, but for the comparison of historical pasture area simulated by MIROC-INTEG (red), AIM/CGE (blue), and LUH (black), using the ratio of cropland area to total area.

Figure 9: Same as Figure 7, but for the historical forest area simulated by MIROC-INTEG (red), AIM/CGE (blue), and FAO (black), using the ratio of cropland area to total area.

According to this modification, the numbering of figures is modified.

*I also have a number of other issues I would like the authors to address in a revised version.*
*- Is livestock feed production included in crop demand?*
In this version, livestock feed production is not included in the crop demand. It is explained in the revised manuscript.

In this study, livestock feed demand is not included in $X_{1,k}$.

*- Why do protected areas protect only from bioenergy and not food crops or pasture?*
In this study, we did not consider the protected area for the calculation of the food cropland and pasture, by assuming that food has a higher priority than ecosystem protection. This point has been explained in the revised manuscript as follows.

In this study, we did not consider the protected area for the calculation of the food cropland and pasture, under the assumption that food has a higher priority than ecosystem protection.

*- Sect. A.5 (P19 L15–21): What is "irrigation water stock" and how does it relate to things calculated in HiGWMAT? It seems completely separate, since the controlling parameter Irr_capacity "is estimated at each cell of the grid by MCMC" (P19 L20–21).*

A part of the explanation in the original manuscript was not for MIROC-INTEG. The "irrigation water stock" and "*Irr_capacity*" were not calculated in PRYSBI2 of MIROC-INTEG. The soil water calculated in HiGWMAT by considering irrigation (Section 3.1) was passed to PRYSBI2, and then the water stress was calculated in PRYSBI2 by using the soil water. It is explained in the revised manuscript. The below explanation in the original manuscript is removed in the revised version.

The soil water balance in version 2.2 is modeled using a method similar to that described by Neitsch et al., (2005), with two soil layers and no lateral flow. In this method, the water content in each soil layer is updated daily to account for rainfall, snowmelt, sublimation, transpiration, evaporation, and percolation. However, our model does not consider the nitrogen cycle. Moreover, we do not use the irrigation sub-model used in the SWAT model. Instead, we use a simple protocol in which irrigation water is supplied to the top layer of the soil if the crop experiences water stress. Irrigation water is supplied until its stock is exhausted. The size of the irrigation water stock is determined by the parameter Ircapacity, which is estimated at each cell of the grid by MCMC.

Instead, the method for the calculation of water stress is added in the revised manuscript as follows.

In PRYSBI2, the calculation of water stress follows the SWAT (Neitsch et al., 2005) algorithm. In SWAT, the daily water stress is calculated according to soil water, soil characteristics (field capacity and water content at saturation), root depth and crop field evapotranspiration. PRYSBI2 uses the soil water calculated in HiGW-MAT as explained in Section 3.2. The crop field evapotranspiration is calculated in SWAT according to the leaf area index.

*- By using the Kato & Yamagata (2014) biofuel yields map, it seems that changes in biofuel yield are not considered. Could this be added by scaling based on changing yield of wheat and/or maize?*

In Kato and Yamagata (2014), the future changes in climate and fertilizer input are considered. It is explained in the revised manuscript as follows.

For biofuel crop yield $y_{(bio,j)}$, the yield for miscanthus or switchgrass, whichever is greater in a given cell, is calculated for the entire globe by using the biofuel crop model developed in Kato and Yamagata

(2014). The biofuel crop model in Kato and Yamagata (2014) considers the future changes in climate based on the RCP scenarios. In this study, we also consider the future changes in fertilizer input based on the SSPs adopted in Mori et al. (2018). Because of the uncertainty in future fertilizer application for crop management, we set the high end of the N fertilizer input threshold according to Tilman et al. (2011). The nitrogen fertilizer application was set to increase from the current level according to the increasing rate of GDP in the SSP2 scenario up to 160 kg N ha$^{-1}$ yr$^{-1}$ if the fertilizer input at the country level was below 160 kg N ha$^{-1}$ yr$^{-1}$ in the 2000s. Also, the phosphorus fertilizer input in each country was set to follow the same annual increase rate as the nitrogen fertilizer application.

*- Historical domestic and industrial water extractions are taken from FAOSTAT; how are projected extractions calculated?*
We also use the same data for the future projection. In the revised manuscript, we described this as follows.

While irrigation demand is simulated by the irrigation module, domestic and industrial water uses are prescribed based on the AQUASTAT database of the Food and Agricultural Organization (FAO; see Pokhrel et al., 2012b). We use the same prescribed values for domestic and industrial water uses in both historical and future simulations, as future projections of water withdrawal are not available.

*- With regard to adjustment factors for matching LUH2 or AIM land use areas/demand: What is the justification for using a gridcell-level adjustment factor for cropland but region-level adjustment factors for others?*
As described in Eq. (B-16), the adjustment factor for pasture is formulated as grid-cell level ($C_{past,j}$, where $j$ denotes index for the 0.5° cell), by using the LUH2 historical data. We use the region-level adjustment factors for managed forest ($C_{manfr,k}$, where $k$ denotes index for the 17 regions defined in AIM) because the grid-level reference data is not available. This is explained in the revised manuscript.

We use the region-level adjustment factors for managed forest ($C_{manfr,k}$) because the grid-level reference data is not available.

*- Fig. 2 should be revised to separate (a) input data coming from outside the model system and (b) data being moved between sub-models. Generally, using different kinds of boxes (rounded rectangles vs. circles, for instance) for models vs. data would be helpful. "Atmosphere" should be added to the arrow pointing to Land Ecosystem.*

According to the suggestion, Fig. 2 is revised. Input data is outside the model system. We use different kinds of boxes for models and data. We also added arrows pointing from "Atmosphere" to Land ecosystem. Data box cannot be moved between sub-models, but we use a similar format to that used in Robinson et al. 2018, Figs. 3-6, where the data box is placed at the right. The caption of Figure 2 is modified as follows.

Boxes indicate the sub-models and data. For the sub-models, the name and time-step of the models are indicated in the boxes. In the "data" box, the name of the variable saved as a file is 5 indicated. In the "input data" box, information regarding the input data is indicated.

*- P3 L2: Misspelling: "temperture"*

*- Throughout: "Chapters" should be referred to as "sections"*

*- P6 L25: Unnecessary comma in "et al., (2011)"*

*- P11 L21: "LUH2h2v"*

Thank you very much for the corrections. They are corrected in the revised manuscript.

*- P12 L6: What is "mascon"?*

We modified the manuscript as follows. We also added new citations (Save et al. 2016, Watkins et al. 2015, Wiese et al. 2016).

For the GRACE data, we use the mean of mass concentration (mascon) products from the Center for Space Research (CSR; Save et al., 2016) at the University of Texas at Austin and the Jet Propulsion Laboratory (JPL; Watkins et al., 2015; Wiese, Yuan, et al., 2016) at the California Institute of Technology.

In the caption of figure 3, it is also described as follows.

The GRACE data shown are the mean of the mass concentration products from two processing centers: CSR and JPL.

*- P13 L10: Closing parenthesis should come after "satellite"*

*- P14 L1: "reproducibility" doesn't seem like the right word. Perhaps "performance"?*

*- P14 L13: 2a should be 8b*

*- P14 L20: "global average" should be "global total" or just "total"*

Thank you very much again for the corrections. They are corrected in the revised manuscript.

*- P16 L15: I'm unclear as to what this explanation means: "Soil carbon is less impacted by the land-use change compared to the above-ground biomass, likely because of the carbon supply from crops in the VISIT calculation."*

In the revised manuscript, the reason for "less impacted" is added as follows.

The decrease in soil carbon after deforestation is much smaller than the decrease in above-ground biomass, as the carbon supply from crop residue compensates for the soil carbon loss.

*- P16 L20–22: It is unclear how increasing above-ground biomass would negatively affect ecosystem services." In Asia, the decrease in food cropland area tends to increase the above-ground biomass in both the RCP2.6 and RCP8.5 scenarios. Accordingly, even under the mitigation-oriented scenario, considerable changes in ecosystem structure and functions would occur in certain regions, leading to serious deterioration in ecosystem services."*

The description in the original manuscript was misleading, and thus we modified the manuscript as follows.

The impact on above-ground biomass is projected to be greater in northwest South America, central Africa, northeast North America, and Australia, where the bioenergy cropland area is expanding. In these regions, even under the mitigation-oriented scenario, considerable declines in ecosystem structure and functions would occur, leading to deterioration, for example, of habitats for natural organisms, water holding capacity, and soil nutrients. Consequently, these functional degradations would degrade ecosystem services such as biodiversity, regulation, and provision. On the other hand, in Asia, the decrease in food cropland area tends to increase the above-ground biomass in both the RCP2.6 and RCP8.5 scenarios, possibly leading to leading to the enhancement of above-ground biomass, and thus ecosystem services.

*- P21 L9–10: This is unclear; please revise.*

This paragraph describes the differences between TeLMO and the integrated assessment model (IAMs). In general, the IAMs are not grid-based, but divides the world into dozens of regions and describes economic activity in these regions. Therefore, the IAMs 1) calculate the area of agricultural land by using the information of the yield averaged over these regions based on the balance between supply and demand, and 2) allocate the agricultural land by the down-scale method (e.g., Hasegawa et al. 2017). As pointed out by previous works (Alexander et al. 2017), the problem with this method is that it is not possible to explicitly consider spatiotemporal information such as crop yield and production cost when determining land use change in the procedure of 1). TeLMO solves this problem, making it possible to consistently consider the spatiotemporal information such

as crop yields and the balance between supply and demand when allocating the agricultural land, by using the Food Cropland Down-scale Module and the International Trade Module. This is a very important point and was explained in a new section of the text, "2.2 Novelty of MIROC-INTEG-LAND".

*- P28 L6: Citation missing*
The citation is added to the revised manuscript (Friedl et al. 2010).

*- Fig. 4: Labels (a) and (b) should be referenced in caption.*
The caption is modified in the revised manuscript as follows.

Figure 4: Comparison of irrigation demands simulated by MIROC-INTEG-LAND (a) with the results from offline simulations using HiGW-MAT (b) forced by observed climate forcing data (Pokhrel et al., 2015) for 1°×1° grids shown as the mean for 1998- 2002 period.

*- Fig. 10: Y-axis units?*
It is the cropland in that year as a fraction of total land area. The caption of Figure 12 (Figure 10 in the original manuscript) is modified in the revised manuscript as follows.

Figure 12: Time series of changes in cropland area based on the forcings of the five climate models. The vertical axis is the cropland area as a fraction of total land area.

*- Fig. B-1: (1) Bin boundaries should be labeled in terms of real units. (2) What year(s) are being compared? At what resolution? (Include this info in caption, not just main text.)*
The information on (1)-(3) is included in the caption of Fig. B-1. The label in Fig. B-1 is also modified. The caption of Figure B-1 is modified as follows.

Figure B-1: Comparison of the global MODIS cropland area and the calculated area using the agricultural suitability index (ASI). Here, 23,000 randomly selected cropland area values are arranged in descending order and divided into 10 categories; the average value of MODIS (black) and ASI values calculated by TeLMO (red) in each category are compared. The horizontal axis is the higher percentile of cropland area data that is randomly selected from the global 0.5 degree grids at year 2005.

In the main text of Appendix B1.1, the sentence is modified as follows.

The logistic regression coefficient was derived from 23,000 data values that were randomly selected from the set of global 0.5° grids at year 2005.

**WORKS CITED IN REVIEW**

*Robinson, D.T., Di Vittorio, A., Alexander, P., Arneth, A., Barton, C.M., Brown, D.G., Kettner, A., Lemmen, C., O'Neill, B.C., Janssen, M., Pugh, T.A.M., Rabin, S.S., Roun- sevell, M., Syvitski, J.P., Ullah, I., Verburg, P.H., 2018. Modelling feedbacks between human and natural processes in the land system. Earth Syst. Dynam. 9, 895–914. doi:10.5194/esd-9-895-2018.*

---

## Author Comment (AC2) · 31 Mar 2020

*Summary*

*The authors present a new land model that includes the effects of climate on land allocation by constraining irrigation due to water availability and by calculating yield based on current climate. The primary novelty here is that the land allocation scheme has been included in the land component of an earth system model. This enables land allocation to be determined by the changing state of the earth system in conjunction with estimates of demand for food and other land-based commodities. The model reproduces historical conditions well, and future projections show reasonable results. Future goals include full coupling with the atmosphere and ocean components to incorporate additional human-earth system feedbacks.*

*Overall impression*

*This is a big step toward full human-earth system coupling, with a couple of novel developments including the impacts of water scarcity on land allocation and the inclusion of land allocation in the land model. My main concern is that these developments presented in this paper are not highlighted as providing new information. The examples do not show the benefits of these developments over not having them, and as such their value is not made clear. The paper can be strengthened by some reframing that brings these novel improvements to forefront, along with more critical examination of their strengths and weaknesses. I recommend some considerable revisions, and please see the detailed comments that follow.*

*1) There are two novel developments here: water availability effects on irrigation/land allocation, and the inclusion of land allocation in the land model. They each have unique contributions that should be highlighted. The inclusion of land allocation in the land model is unique and enables direct response of land use to changing conditions, including both the climate and the water availability as determined through the hydrological model. While there is not yet feedback with the atmosphere, the response of growth/yield to climate is more detailed than is otherwise considered in IAMs and some other land use models, and is also directly embedded in the full land model, which is a feat in itself. The water/irrigation linkage to land allocation is even more novel, as there have been only regional studies on this with loose coupling, I know of only one IAM that has just finally made this work but without connections to a full land model. The uniqueness of this new system should be more clearly defined such that your examples show the benefits of these developments.*

Thank you for suggesting a description of the novel developments in the paper. The novelty of the model is summarized in a new section (2.2 Novelty of MIROC-INTEG-LAND). The name of the model has been changed based on the suggestion of reviewer #1. In the revised manuscript, the title of Section 2 is "Overall feature of MIROC-INTEG-LAND", and that of section 2.1 is "Model

structure". In the new section 2.2, we clearly emphasized the novelty of MIROC-INTEG-LAND as suggested. We also summarized the novelty of the model in Abstract.

*2) Provide examples that show the benefits of your novel developments. This will require additional simulations that shut off water scarcity effects on potential irrigation and alternatively shut off the climate effects on yield. As it is, your examples just show outputs that can be generated by a variety of other models. You want to highlight the value added of your developments.*

We again appreciate your suggestions. We agree that additional simulations that shut off the interactions between the sub-models should be helpful to show the benefits of our developments. Our response to these points is explained in the reply to your comments on page 16, lines 23-24 below.

*3) Discuss how these developments relate to existing alternatives and what the limitations are. For example, IAMs and other land models can project land use/cover under changing climate and feed this information to a land model. Why not just do this in AIM and feed it to MIROC? What do you gain and lose with TeLMO inside MIROC? How is TeLMO different from AIM and when do you expect one to be more robust than the other? The same questions could be asked with respect to water.*

The difference between TeLMO and IAMs, and the advantages of coupling (running the sub-models together) are discussed in the second and third paragraphs in Section 2.2 "Novelty in MIROC-INTEG-LAND" as follows.

2.2 Novelty of MIROC-INTEG-LAND

An important feature of MIROC-INTEG-LAND is that the land allocation model is coupled to the state-of-the-art land surface model, and that the impact of future climate and socio-economic changes on water resources and land use can be considered consistently. In general, future land-use changes are often assessed by using an IAM. However, as mentioned earlier, IAMs are not grid-based, but rather they divide the world into dozens of regions and describes the entirety of economic activity in these regions. Therefore, IAMs has a simplified description of the processes related to water resources and crop growth. In contrast, MIROC-INTEG-LAND provides capabilities to calculate complex physical processes over the land, and considers the changes in water resources, taking into account human activities such as irrigation and reservoir operation. Furthermore, process-based crop models allow for an explicit and detailed consideration of growth process of five different crops.

For the projection of future land use, IAMs usually 1) calculate the area of agricultural land by using yield information averaged over these regions based on the balance between supply and demand, and

2) allocate the agricultural land by using a downscaling approach (e.g., Hasegawa et al. 2017). As pointed out in previous studies (Alexander et al. 2017), the problem with this method is that it is does not allow an explicit consideration of spatiotemporal information such as yield and production cost when determining land use change. The Food Cropland Model in TeLMO addresses this issue by making it is possible to consistently consider the spaciotemporal information such as crop yields and the balance between supply and demand when allocating the agricultural land, by using the Food Cropland Down-scale Module and the International Trade Module as explained in Appendix B.

As for the projection of future land use change, TeLMO enables the calculation of future land use change as an offline simulation, by using the crop yield data calculated in advance. On the other hand, crop yield depends on water resource availability that is affected by the changes in soil physical processes due to future climate change, as well as the changes in irrigated cropland area caused by the increases in future food demands. MIROC-INTEG-LAND couples the models of land-physical processes, human water management, and crop growth processes with the land-use allocation model to consider these various interactions, as explained above.

In Section 6.4, we also compared the performance of TeLMO and AIM for the area of food cropland, pasture, and forest (Figure 7-9). The figures for pasture and forest area are added according to suggestions by reviewer #1.

*4) One concern I have about the model itself is the inconsistency between the crop model for growth/yield and the biogeochemical / biophysical model for cropland. You may not be able to fix this right now, but it is a problem that there are two different crop growth models to represent different processes of the same land area. In particular, your yield model does not have explicit fertilizer, but your VISIT model does. For a variety of reasons, the growth values will not be the same between the two representations, but they should be because the growth determines the geochemical and physical characteristics of the cropland. In the end your yields are not consistent with how cropland affects the geochemical and physical processes in the land model that will eventually feedback to the atmosphere. This should be fixed before full coupling with the atmosphere.*

In the revised manuscript, we added an explanation of the inconsistency among the sub-models in the integrated model. This is explained in accordance with your comment, "*It seems that there are two crop models: PRYSBI2 and also one in HiGWMAT. Please clarify how these are different and why they are separate.*", and "*page 8 line 14-16*" below.

*5) Some of the description is not very clear or complete for the reader to understand what has been done or how the components interact. See comments below for details on what needs clarification. In particular, it isn't clear how the non-cropland is affected by climate in this model.*

Explanation of the details in the sub-models and the interactions between the components are modified according to the suggestions. It is explained according to your comments, "*page 15, lines 19-21*", "*page 26, lines 8-24*", and "*page 27, lines 14-19*" below.

***Specific comments and suggestions***

***Abstract***

*The abstract is not clear about what is presented here or what the outcomes of evaluation are. While some interactions from climate to land allocation are included, feedbacks between the land system and climate are not because the atmospheric inputs are fixed. Be clear about the novelty here, and state how well the evaluation performs.*

Thank you very much for your suggestions. First, the abstract was modified to clarify the novelty of the model development as follows.

To investigate these interrelationships, we developed MIROC-INTEG-LAND (MIROC INTEGrated LAND surface model version 1), an integrated model that combines the land surface component of global climate model MIROC (Model for Interdisciplinary Research on Climate) with water resources, crop production, land ecosystem, and land use models. The most significant feature of MIROC-INTEG-LAND is that the land surface model that describes the processes of the energy and water balance, human water management, and crop growth incorporates a land use decision-making model based on economic activities. In MIROC-INTEG, spatially detailed information regarding water resources and crop yields is reflected in the prediction of future land use change, which cannot be considered in the conventional integrated assessment models.

In addition, we also state how well the evaluation performs and the outcomes of simulations. The final sentence of Abstract in the original manuscript as follows.

By evaluating the historical simulation, we have confirmed that the model reproduces the observed states well. The future simulations indicate that the changes in climate has significant impacts on crop yields, and thus on land use change. The newly developed MIROC-INTE-LAND could be combined with atmospheric and ocean models to develop an integrated Earth system model to simulate the interactions among coupled natural-human Earth system components.

***Introduction***

*page 2, line 21: Bond-Lamberty this reference is incorrect throughout*

The reference is modified in the main text.

**Model structure**

**Sub models**

*It seems that there are two crop models: PRYSBI2 and also one in HiGWMAT. Please clarify how these are different and why they are separate.*

The reason that different crop models are used for HiGWMAT and PRYSBI2 is that 1) HiGWMAT uses a crop model based on SWIM for the calculation of the irrigation process, and it has been validated that the water withdrawal of HiGWMAT in various regions is consistent with the statistical data, and 2) PRYSBI2 uses a crop model based on SWAT and crop yield in PRYSBI2 has been calibrated using the agricultural statistics. MIROC-INTEG-LAND uses different crop models to obtain realistic water withdrawal in HiGWMAT and to calculate realistic crop yields in PRYSBI2. The differences in the formulation between the crop models in PRYSBI2 and HiGWMAT are that the former uses more detailed crop modeling of the two-layer crop canopy, Farquhar photosynthetic $CO_2$ assimilation, and the reported planting date of Sacks et al. (2010), while the latter employs the simpler crop modeling of the single-layer crop canopy, radiation-use efficiency type biomass accumulation, and the hypothetical planting date that gives the highest yield under the given weather conditions. As pointed out by the reviewer, it is an important future work to tune the model parameters to obtain the realistic water withdrawal and crop yields by using a single crop model. In Section 3.1.2, this is explained as follows. The new reference (Okada et al. 2015) is added in the reference list.

The crop growth module is based on the H08 model (Hanasaki et al., 2008a, 2008b), where the crop vegetation formulations and parameters are adopted from the Soil and Water Integrated Model (SWIM) (Krysanova et al., 1998). The crop growth module in HiGWMAT estimates the cropping period necessary to obtain mature and optimal total plant biomass for 18 different crop types. Irrigation is activated during the entire growing season but only for the irrigated portion of a grid cell using a tile approach. Crop growth for the irrigation processes is simulated within the HiGWMAT model (i.e., independent of PRYSBI2).

The reason that different crop models are used for HiGWMAT and PRYSBI2 is that 1) HiGWMAT has been used a crop model based on SWIM, and it has been validated that the water withdrawal in various regions is consistent with the statistical data (Pokhrel et al. 2014), and 2) PRYSBI2 has been used a crop model based on SWAT, and crop yield in PRYSBI2 has been calibrated using the agricultural statistics (Sakurai et al. 2014). MIROC-INTEG-LAND uses different crop models to obtain realistic water withdrawal in HiGWMAT and to calculate realistic

crop yields in PRYSBI2. The differences in the formulation between the crop models in PRYSBI2 and HiGWMAT are that the former uses more detailed crop modeling of the two-layer crop canopy, Farquhar photosynthetic $CO_2$ assimilation, and the use of the reported planting date of Sacks et al. (2010), while the latter employs the simpler crop modeling of the single-layer crop canopy, radiation-use efficiency type biomass accumulation, and the hypothetical planting date that gives the highest yield under the given weather conditions (Okada et al. 2015).

**VISIT**

*page 8, lines 12-13: abandoned cropland recovers to mean biomass of what? and is always considered secondary, or can secondary land revert to primary?*

It is the natural vegetation in the same grid. This is explained in the revised manuscript as follows.

regrowth of abandoned croplands is also simulated as the recovery of the mean biomass of the natural vegetation in the same grid.

*page 8, lines 14-16: these fertilizer and crop calendar inputs seem inconsistent with the crop model. if the crop model doesn't use fertilizer inputs, then how are they used in VISIT? if crop growth is calculated with implicit fertilizer, then this specific nitrogen input doesn't match. And why would VISIT be using a crop calendar and not the crop model? your biogeochemical fluxes are not going to correspond with your crop growth.*

As described in Section 3.2 and Appendix A.7, PRYSBI2 describes the effects of fertilizer by technological factors without considering the fertilizer input process. MIROC-INTEG-LAND adopted this method because crop yields have been calibrated by using technological factors in PRYSBI2. On the other hand, VISIT considers the fertilizer input processes in the manner described in Section 3.4, and it has been validated that the calculated carbon and nitrogen cycle is consistent with various observations (Ito et al. 2017). Therefore, the handling of fertilizer is different between PRYSBI2 and VISIT. As the reviewer pointed out, it is important to ensure that the fertilizer processes is consistent between these sub-models. The text in Section 3.3 has been modified as follows.

In PRYSBI2, the effects of fertilizer are included in the technological factors, and crop yields are calibrated based on the technological factors, As described in Section 3.2 and Appendix A.7. On the other hand, VISIT has been applied and validated at various scales from flux measurement sites to the global scale (e.g., Ito et al., 2017) based on the treatment of fertilizer input, as described above. The consistent treatment of fertilizer processes in PRYSBI2 and VISIT should be important future work.

*Model coupling*

*Experimental settings*

*page 11, lines 3-19: which SSPs for which RCPs?*

In this study, we use outputs of the SSP2 scenario calculated by AIM/CGE (Fujimori et al. 2017). Since the RCP8.5 scenario is not available in SSP2, we use the output of the baseline scenario from AIM/CGE for the MIROC-INTEG-LAND calculation of RCP8.5. This is explained in the revised manuscript.

*Historical simulations*

*page 13, lines 1-3: The results in figure 5 do not all line up along the 1:1 line. You have to adjust this statement.*

We modified the statement in the revised manuscript as follows:

For all crops, most of the relationship between the simulated and reported data was distributed around the 1:1 line.

*page 14, lines 1-2: TeLMO crop area is not very similar to AIM, and it is more similar to FAO in most cases presented. Since AIM is a driver of TeLMO, more explanation is required here of why they are different. Furthermore, the similarity to FAO is more compelling as evidence for usability of TeLMO, than any similarity to AIM.*

Thank you very much for your suggestions. The difference between TeLMO and AIM/CGE is due to the difference in crop yield as well as the mechanism for the allocation of the agricultural land. As explained in Appendix B-1, TeLMO can consider the spatial distribution of crop yield when allocating the agricultural land. On the other hand, in integrated assessment models such as AIM/CGE, land use change is calculated by aggregating crop yield information in the regions (AIM/CGE divides the world into 17 regions). In large countries such as Australia, Brazil and Russia, the allocation method in TeLMO may show good performance. This is explained in the revised manuscript.

In MIROC-INTEG-LAND, TeLMO uses the food demand and GDP per capita calculated by AIM/CGE under the socio-economic scenario SSP2 (Fujimori et al., 2017). Therefore, the difference between TeLMO and AIM/CGE is due to the difference in crop yield as well as the mechanism for the allocation of agricultural land. As explained in Appendix B.1, TeLMO can consider the spatial distribution of crop yield when allocating agricultural land. On the other hand, in AIM/CGE, land use change is calculated by aggregating crop yield information in the regions where the model

calculation is performed (AIM/CGE divides the world into 17 regions). In large countries such as Australia, Brazil and Russia, the allocation method in TeLMO shows good performance.

***Future simulations***

*page 15, line 8: why the maximum value? this would underestimate the land area because each crop may be grown in a cell, with varying yields.*

*+Comments on Appendix B:*

*page 22, lines 7-11: it isn't clear why you are using the max yield, and it sounds like it is per crop here, while in the text it sounded like it was the max across crops. don't you need to apply each crop yield to its own prices in the 30sec cells to get distinct ASI values for crops? maybe a crop with a lower yield has higher ASI due to higher price. so are you essentially just selecting one crop for the half-degree cell? this will underestimate cropland.*

In the TeLMO food cropland model, the cropland area is calculated using the maximum value of the five crops in each grid. This formulation is due to the simplification of the model structure as described below. As pointed out by the reviewer, using the maximum yield to determine the cropland area would underestimate the cropland area. However, TeLMO applies an adjustment parameter ($C_j$ in Eq. B-1) so that the cropland area of the base year can be close to the observed (LUH) data. For this reason, cropland area is not necessarily underestimated by the model. This is explained as below.

If the cropland area predicted by TeLMO is $A_{TeLMO}$, the actual cropland area is $A_{Real}$, the maximum yield is $Y_{max}$, the actual yield that determines the cropland area is $Y_{Real}$, the food demand is $D$, and the adjustment parameter is $C_j$, then

$$A_{TeLMO} \sim \frac{D}{Y_{max}} \times C_j$$

$$A_{Real} \sim \frac{D}{Y_{Real}}$$

At the base year (2005), $A_{TeLMO} \sim A_{Real}$ and thus

$$C_j \sim \left(\frac{Y_{max}}{Y_{Real}}\right)_{y=2005}$$

Therefore, the ratio of $A_{TeLMO}$ and $A_{Real}$ except the base year $y \neq 2005$ is

$$\frac{A_{TeLMO}}{A_{Real}} \sim \left(\frac{Y_{max}}{Y_{Real}}\right) \Big/ \left(\frac{Y_{max}}{Y_{Real}}\right)_{y=2005}$$

Namely, the ratio of $A_{TeLMO}$ and $A_{Real}$ can be approximated as the ratio of $Y_{max}/Y_{Real}$ between the calculation and base years. Since the actual calculation of TeLMO considers the food trade and

allocates the cropland area at the grid level, this is not entirely the case. However, the cropland area in TeLMO is not necessarily underestimated because of the adjustment parameter.

As the reviewers point out, it is possible to formulate the prices for different crops and allocate the cropland areas according to agricultural suitability indices. In that case, it is necessary to increase the number of sectors in the general equilibrium model in the International Trade Module, and solve the prices for each sector. In fact, in the course of the development of TeLMO, we tried to determine the price for each crop and allocate cropland area according to each agricultural suitability index. However, the results obtained in this formulation were roughly similar to those obtained by the current formulation. On the other hand, in some cases, the solution did not converge due to the complexity of the general equilibrium model (particularly when demand increased). For this reason, we decided to adopt the current formulation. However, as the reviewers point out, calculating cropland area for each crop is a very important future work. In the main text, it is described as follows.

As described in Section 3.4 and Appendix B, TeLMO uses the yield calculated by PRYSBI2 (grid maximum value as shown in Figure 11f) and the food demand output of AIM/CGE.

The description in Appendix B.1.1 is modified as follows.

In TeLMO, total food cropland area is projected by using the maximum yield across the five cereal types (winter and spring wheat, maize, soybean, and rice). The reason for this formulation is explained in Section B.1.2. $y_j$ in Eq. (B-1) is calculated from the yields of the five cereals types by PRYSBI2.

At the end of Appendix B1.2, it is explained as follows.

As explained in Section B.1.1, TeLMO uses the maximum yield of five cereals types to project the total cropland area. Alternatively, it is possible to increase the number of agricultural sectors in Eqs. (B-3) to (B-12), solve the prices for each crops, and allocate the cropland area according to the ASIs for each crop. Although we attempted this formulation in the course of our development of TeLMO, it was found that the results were similar to those obtained from the current formulation. On the other hand, the solution of general equilibrium models did not converge in some cases because the number of sectors increases in the equations. For this reason, we decided to adopt the current formulation, while recognizing that calculating cropland areas for each crop is an important future work.

*page 15, lines 19-21: what is the basis for your bioenergy crop calculations here?*

The method for the calculation of bioenergy crop yield is described in the revised manuscript as follows.

For biofuel crop yield $y_{bio,j}$, the yield for miscanthus or switchgrass, whichever is greater in a given cell, is calculated for the entire globe by using the biofuel crop model developed in Kato and Yamagata (2014). The biofuel crop model in Kato and Yamagata (2014) considers the future changes in climate based on the RCP scenarios. In this study, we also consider the future changes in fertilizer input based on the SSPs adopted in Mori et al. (2018). Because of the uncertainty in future fertilizer application for crop management, we set the high end of the N fertilizer input threshold according to Tilman et al. (2011). The nitrogen fertilizer application was set to increase from the current level according to the increasing rate of GDP in the SSP2 scenario up to 160 kg N ha$^{-1}$ yr$^{-1}$ if the fertilizer input at the country level was below 160 kg N ha$^{-1}$ yr$^{-1}$ in the 2000s. Also, the phosphorus fertilizer input in each country was set to follow the same annual increase rate as the nitrogen fertilizer application.

*page 16, lines 23-24: Not sure what you mean here. You haven't untangled land use effects here, just showed results of land area changes and the biomass affected. Many ecosystem and earth and integrated assessment models do this. what is unique here? If you were to show how the land allocation and the biomass effects differed due to the climate effects on yields, then this would show the benefit of this model. To do this you have to do another set of runs where the yields are the base year yields plus the non-climate changes in yield, and compare these to the runs you have done. Alternatively, you could show how the inclusion of water availability in determining irrigation changes the crop area/production by turning off the irrigation dependence on the available water.*

Thank you very much for your very constructive suggestions. According to the reviewer's comments, we performed additional simulations where climate effects on yield are switched off (new Figure 16) to shows the benefits of this model. We removed this sentence of the original manuscript, and added the new paragraphs at the end of Section 7. In fact, we are now preparing a paper using MIROC-INTEG-LAND to investigate the impacts of various natural and socio-economic factors (climate, irrigation, fertilization effects, population, food demands, etc.) on land use and land ecosystems. Therefore, details of the analysis on interactions between sub-models will be presented in the next paper. In the revised manuscript, we added discussions as follows.

Figure 16 shows the results of simulations to evaluate the effects of climate change on crop yield and land use. In Figure 16, the RCP8.5 simulations with climatic factors (temperature, water vapor, wind speed, soil moisture, soil temperature) and $CO_2$ concentration fixed at 2006 (noCL+noFE), those

with climatic factors fixed (noCL), and those with varying climate and $CO_2$ concentration (CL+FE) are compared. The CL+FE simulations are the same as the RCP8.5 results shown in Figure 12. As shown in Figure 16a, in the noCL+noFE simulations, the crop yield was much lower than that in the CL+FE simulations. In the noCL+noF experiment, the crop yield is increased due to the technological development (Section 3.2 and Appendix A.7). The reason that the yield in the CL+FE experiment is higher than that in noCL+noFE experiments is that the crop yield increases due to the fertilization effect in the former. In the noCL+FE experiment (Figure 16), the crop yield is approximately 1.7 times as large as in the noCL+noFE experiment. Although there is a great deal of uncertainty in the treatment of fertilizer effects in crop models (Sakurai et al. 2014), the increase in crop yields is significant in the simulations by MIROC-INTEG-LAND.

As shown in Figure 16a, the crop yield is significantly smaller in the CL+FE than in the noCL+FE experiment. This result indicates that climate change can significantly reduce crop yields. One of the reasons for this reduction in crop yield is that the growing season is shortened due to a rise in surface air temperature, adversely affecting the growth of crops (Sakurai et al. 2014). The impact of climate change on crop growth increases with increasing temperature, and in 2100 crop yield in the CL+FE experiment is projected to decrease roughly 60% relative to the yields in the noCL+FE experiments.

Due to the changes in crop yields caused by the changes in climate and fertilization effects, future cropland area will also change significantly. As shown in Figure 16b, the noCL+noFE experiment requires more cropland area compared to the CL+FE experiments, due to the smaller increase in crop yields (Figure 16a). As explained in Figure 12, cropland area could expand in the first half of the 21st century to meet the increasing demand due to population growth, and then gradually decrease in the latter half of the 21st century. On the other hand, in the noCL+FE experiments, the increase in crop yield is larger than that in the CL+FE experiment, and thus the cropland area in 2100 will be about 76% of that in 2005. In sum, it is found that the changes in climate and fertilization effects have large impacts on crop yields and land use change.

Caption of Figure 16 is added as follows.

Figure 16: Time series of changes in a) cropland yield (maximum across five crops at each grid, t/ha), and b) food cropland area (a fraction of total land area) based on the forcings of the five climate models under the RCP8.5 scenario. Simulations with climatic 5 factors and $CO_2$ concentration fixed at 2006 (light green, noCL+noFE), those with climatic factors fixed (cyan, noCL), and those with varying climate and CO2 concentration (red, CL+FE).

***Implications***

*page 16, lines 30-31: this is where things get more interesting*

Thank you very much again for your suggestions. The analysis of the interactions and feedbacks is presented at the last few paragraphs of Section 7 in the revised manuscript. According to this modification, the first paragraph of Section 8 is revised as follows.

With MIROC-INTEG-LAND, it is possible to calculate the interaction between climate, water resources, crops, land use, and ecosystems. The discussion in Section 7 suggests the type of feedback processes that can occur. While this study showed only the results of the SSP2 scenario, in the SSP3 scenario, where the world is divided, the demand for food will be greater and more cropland area will be needed (O'Neill et al., 2017). Investigating the impacts of various natural and socio-economic factors (climate, irrigation, fertilization effects, population, food demands, etc.) on land use change and land ecosystems is an important future research direction as an extension of the present study.

*page 17, lines 6-8: it seems like you have outputs in this paper that could be used for this*
*In the original manuscript*
*MIROC-INTEG1 can also be used to evaluate the effectiveness of climate mitigation measures by quantitatively evaluating the cultivated land area of biofuel crops and the budget of greenhouse gases via the terrestrial ecosystem model, VISIT.*
Thank you very much for your suggestion. This sentence is removed in the revised manuscript.

***Figures and tables***
*Figure 3 What do you mean by multi-model mean? You are using only one model here.*
We modified the manuscript as follows.

Simulated results are the average of five climate model simulations. Grey shading indicates the uncertainty range shown by one standard deviation from the mean.

*Figure 10 What is the vertical axis? Is it the annual change in cropland area as a fraction of total land area, or is it the cropland in that year as a fraction of total land area?*
It is the cropland in that year as a fraction of total land area. The Figure caption of Figure 12 (Figure 10 in the original manuscript) is described in the revised manuscript.

Figure 12: Time series of changes in cropland area relative to land area based on the forcings of the five climate models. The vertical axis is the cropland area in that year as a fraction of total land area.

***Appendix A***

*page 17, line 28: what are these? can you give the types of data if not able to list them all here?*

We modified "A.1 Input data" in the revised manuscript as follows.

As input data, the PRYSIB2 Version 2.2 uses the planting and harvesting date (Saccs et al. 2008), soil field capacity (Scholes et al. 2011), and atmospheric data (average, maximum and minimum daily temperature, daily shortwave and longwave radiation, daily humidity, and $CO_2$ concentration). We use the same atmospheric data as HiGWMAT described in Section 5 (i.e., ISIMIP fast track data by Hempel et al. 2013).

*page 19, line 3: i think you mean water stress and shouldn't this go into section A.5?*

As you pointed out, the water stress is explained in A5. It is described in the revised manuscript.

*page 19, line 11: so there is a cold stress factor applied on top of the temperature dependent equation?*

We removed the sentence in the original manuscript "*For soybean, we considered cold stress in addition to the temperature stress explained above. The details are the same as in version 2.0.*". The cold stress was not considered in the MIROC-INTEG because of technical reasons.

*page 20, lines 7-16: it isn't clear what this is or how it is applied. is it a fraction that is applied to a given yield and then added to that yield? what are equations 32 and 33?*

"equations 32 and 33" in the original manuscript were Eq. (A-5) and (A-6). The trend of the parameter relevant to agricultural management is expressed by substituting Θ obtained in Eq. (A-14) into Eq. (A-5) and (A-6).

**Appendix B**

*page 22, lines 7-11: it isn't clear why you are using the max yield, and it sounds like it is per crop here, while in the text it sounded like it was the max across crops. don't you need to apply each crop yield to its own prices in the 30sec cells to get distinct ASI values for crops? maybe a crop with a lower yield has higher ASI due to higher price. so are you essentially just selecting one crop for the half-degree cell? this will underestimate cropland.*

This is addressed in the above comments for page 15, line 8.

*page 25, lines 20-21: for bionenergy crops, does this mean that these are fixed values, or are they dependent on the atmospheric inputs? More info is needed here.*

We consider changes in climate and socio-economic factors for the calculation of bioenergy crops. The manuscript is modified as follows.

For biofuel crop yield $y_{bio,j}$, the yield for miscanthus or switchgrass, whichever is greater in a given cell, is calculated for the entire globe by using the biofuel crop model developed in Kato and Yamagata (2014). The biofuel crop model in Kato and Yamagata (2014) considers the future changes in climate based on the RCP scenarios. In this study, we also consider the future changes in fertilizer input based on the SSPs adopted in Mori et al. (2018). Because of the uncertainty in future fertilizer application for crop management, we set the high end of the N fertilizer input threshold according to Tilman et al. [2011]. The nitrogen fertilizer application was set to increase from the current level according to the increasing rate of GDP in the SSP2 scenario up to 160 kg N ha$^{-1}$ yr$^{-1}$ if the fertilizer input at the country level was below 160 kg N ha$^{-1}$ yr$^{-1}$ in the 2000s. Also, the phosphorus fertilizer input in each country was set to follow the same annual increase rate as the nitrogen fertilizer application.

*page 26, lines 8-24: Is this NPP for pasture calculated beforehand, offline? Does it include matching climate drivers to the scenarios here? Or are these NPP numbers independent of the climate change?*

The NPP used in the pasture model was calculated in offline simulations in advance, with fixed boundary conditions at 2005. This is explained in the revised manuscript.

The results of an off-line simulation by VISIT (Ito and Inatomi 2012) assuming the entire world to be grassland are used here for $NPP_j$. The boundary condition of the VISIT off-line simulations is fixed at year 2005.

*page 27, line 10: where is the base-year managed forest area from?*

The base year of the managed forest is also 2005. It is added in the revised manuscript.

Moderate Resolution Imaging Spectroradiometer (MODIS) satellite data (Friedl et al., 2010) are used for the base-year forest area (2005)

*page 27, lines 14-19: also for forest, are these NPP values calculated beforehand, offline? and are there matching values corresponding with the scenarios here, or are they a single set of values for all sims?*

As in the case of the pasture model, the NPP in the Forest Model is calculated by offline simulations with fixed boundary conditions. This is explained in the revised manuscript.

calculated by VISIT (Ito and Inatomi 2012) off-line simulations assuming the entire world to be forest with fixed boundary conditions (2005).

---

## Author Response (AR2)

**Report #1**

*P18 L26–30: It should be made clear that this is referring to CO2 fertilization, not application of fertilizer.*

Thank you very much for your comments. We changed this to read "$CO_2$ fertilization effect" in the revised manuscript.

**Report #2**

*Overall response*

*This is a revision of a previously reviewed manuscript. The authors have done a good job responding to the previous review, but some further clarifications and discussions need to be made, and some additional strengthening of the results would be helpful.*

1) *The novelty of this work is still understated. It is shown in figure 16b, but is not explained or highlighted as such. Similar plots of irrigation demand across these same sims should show further importance of incorporating climate/co2 into the land resource allocation, which is a primary new feature of miroc-integ-land that has very limited representation in global modeling.*

2) *please see comments below regarding further clarification and discussion.*

I sincerely appreciate that the reviewer carefully read the manuscript and gave us valuable comments and suggestions. According to your suggestions, we added a new figure, strengthened the discussion, and provided clarifications in the revised manuscript.

*Specific comments and suggestions:*

*Abstract*

*page 1, line 31: I think this should also be labelled as 'MIROC-INTEG-LAND' because this is what you are running here, with offline climate forcing.*

This term has been modified as suggested.

*page 2 line 2: You should expand on this statement. this is the real novelty of this work. The land allocation (and likely irrigation demand - do you have these outputs for figure 16?) is very different when the climate/co2 effects are included. this is what makes mirocl-integ-land different from the current IAM scenario formulation.*

Thank you very much for this suggestion. We added the future changes in irrigation demand in Figure 16. The features of changes in land allocation and irrigation water demands are also discussed in Section 7, and their implications are discussed in Section 8.

*Model structure*

*It would be more clear if figure 1 included the names of the sub-models discussed here on pages 4*

*and 5. The names can be in the figure or in the caption.*

The names of the models are explained in the caption of Figure 1.

**Sub models**

*page 7, lines 15-27: What exactly does HiGWMAT contribute to the crop growth model and the rest of the biogeochemical and biophysical system? Here you mention cropping period and growth, but growth is simulated separately by PRYSBI2. And in the next section you mention only water stress from HiGWMAT, and not cropping period. So, what does PRYSBI2 use? And are the biophysical fluxes in the land model updated after HiGWMAT determines water management/use?*

In MIROC-INTEG-LAND, the crop scheme used in HiGWMAT to estimate irrigation water is different from that used in PRYSBI2 to determine crop yields. The description on page 7 line 15, pointed out by the reviewer, is an explanation of the crop scheme that used to estimate the irrigation water. The cropping period used for estimating irrigation water in HiGWMAT is also different from the cropping period used for calculating crop yields in PRYSBI2. In PRYSBI2, the cropping period is determined by giving the planting and harvesting date based on the data of Sacks et al. (2010).

In the revised manuscript, it is explained that the description in Section 3.1.2 pertains to the cropping period used to estimate irrigation water in page 7, line 23. In addition, it is explained that the crop scheme used for estimating irrigation water (HiGWMAT) and the crop scheme used for obtaining crop yield (PRYSBI2) are different, in page 7, line 28. Furthermore, the cropping period of PRYSBI2 is explained in page 8 line 3.

For the explanation of cropping period in PRYSBI2, Section 3.2 PRYSBI2 and Appendix A.2 have been modified in the revised manuscript.

The role of HiGWMAT in MIROC-INTEG-LAND is to calculate human water management/use (3.1.2 Human water management scheme) and to calculate the energy and water budget on the land surface (3.1.1 MATSIRO land surface model). In HiGWMAT, the biophysical fluxes are updated after water management/use is determined. In the revised manuscript, this point is explained on page 6, line 23.

*page 8, lines 14-15: Does PRYSBI2 use only soil moisture from HiGWMAT, or does it use cropping period also? Later you state that it is just soil moisture and temperature.*

As explained above, the cropping periods of PRYSBI2 are different from that in HiGWMAT. In addition, PRYSBI2 uses soil moisture and temperature to calculate water stress and crop yields. This is explained in the revised manuscript on page 8, line 26.

**Numerical coupling**

*pages 10-11: Some of the important info wanted in the previous section is here, but the overall*

*connections are not clear. For example, some of the variables passed between models are listed. Are these complete lists? It is still unclear which models are responsible for which overall land outputs. Does VISIT do all mass and energy land-atmosphere-flux outputs, or just the carbon and nitrogen cycles?*

All of the variables passed between sub-models are listed in Section 4. These are also explained in Figure 2. In order to explain which models are responsible for which overall land outputs, we added a description of the roles and outputs of each sub-model in Section 3.

*Figure 2 indicates that VISIT gets soil moisture and temperature from HiGWMAT, but the text states that there is no communication between these models.*

There is no communication between VISIT and HiGWMAT in this version of MIROC-INTEG-LAND. The connection in Figure 2 has been changed to reflect this.

*What appears to take place is that TeLMO first estimates land use, then HiGWMAT estimate water use, then PRYSBI2 estimates crop growth/yields, which then feedback to TeLMO for updated future land use estimates. When does VISIT get the TeLMO outputs? Before or after being informed by the water and crop models? And where are output water flux variables calculated? This flow needs to be completed and made clear.*

As described above, the roles of sub-models are explained more clearly in Section 3.1-3.4, in order to clarify the overall flow of the calculation. The outputs of energy water flux variables are calculated in HiGWMAT. This is also explained in Section 4 "Numerical procedure of model coupling", (2) HiGWMAT + PRYSBI2.

*Including a more detailed discussion of how the final surface model (VISIT) doesn't use the more detailed models to determine crop growth and water exchange and eventually land-atmosphere fluxes and carbon storage, but only the land use estimated from the more detailed models.*

VISIT receives only the output of land use change by TeLMO and is used to calculate the carbon and nitrogen cycle. For this reason, the energy and water balances are calculated independently within VISIT. The initial goal of model development was to use the energy and water budget calculated in HiGWMAT to calculate carbon and nitrogen cycles in VISIT, as the reviewer point out. However, as described in the original manuscript, the model structure of VISIT was very different from the model structure of HiGWMAT, so it was difficult to achieve this goal. Specifically, because HiGWMAT is based on MATSIRO being included in the climate model, it calculates the latitude-longitude loop in the time loop, but VISIT calculates the time loop in the latitude-longitude loop because there is no horizontal interaction in the model. We tried to rearrange the order of time and space loops in VISIT and combine them with HiGWMAT, but this was unsuccessful because VISIT is a huge program that

computes various complicated processes. Consequently, we decided that VISIT should use only the TeLMO output to calculate the carbon-nitrogen cycle. In the current version of MIROC-INTEG-LAND, we first calculate the TeLMO-HiGWMAT-PRYSBI2 until 2100, and then perform VISIT calculation from the preindustrial period (including spin-up simulations) to the end of the 21st century using the TeLMO output; TeLMO is thus only used for the future period, and LUH data is used for other periods). VISIT has been used to perform a variety of calculations for ISIMIP etc. (e.g., Ito et al. 2020, Pronounced and unavoidable impacts of low-end global warming on northern high-latitude land ecosystems. *Environmental Research Letters*), and thus MIROC-INTEG-LAND uses the original VISIT structure to calculate the carbon and nitrogen cycle.

As described in Section 8, we are developing MIROC-INTEG-ES, in which the human activity models in this study are coupled to a state-of-the-art earth system model (MIROC-ES2L, Hajima et al. 2020). In MIROC-INTEG-ES, the inconsistency issue of the energy and water budget between VISIT and HiGWMAT has been resolved. In the revised manuscript, this is explained in Section 4, (4) VISIT.

***Experimental settings***
*What are the historical isimip forcings? figures 3 and 5 suggests that there are multiple realizations of the historical forcings.*
In ISIMIP, the historical and future climate simulations by five global climate models (GCMs) with bias correction are distributed as the forcing data. The methodology of bias correction is described in Hempel et al., (2013). These aspects are explained in the revised manuscript.

***Historical simulations***
*page 14, lines 6-7: If the model output and the reference data are on grids, why do you use another data set to aggregate to country scale? Is this just to get a ratio of physical area to harvested area to comparison with FAO?*
Yes, it is just to get a ratio of cropland/pasture/forest area to perform a comparison with FAO. This is explained in the revised manuscript.

*page 15, lines 21-25: if telmo matches aim regions, why are austaralia and russia so different from aim?*
TeLMO uses the food demand calculated by AIM/CGE (17 regions) and allocates the cropland area with a resolution of 0.5 degrees. On the other hand, the AIM/CGE results in Figure 7 are based on the 0.5-degree downscaled land use data using the AIM/CGE output (the methodology is described in Fujimori et al. 2017a). This is explained in the revised manuscript. The cropland area of Australia and Russia in Figure 7 differ among TeLMO and AIM/CGE, partly because the land allocation

methods of TeLMO and AIM/CGE (Fujimori et al. 2017a) are different. The reason for the difference between TeLMO and AIM/CGE is discussed in the third paragraph of page 16. The reference Fujimori et al. 2017a is also added.

*page 15, lines 25-29: there are some large differences for forest in usa and australia. why is this? With the apparent country-level differences in pasture and forest it is not correct to state that telmo, aim and fao closely agree at the regional scale. This statement needs to be tempered and the differences acknowledged.*

The difference between TeLMO and FAO is likely because TeLMO refers to MODIS, but not to FAO. This is explained in the revised manuscript. The statement "Overall, TeLMO, AIM, and FAO closely agree at the regional scale." related to a previous version, which only included cropland area (Figure 7). That statement has been removed from the revised manuscript

***Future simulations and interaction of submodels***

*The isimip1 climate forcings do not match with the SSP scenarios you use. Not only were the climate forcings created from different IAMs, but the current SSP2 formulations are different than the socio-economic scenarios used for the RCPs from isimip. So you have inconsistencies in your forcing data. You should be using the CMIP5 SSP socio-economic drivers. This may not be a huge issue here because the atmosphere is not coupled to the land, which means that your climate drivers are somewhat independent of the land processes. If you cannot use the more closely matched driving data, you need to explain the mismatch and why it it exists and what impact it may have on your results. For example, while you can examine the effects of different RCP forcings on the land surface, these forcings are not necessarily consistent with the land processes, not only because the land model is not coupled to the atmosphere, but because the climate forcings do not match the land activities to begin with.*

Thank you very much for this suggestion. It is true that the socio-economic scenario in ISIMIP1 is not consistent with SSPs because it is based on the CMIP5 simulations. On the other hand, in ISIMIP3, the climate forcing will be based on the CMIP6 simulations. The ISIMIP3 forcing data has only recently been published. This is explained in Section 5, Experimental setting.

*page 16, line 6 and beyond: Figure 10, 10a, 10b, 10c*
*page 16, lines 16 and beyond: Your figures are out of order in referenced incorrectly. Ensure that they are numbers and referenced in order.*

Thank you very much for this correction. The numbers of the figures have been corrected in the revised manuscript.

*page 18 lines 20 forward: Do you have the irrigation results for these sims as well? figure 16b shows the real novelty of your development. The more outputs you can show that vary based on climate/co2 info and that matter for projecting human resource use, the better.*

Thank you very much for your suggestion. We added a new Figure 16c and discussed the impact of climate and CO2 fertilizer effects on irrigation demands in the last three paragraphs in Section 7.

**Implications and future research**

*So here you want to highlight how irrigation demand and cropland allocation change significantly if climate and/or CO2 effects are considered in this process. This interaction is the real novelty of miroc-integ-land. The IAMs are allocating land based on the green yield line in figure 16. But using miroc-integ-land, you get a very different allocation.*

We really appreciate the suggestion that we should highlight the novelty of MIROC-INTEG-LAND. In the revised manuscript, we described how MIROC-INTEG-LAND can be applied to investigate interactions in which climate change causes changes in crop yields, land use, terrestrial ecosystems, and water resources. This clarification appears in the first paragraph of Section 8.

**Tables and figures**

*Figure 8: Caption needs to be edited to state "ratio of pasture area to total area"*
*Figure 9: Caption needs to be edited to state "ratio of forest area to total area"*

Thank you very much. These captions have been corrected as suggested.

[revised manuscript text omitted]